# Improved double Fourier series on a sphere and its application to a semi-implicit semi-Lagrangian shallow water model

Hiromasa Yoshimura[1]

[1]Meteorological Research Institute, Japan Meteorological Agency, 1-1 Nagamine, Tsukuba, Ibaraki 305-0052, Japan

*Correspondence to*: Hiromasa Yoshimura (hyoshimu@mri-jma.go.jp)

**Abstract.** One way to reduce the computational cost of a spectral model using spherical harmonics (SH) is to use double Fourier series (DFS) instead of SH. The transform method using SH usually requires $O(N^3)$ operations, where $N$ is the truncation wavenumber, and the computational cost significantly increases at high resolution. On the other hand, the method using DFS requires only $O(N^2 \log N)$ operations. This paper proposes a new DFS method that improves the numerical

stability of the model compared with the conventional DFS methods by adopting the following two improvements: a new expansion method that employs the least-squares method (or the Galerkin method) to calculate the expansion coefficients in order to minimize the error caused by wavenumber truncation, and new basis functions that satisfy the continuity of both scalar and vector variables at the poles. Partial differential equations such as the Poisson equation and the Helmholtz equation are solved by using the Galerkin method. In the semi-implicit semi-Lagrangian shallow water model using the new

DFS method, the Williamson test cases and the Galewsky test case give stable results without the appearance of high-wavenumber noise near the poles, even without horizontal diffusion and without a zonal Fourier filter. In the Eulerian advection model using the new DFS method, the Williamson test cases 1, which simulates a cosine-bell advection, also gives stable results without horizontal diffusion but with a zonal Fourier filter. The shallow water model using the new DFS method is faster than that using SH, especially at high resolutions, and gives almost the same results, except that very small

oscillations near the truncation wavenumber in the kinetic energy spectrum appear only in the shallow water model using SH.

## 1 Introduction

Global spectral atmospheric models using the spectral transform method with spherical harmonics (SH) as basis functions are widely used. They are used in the Japan Meteorological Agency (JMA, 2019) and the Meteorological Research Institute (MRI; Yukimoto et al., 2011, 2019) for a range of applications, including operational weather prediction, operational

seasonal prediction, and global warming projection. The spectral model has the advantage that the horizontal derivatives are accurate, and the semi-implicit scheme, which improves numerical stability, can be easily applied because the Helmholtz equation and the Poisson equation are easily solved in spectral space. The application of the semi-implicit semi-Lagrangian scheme allows for timesteps longer than the Courant–Friedrichs–Lewy (CFL) condition, which makes the model computationally efficient. In the spectral model using SH, the Legendre transform used in the latitudinal direction

significantly increases the computational cost at high resolutions since the Legendre transform usually requires $O(N^3)$ operations and $O(N^3)$ memory usage (unless using the fast Legendre transform or on-the-fly computation of the associated Legendre functions shown below), where $N$ is the truncation wavenumber. One way to reduce the operation count and the memory usage at high resolutions with large $N$ is to use the fast Legendre transform (Suda, 2005; Tygert, 2008; Wedi et al.,

2013; Wedi 2014), which requires only $O(N^2 (\log N)^3)$ operations and also effectively reduces the memory usage. In the fast Legendre transform, the threshold parameter affecting the accuracy-cost balance is chosen so that a loss of accuracy is sufficiently small. Dueben et al. (2020) presented global simulations of the atmosphere at 1.45 km grid-spacing in the SH model using the fast Legendre transform. Another approach to improve the Legendre transform is on-the-fly computation of the associated Legendre functions (Schaeffer, 2013; Ishioka, 2018), which still requires $O(N^3)$ operations but requires only

$O(N^2)$ memory usage. This small memory usage also contributes to speeding up calculations by taking advantage of the cache memory.

Alternatively, we can use double Fourier series (DFS) as basis functions to reduce the operation count and the memory usage in the global spectral model. In the DFS model, the fast Fourier transform (FFT; Cooley and Tukey, 1965; Swarztrauber, 1982) is used not only in the longitudinal (zonal) direction but also in the latitudinal (meridional) direction.

The FFT requires only $O(N^2 \log N)$ operations and $O(N)$ or $O(N^2)$ memory usage, and it is faster than the fast Legendre transform.

In DFS models (and also in SH models), the scalar variable $F(\lambda, \theta)$ is zonally expanded as

$$F(\lambda, \theta) \cong \sum_{m=-M}^{M} F_m(\theta) e^{im\lambda}, \tag{1}$$

where $\lambda$ is longitude, $\theta$ is colatitude, and $M$ is the zonal truncation wavenumber. Several methods have been proposed for

meridional expansion with DFS. Merilees (1973b), Boer and Steinberg (1975), and Spotz et al. (1998) performed the Fourier transform meridionally along a great circle. Spotz et al. (1998) showed that by using the spherical harmonic filter, the explicit DFS shallow water model using the pseudo-spectral method can produce results comparable with the SH model in terms of accuracy and stability. However, the spherical harmonic filter consists of the forward SH transform (from grid space to spectral space) followed by the inverse SH transform (from spectral space to grid space), which increases the

computational cost.

Orszag (1974) and Boyd (1978) expanded $F_m(\theta)$ meridionally as

$$F_m(\theta) \cong \begin{cases} f_m(\theta) & \text{for even } m, \\ \sin\theta \, f_m(\theta) & \text{for odd } m, \end{cases} \tag{2a}$$

$$f_m(\theta) \equiv \sum_{n=0}^{N} f_{n,m} \cos n\theta, \tag{2b}$$

where $N$ is the meridional truncation wavenumber. The coefficients $f_{n,m}$ for odd $m$ are calculated by the forward Fourier

cosine transform of $F_m(\theta)/\sin\theta$. Orszag (1974) imposed the following conditions at the poles:

$$f_m(0) = 0 \text{ and } f_m(\pi) = 0 \text{ for } |m| \geq 2, \tag{3}$$

which can be expressed in terms of the expansion coefficients $f_{n,m}$ as

$$\sum_{\substack{n=0 \\ n \text{ is even}}}^{N} f_{n,m} = 0 \text{ and } \sum_{\substack{n=1 \\ n \text{ is odd}}}^{N} f_{n,m} = 0 \text{ for } |m| \geq 2. \tag{4}$$

Satisfying the above conditions ensures that the scalar variable $F(\lambda, \theta)$ and its gradient $\nabla F$ are continuous at the poles. In Orszag (1974), only $f_{N-1,m}$ and $f_{N,m}$ were modified to satisfy Eq. (4), but this is not the best way to satisfy the same conditions as Eq. (3) or Eq. (4), as will be shown in Sect. 4.

Yee (1981), Akahori et al. (2001) and Layton and Spotz (2003) expanded $F_m(\theta)$ as

$$F_m(\theta) = \begin{cases} \displaystyle\sum_{n=0}^{N} F_{n,m} \cos n\theta & \text{for even } m, \\ \displaystyle\sum_{n=1}^{N} F_{n,m} \sin n\theta & \text{for odd } m. \end{cases} \tag{5}$$

In the semi-implicit semi-Lagrangian shallow water model in Layton and Spotz (2003), the spherical harmonic filter was applied to the prognostic variables for stability and accuracy. Layton and Spotz (2003) explained that the expansion with Eq. (5) permits discontinuity at the poles and nonisotropic waves, which may lead to a prohibitive timestep restriction and numerical instability, and these problems can be avoided by applying the spherical harmonic filter.

Cheong (2000a, 2000b) proposed expanding $F_m(\theta)$ as

$$F_m(\theta) \cong \begin{cases} \displaystyle\sum_{n=0}^{N} F_{n,m} \cos n\theta & \text{for } m = 0, \\ \displaystyle\sum_{n=1}^{N'} F_{n,m} \sin n\theta & \text{for odd } m, \\ \displaystyle\sum_{n=1}^{N'} F'_{n,m} \sin \theta \sin n\theta & \text{for even } m \, (\neq 0). \end{cases} \tag{6}$$

The meridional basis functions $\sin \theta \sin n\theta$ for even $m \, (\neq 0)$ are different from Eq. (5). The coefficients $F'_{n,m}$ for even $m \, (\neq 0)$ are calculated by forward Fourier sine transform of $F_m(\theta)/\sin \theta$. The basis functions in Eq. (6) automatically satisfy the same conditions at the poles as Eq. (3) for even $m$, and guarantee the continuity of the scalar variable $F$ at the poles, which is an advantage compared with the basis functions in Eq. (5). On the other hand, Eq. (6) does not automatically satisfy the conditions in Eq. (3) for odd $m$, and does not guarantee the continuity of $\nabla F$ at the poles. The shallow water model and the vorticity equation model using a semi-implicit Eulerian scheme ran stably without the spherical harmonic filter by using high-order horizontal diffusion with $O(N^2)$ operations to smooth out the high-wavenumber components (Cheong, 2000b; Cheong et al., 2002; Kwon et al., 2004). The semi-implicit Eulerian hydrostatic atmospheric model also ran stably with high-order horizontal diffusion (Cheong, 2006; Koo and Hong, 2013; Park et al., 2013). However,

the computational results of these models appear to be a little different from (slightly worse than) the models using SH. One reason for this seems to be the appearance of high-wavenumber oscillation resulting from the meridional wavenumber truncation with $N = N' \cong 2J/3$ or $J/2$ for even $m$ ($\neq 0$) (See Sect. 4), and the use of high-order horizontal diffusion to smooth out the oscillation, where $J$ is the number of grid points in the latitudinal direction.

Yoshimura and Matsumura (2005) and Yoshimura (2012) stably ran the two-time-level semi-implicit semi-Lagrangian hydrostatic and nonhydrostatic atmospheric models using the DFS basis functions of Cheong in Eq. (6). These models used the same fourth-order horizontal diffusion as the SH models, and did not require the spherical harmonics filter or the strong high-order horizontal diffusion for stability. The numerical stability of the models was improved by adopting the followings:

1. The semi-Lagrangian scheme is used, which avoids the numerical instability due to the nonlinear advection term.

2. The meridional truncation with $N = J - 1$ and $N' = J$ is used, which enables to reconstruct accurately the given grid-data with the expansion coefficients (Cheong et al., 2004) and avoid the error due to the meridional truncation.

3. $U = u \sin \theta$ and $V = v \sin \theta$ instead of $u/\sin \theta$ and $v/\sin \theta$ are transformed from grid space to spectral space, where $u$ is the zonal wind and $v$ is the meridional wind.

The results of these models were very similar to those of the SH models. However, we found the following two problems in these models:

1. High wavenumber noise appears near the poles.

2. The meridional truncation wavenumber $N'$ needs to be equal to $J$ for even $m$ ($\neq 0$) because $N' < J$ (e.g., $N' \cong 2J/3$) for even $m$ ($\neq 0$) causes the high-wavenumber oscillation (See Sect. 4) and the numerical instability.

To solve these problems, we propose a new DFS method that adopts the following two improvements:

1. A new expansion method to calculate DFS expansion coefficients of scalar and vector variables, which adopts the least-squares method (or the Galerkin method) to minimize the error due to the meridional wavenumber truncation.

2. New DFS basis functions that automatically satisfy the pole conditions in Eq. (3), which guarantee continuity of not only scalar variables but also vector variables at the poles.

We also use the Galerkin method to solve partial differential equations such as the Poisson equation, the Helmholtz equation, 25 and the shallow water equations.

     Section 2 describes the arrangement of equally spaced latitudinal grid points used in the new DFS method. Section 3 describes the details of the new DFS method using the new DFS expansion method and the new DFS basis functions, and also includes the essential summary of the new DFS method. Section 4 examines the error due to the wavenumber truncation in the DFS method of Orszag (1974), the old DFS method (Cheong, 2000a, 2000b; Yoshimura and Matsumura, 2005), and 30 the new DFS method. Section 5 examines the accuracy of the old and new DFS methods and the SH method for the Laplacian operator and the Helmholtz equation. Section 6 compares the results of the shallow water test cases between the model using the new DFS method, that using the old DFS method of Yoshimura and Matsumura (2005), and that using the SH method. Section 7 presents conclusions and perspectives.

## 2 Arrangement of equally spaced latitudinal grid points

In DFS models, equally spaced latitudinal grid points are used. We use the following three ways of arranging equally spaced latitudinal grid points in the model using the new DFS method:

$$\text{Grid } [0]: \quad J = J^0, \qquad \theta_j = \pi(j + 0.5)/J^0, \quad j = 0, \dots, J^0 - 1, \tag{7a}$$

$$\text{Grid } [1]: \quad J = J^0 + 1, \quad \theta_j = \pi j/J^0, \qquad\quad j = 0, \dots, J^0, \tag{7b}$$

$$\text{Grid } [-1]: \quad J = J^0 - 1, \quad \theta_j = \pi j/J^0, \qquad\quad j = 1, \dots, J^0 - 1, \tag{7c}$$

where $\theta_j$ is the colatitude at each grid point, and $J^0$ is the number of latitudinal grid points in Grid [0]. When the grid intervals in Grids [0], [1], and [−1] are set equal, the number of grid points $J$ in Grid [1] is $J^0 + 1$, and the number of grid points $J$ in Grid [−1] is $J^0 - 1$. Figure 1 shows Grids [0], [1], and [−1] when $J^0 = 4$ and the grid interval $\Delta\theta = \pi/4$. Grid [0] has been widely used in DFS models (e.g. Merilees, 1973b; Orszag, 1974; Cheong, 2000a, 2000b; Yoshimura and Matsumura, 2005), and in DFS expansion (e.g. Cheong et al. 2004). Grid [1] was used in DFS expansion (e.g. Yee, 1981; Cheong et al., 2004). Grid [−1] was used, for example, in the SH model using Clenshaw–Curtis quadrature (Hotta and Ujiie, 2018). All of Grids [0], [1], and [−1] were used in SH expansion (Swarztrauber and Spotz, 2000).

In the new DFS method, the wind vector components $u$ and $v$ (instead of $u/\sin\theta$ and $v/\sin\theta$ or $u\sin\theta$ and $v\sin\theta$) are transformed from grid space to spectral space and vice versa, as shown in Sects. 3.5 and 3.6 below. This makes it possible to use Grid [1] that has grid points at the poles.

## 3 Improved double Fourier series on the sphere

In Sect. 3, we describe the new basis functions for scalar and vector variables, and the new method to calculate expansion coefficients which minimizes the error due to wavenumber truncation. We compare the new DFS method with the SH method to see the difference between them. We also describe how to calculate the Laplacian operator, the Poisson equation, the Helmholtz equation, and horizontal diffusion in the new DFS method. The essential summary (cook book) of the new DFS method is in Sect. 3.10.

### 3.1 New basis functions for a scalar variable

We propose the following new DFS basis functions that automatically satisfy the continuity conditions at the poles in Eq. (3). The scalar variable $T(\lambda, \theta)$ is expanded zonally as

$$T(\lambda, \theta) \cong \sum_{m=0}^{M} T_m^c(\theta) \cos m\lambda + \sum_{m=1}^{M} T_m^s(\theta) \sin m\lambda, \tag{8}$$

where $T_m^c(\theta)$ and $T_m^s(\theta)$ are calculated from $T(\lambda, \theta)$ by the forward Fourier transform as

$$T_m^c(\theta) = \frac{a}{2\pi} \int_0^{2\pi} \cos m\lambda \, T(\lambda, \theta) d\lambda, \quad a \equiv \begin{cases} 1 \text{ for } m = 0 \\ 2 \text{ for } m \geq 1, \end{cases} \tag{9a}$$

$$T_m^s(\theta) = \frac{1}{\pi} \int_0^{2\pi} \sin m\lambda \, T(\lambda, \theta) d\lambda. \tag{9b}$$

The variables $T_m^c(\theta)$ and $T_m^s(\theta)$ are meridionally expanded as

$$T_m^{c(s)}(\theta) \cong T_m^{c(s),N}(\theta) \equiv \sum_{n=N_{\min,m}}^{N_{\max,m}} T_{n,m}^{c(s)} S_{n,m}(\theta), \tag{10}$$

where

$$S_{n,m}(\theta) \equiv \begin{cases} \cos n\theta & \text{for } m = 0, \\ \sin\theta \cos n\theta & \text{for } m = 1, \\ \sin\theta \sin n\theta & \text{for even } m \geq 2, \\ \sin^2\theta \sin n\theta & \text{for odd } m \geq 3, \end{cases} \tag{11}$$

$$N_{\min,m} = \begin{cases} 0 \text{ for } m = 0, \\ 0 \text{ for } m = 1, \\ 1 \text{ for even } m \geq 2, \\ 1 \text{ for odd } m \geq 3, \end{cases} \quad N_{\max,m} = \begin{cases} N & \text{for } m = 0, \\ N-1 & \text{for } m = 1, \\ N-1 & \text{for even } m \geq 2, \\ N-2 & \text{for odd } m \geq 3. \end{cases} \tag{12}$$

Here, the superscript c(s) means c or s, and for example, $T_m^{c(s)}(\theta)$ means $T_m^c(\theta)$ or $T_m^s(\theta)$. In Eq. (8), $\cos m\lambda$ and $\sin m\lambda$ are used instead of $e^{im\lambda}$ as zonal basis functions for convenience in calculating the expansion coefficients using the least-squares method described below in Sects. 3.3 and 3.6. In Eq. (11), the meridional basis functions $\sin^2\theta \sin n\theta$ for odd $m \geq$ 3 are especially different from the basis functions of Cheong in Eq. (6). Either $\sin n\theta$ or $\sin\theta \cos n\theta$ can be used as the basis functions for $m = 1$ because it can be shown using Eq. (A2a) from Appendix A that $\sin\theta \cos n\theta$ $(n = 0, ..., N-1)$ are the linear combination of $\sin n\theta$ $(n = 1, ..., N)$, and vice versa. Here we use $\sin\theta \cos n\theta$ for $m = 1$ because it can be more easily divided by $\sin\theta$, which is convenient for calculating $\nabla T$.

Using Eq. (A2a–c), Eq. (10) can be converted as

$$T_m^{c(s),N}(\theta) = \begin{cases} \displaystyle\sum_{n=0}^{N} T_{n,m}^{c(s)'} \cos n\theta & \text{for even } m, \\ \displaystyle\sum_{n=1}^{N} T_{n,m}^{c(s)'} \sin n\theta & \text{for odd } m, \end{cases} \tag{13}$$

where

$$T_{n,m}^{c(s)'} = T_{n,m}^{c(s)} \qquad (n = 0, ..., N) \qquad \text{for } m = 0, \tag{14a}$$

$$T_{n,m}^{c(s)'} = \frac{T_{n-1,m}^{c(s)} - T_{n+1,m}^{c(s)}}{2} \qquad (n = 1, ..., N) \qquad \text{for } m = 1 \tag{14b}$$

$$\text{except for } T_{1,m}^{c'(s')} = \frac{2T_{0,m}^{c(s)} - T_{2,m}^{c(s)}}{2} \quad (n = 1),$$

$$T_{n,m}^{c(s)'} = \frac{-T_{n-1,m}^{c(s)} + T_{n+1,m}^{c(s)}}{2} \qquad (n = 0, ..., N) \qquad \text{for even } m \geq 2, \tag{14c}$$

$$T_{n,m}^{c(s)'} = \frac{-T_{n-2,m}^{c(s)} + 2T_{n,m}^{c(s)} - T_{n+2,m}^{c(s)}}{4} \quad (n = 1, \dots, N) \qquad \text{for odd } m \geq 3 \tag{14d}$$

$$\text{except for } T_{1,m}^{c(s)'} = \frac{3T_{1,m}^{c(s)} - T_{3,m}^{c(s)}}{4} \quad (n = 1).$$

The value of $N_{\max,m}$ in Eq. (12) is determined so that the maximum value of $n$ for each $m$ in Eq. (13) becomes $N$. In Grid [0] and Grid [1] (See Sect. 2), the upper limit of $N$ is $J^0 - 1$ for each $m$. In Grid [−1], the upper limit of $N$ is $J^0 - 1$ for $m \geq$ 2, but $J^0 - 2\ (= J - 1)$ for $m = 0$ or 1. This reason is shown in Appendix C.

When calculating the values of $T_m^{c(s),N}(\theta)$ in grid space from $T_{n,m}^{c(s)}$ in spectral space, the coefficients $T_{n,m}^{c(s)'}$ are calculated from $T_{n,m}^{c(s)}$ using Eq. (14) and inverse discrete cosine and sine transforms (See Appendix B) are performed using Eq. (13). The calculation of $T_{n,m}^{c(s)}$ in spectral space from $T_m^{c(s)}(\theta)$ in grid space is described in Sect. 3.3 below.

The truncated variable $T^{N,M}(\lambda, \theta)$ is defined as

$$T^{N,M}(\lambda, \theta) \equiv \sum_{m=0}^{M} T_m^{c,N}(\theta) \cos m\lambda + \sum_{m=1}^{M} T_m^{s,N}(\theta) \sin m\lambda. \tag{15}$$

From Eqs. (10), the values of $T_m^{c,N}(\theta)$ at the poles are finite for $m = 0$, and the values of $T_m^{c(s),N}(\theta)$ at the poles are zero for $m \neq 0$. Therefore $T^{N,M}(\lambda, \theta)$ is continuous at the poles.

### 3.2 Gradient of a scalar variable

The gradient $\nabla T^{N,M} = \left(T_\lambda^{N,M}, T_\phi^{N,M}\right)$ is obtained as follows:

$$T_\lambda^{N,M} \equiv \frac{1}{a \sin \theta} \frac{\partial T^{N,M}}{\partial \lambda} = \sum_{m=1}^{M} T_{\lambda,m}^{c,N}(\theta) \cos m\lambda + \sum_{m=1}^{M} T_{\lambda,m}^{s,N}(\theta) \sin m\lambda, \tag{16a}$$

$$T_{\lambda,m}^{c,N}(\theta) \equiv \frac{m}{a \sin \theta} T_m^{s,N}(\theta) = \sum_{n=N_{\min,m}}^{N_{\max,m}} \left( T_{n,m}^{s} \frac{m S_{n,m}(\theta)}{a \sin \theta} \right), \tag{16b}$$

$$T_{\lambda,m}^{s,N}(\theta) \equiv -\frac{m}{a \sin \theta} T_m^{c,N}(\theta) = \sum_{n=N_{\min,m}}^{N_{\max,m}} \left( -T_{n,m}^{c} \frac{m S_{n,m}(\theta)}{a \sin \theta} \right), \tag{16c}$$

$$T_\phi^{N,M} \equiv \frac{1}{a} \frac{\partial T^{N,M}}{\partial \phi} = -\frac{1}{a} \frac{\partial T^{N,M}}{\partial \theta} = \sum_{m=0}^{M} T_{\phi,m}^{c,N}(\theta) \cos m\lambda + \sum_{m=1}^{M} T_{\phi,m}^{s,N}(\theta) \sin m\lambda, \tag{17a}$$

$$T_{\phi,m}^{c(s),N}(\theta) \equiv -\frac{1}{a} \frac{\partial T_m^{c(s),N}(\theta)}{\partial \theta} = \sum_{n=N_{\min,m}}^{N_{\max,m}} \left( -T_{n,m}^{c(s)} \frac{1}{a} \frac{\partial S_{n,m}(\theta)}{\partial \theta} \right), \tag{17b}$$

where $a$ is the radius of the earth, and $\phi$ is the latitude. From Eqs. (16b,c) and (A2b), we obtain

$$
T_{\lambda,m}^{c(s),N}(\theta) = \begin{cases}
0 & \text{for } m = 0, \\
\displaystyle\sum_{n=0}^{N-1} T_{\lambda,n,m}^{c(s)} \cos n\theta & \text{for } m = 1, \\
\displaystyle\sum_{n=1}^{N-1} T_{\lambda,n,m}^{c(s)} \sin n\theta & \text{for even } m \geq 2, \\
\displaystyle\sum_{n=0}^{N-1} T_{\lambda,n,m}^{c(s)} \cos n\theta \left( = \sum_{n=1}^{N-2} T_{\lambda,n,m}^{c(s)'} \sin\theta \sin n\theta \right) & \text{for odd } m \geq 3,
\end{cases}
\tag{18}
$$

where

$$
T_{\lambda,n,m}^{c} = \frac{1}{a} m T_{n,m}^{s} \qquad (n = 0, \ldots, N-1) \qquad \text{for } m = 1, \tag{19a}
$$

$$
T_{\lambda,n,m}^{c} = \frac{1}{a} m T_{n,m}^{s} \qquad (n = 1, \ldots, N-1) \qquad \text{for even } m \geq 2, \tag{19b}
$$

$$
T_{\lambda,n,m}^{c} = \frac{1}{a} \frac{m(-T_{n-1,m}^{s} + T_{n+1,m}^{s})}{2} \qquad (n = 0, \ldots, N-1) \qquad \text{for odd } m \geq 3. \tag{19c}
$$

The equations for $T_{\lambda,n,m}^{s}$ are the same as Eq. (19), except that $T_{\lambda,n,m}^{c}$ and $T_{n,m}^{s}$ are replaced with $T_{\lambda,n,m}^{s}$ and $-T_{n,m}^{c}$, respectively. From Eqs. (17b), (13) and (14), we obtain

$$
T_{\phi,m}^{c(s),N}(\theta) = \begin{cases}
\displaystyle\sum_{n=1}^{N} T_{\phi,n,m}^{c(s)} \sin n\theta & \text{for } m = 0, \\
\displaystyle\sum_{n=0}^{N} T_{\phi,n,m}^{c(s)} \cos n\theta & \text{for } m = 1, \\
\displaystyle\sum_{n=1}^{N} T_{\phi,n,m}^{c(s)} \sin n\theta & \text{for even } m \geq 2, \\
\displaystyle\sum_{n=0}^{N} T_{\phi,n,m}^{c(s)} \cos n\theta \left( = \sum_{n=1}^{N-1} T_{\phi,n,m}^{c(s)'} \sin\theta \sin n\theta \right) & \text{for odd } m \geq 3,
\end{cases}
\tag{20}
$$

where

$$
T_{\phi,n,m}^{c(s)} = -\frac{1}{a} \left( -n T_{n,m}^{c(s)} \right) \qquad (n = 1, \ldots, N) \qquad \text{for } m = 0, \tag{21a}
$$

$$
T_{\phi,n,m}^{c(s)} = -\frac{1}{a} \left[ \frac{n(T_{n-1,m}^{c(s)} - T_{n+1,m}^{c(s)})}{2} \right] \qquad (n = 0, \ldots, N) \qquad \text{for } m = 1, \tag{21b}
$$

$$
\text{except for } T_{\phi,1,m}^{c(s)} = -\frac{1}{a} \left[ \frac{2 T_{0,m}^{c(s)} - T_{2,m}^{c(s)}}{2} \right] \qquad (n = 1),
$$

$$
T_{\phi,n,m}^{c(s)} = -\frac{1}{a} \left[ \frac{n(T_{n-1,m}^{c(s)} - T_{n+1,m}^{c(s)})}{2} \right] \qquad (n = 1, \ldots, N) \qquad \text{for even } m \geq 2, \tag{21c}
$$

$$T_{\phi,n,m}^{c(s)} = -\frac{1}{a}\left[\frac{n\left(-T_{n-2,m}^{c(s)} + 2T_{n,m}^{c(s)} - T_{n+2,m}^{c(s)}\right)}{4}\right] \quad (n = 0, \dots, N) \quad \text{for odd } m \geq 3, \quad (21d)$$

$$\text{except for } T_{\phi,1,m}^{c(s)} = -\frac{1}{a}\left[\frac{\left(3T_{1,m}^{c(s)} - T_{3,m}^{c(s)}\right)}{4}\right] \quad (n = 1).$$

From Eqs. (18)–(21), it can be seen that $T_{\lambda,m}^{c,N}(\theta)$, $T_{\lambda,m}^{s,N}(\theta)$, $T_{\phi,m}^{c,N}(\theta)$, and $T_{\phi,m}^{s,N}(\theta)$ at the poles are finite for $m = 1$ and zero for $m \neq 1$, and moreover the following relations are satisfied for $m = 1$:

$$T_{\lambda,m=1}^{c,N}(\theta) = -T_{\phi,m=1}^{s,N}(\theta)\left(= \frac{1}{a}\sum_{n=0}^{N-1} T_{n,m=1}^{s}\right) \quad \text{at } \theta = 0 \text{ (North Pole)}, \quad (22a)$$

$$T_{\lambda,m=1}^{s,N}(\theta) = T_{\phi,m=1}^{c,N}(\theta)\left(= -\frac{1}{a}\sum_{n=0}^{N-1} T_{n,m=1}^{c}\right) \quad \text{at } \theta = 0 \text{ (North Pole)}, \quad (22b)$$

$$T_{\lambda,m=1}^{c,N}(\theta) = T_{\phi,m=1}^{s,N}(\theta)\left(= \frac{1}{a}\sum_{n=0}^{N-1} (-1)^n T_{n,m=1}^{s}\right) \quad \text{at } \theta = \pi \text{ (South Pole)}, \quad (22c)$$

$$T_{\lambda,m=1}^{s,N}(\theta) = -T_{\phi,m=1}^{c,N}(\theta)\left(= -\frac{1}{a}\sum_{n=0}^{N-1} (-1)^n T_{n,m=1}^{c}\right) \quad \text{at } \theta = \pi \text{ (South Pole)}. \quad (22d)$$

Thus, it is guaranteed that $\nabla T^{N,M} = \left(T_\lambda^{N,M}, T_\phi^{N,M}\right)$ is continuous at the poles.

## 3.3 New method to calculate expansion coefficients for a scalar variable

One way to calculate the coefficients $T_{n,m}^{c(s)}$ from $T_m^{c(s)}(\theta)$ in Eq. (10) is to perform a forward cosine transform of $T_m^{c(s)}(\theta)/\sin\theta$ for $m = 1$, a sine transform of $T_m^{c(s)}(\theta)/\sin\theta$ for even $m$ ($\geq 2$), and a sine transform of $T_m^{c(s)}(\theta)/\sin^2\theta$ for odd $m$ ($\geq 3$). However, this approach with the meridional wavenumber truncation $N < J$ leads to large high-wavenumber oscillations (See Sect. 4). Dividing $T_m^{c(s)}(\theta)$ by $\sin^2\theta$ reduces the numerical stability of the model more significantly than dividing $T_m^{c(s)}(\theta)$ by $\sin\theta$.

Here we propose a new method to calculate expansion coefficients using the least-squares method to minimize the error due to the meridional wavenumber truncation. This method avoids dividing $T_m^{c(s)}(\theta)$ by $\sin\theta$ or $\sin^2\theta$ before the forward cosine or sine transforms. The coefficients $T_{n,m}^{c(s)}$ in Eq. (10) are calculated as follows. First, $T_m^{c(s)}(\theta)$ in Eq. (10) are expanded like Eq. (5) as

$$T_m^{c(s)}(\theta) \cong \tilde{T}_m^{c(s),N}(\theta) \equiv \begin{cases} \displaystyle\sum_{n=0}^{N} \tilde{T}_{n,m}^{c(s)} \cos n\theta & \text{for even } m, \\[2em] \displaystyle\sum_{n=1}^{N} \tilde{T}_{n,m}^{c(s)} \sin n\theta & \text{for odd } m, \end{cases} \quad (23)$$

where the expansion coefficients $\tilde{T}_{n,m}^{c(s)}$ are calculated by the forward discrete cosine transform for even $m$ and the forward discrete sine transform for odd $m$ from the values of $T_m^{c(s)}(\theta)$ at the grid points (See Appendix B).

Next, $T_{n,m}^c$ and $T_{n,m}^s$ are calculated using the least-squares method to minimize the following error $E$ (the squared $L_2$ norm of the residual):

$$E \equiv \frac{1}{2\pi^2} \int_0^{2\pi} \int_0^{\pi} R(\lambda,\theta)^2 d\theta d\lambda, \tag{24}$$

where the residual $R(\lambda,\theta)$ is

$$R(\lambda,\theta) \equiv T^{N,M}(\lambda,\theta) - T(\lambda,\theta). \tag{25}$$

From Eqs. (24), (25) and (15), and the equations $\partial E/\partial T_{n,m}^c = 0$ and $\partial E/\partial T_{n,m}^s = 0$ used in the least-squares method to minimize $E$, we obtain

$$\frac{1}{2\pi^2} \int_0^{2\pi} \int_0^{\pi} \frac{\partial T_m^{c,N}(\theta)}{\partial T_{n,m}^c} \cos m\lambda \, R(\lambda,\theta) d\theta d\lambda = 0, \tag{26a}$$

$$\frac{1}{2\pi^2} \int_0^{2\pi} \int_0^{\pi} \frac{\partial T_m^{s,N}(\theta)}{\partial T_{n,m}^s} \sin m\lambda \, R(\lambda,\theta) d\theta d\lambda = 0. \tag{26b}$$

From Eq. (10), we derive

$$\frac{\partial T_m^{c,N}(\theta)}{\partial T_{n,m}^c} = \frac{\partial T_m^{s,N}(\theta)}{\partial T_{n,m}^s} = S_{n,m}(\theta). \tag{27}$$

Equations (26) and (27) show that the residual $R(\lambda,\theta)$ is orthogonal to each of the new DFS basis functions $S_{m,n}(\theta)\cos m\lambda$ and $S_{m,n}(\theta)\sin m\lambda$, which means that Eq. (26) is the same as the equation derived using the Galerkin method.

From Eqs. (26), (27), (25), (15), (9) and (A3), we derive

$$\int_0^{\pi} S_{n,m}(\theta) T_m^{c(s),N}(\theta) d\theta = \int_0^{\pi} S_{n,m}(\theta) T_m^{c(s)}(\theta) d\theta. \tag{28}$$

From Eqs. (28) and (D4) in Appendix D, we obtain

$$\int_0^{\pi} S_{n,m}(\theta) T_m^{c(s),N}(\theta) d\theta = \int_0^{\pi} S_{n,m}(\theta) \tilde{T}_m^{c(s),N}(\theta) d\theta. \tag{29}$$

By substituting Eqs. (10) and (23) into Eq. (29), the following equations for $T_{n,m}^c$ and $T_{n,m}^s$ are derived, as shown in Appendix E.

For $m = 0$,

$$T_{n,m}^{c(s)} = \tilde{T}_{n,m}^{c(s)} \quad (0 \le n \le N). \tag{30a}$$

For $m = 1$,

$$-T_{n-2,m}^{c(s)} + 2T_{n,m}^{c(s)} - T_{n+2,m}^{c(s)} = -2\tilde{T}_{n-1,m}^{c(s)} + 2\tilde{T}_{n+1,m}^{c(s)} \quad (0 \le n \le N-1), \tag{30b}$$

with the exception of the following underlined values:

$$\underline{1}T_{1,m}^{c(s)} - T_{3,m}^{c(s)} = 2\tilde{T}_{2,m}^{c(s)} \qquad (n = 1),$$

$$-2\underline{T_{0,m}^{c(s)}} + 2T_{2,m}^{c(s)} - T_{4,m}^{c(s)} = -2\tilde{T}_{1,m}^{c(s)} + 2\tilde{T}_{3,m}^{c(s)} \qquad (n = 2).$$

For even $m$ ($\geq 2$),

$$-T_{n-2,m}^{c(s)} + 2T_{n,m}^{c(s)} - T_{n+2,m}^{c(s)} = 2\tilde{T}_{n-1,m}^{c(s)} - 2\tilde{T}_{n+1,m}^{c(s)} \qquad (1 \leq n \leq N-1), \qquad (30c)$$

with the exception of the following underlined values:

$$\underline{3T_{1,m}^{c(s)}} - T_{3,m}^{c(s)} = \underline{4\tilde{T}_{0,m}^{c(s)}} - 2\tilde{T}_{2,m}^{c(s)} \qquad (n = 1).$$

For odd $m$ ($\geq 3$),

$$T_{n-4,m}^{c(s)} - 4T_{n-2,m}^{c(s)} + 6T_{n,m}^{c(s)} - 4T_{n+2,m}^{c(s)} + T_{n+4,m}^{c(s)} = -4\tilde{T}_{n-2,m}^{c(s)} + 8\tilde{T}_{n,m}^{c(s)} - 4\tilde{T}_{n+2,m}^{c(s)} \quad (1 \leq n \leq N-2), \quad (30d)$$

with the exception of the following underlined values:

$$\underline{10T_{1,m}^{c(s)}} - \underline{5T_{3,m}^{c(s)}} + T_{5,m}^{c(s)} = \underline{12\tilde{T}_{1,m}^{c(s)}} - 4\tilde{T}_{3,m}^{c(s)} \qquad (n = 1),$$

$$\underline{5T_{2,m}^{c(s)}} - 4T_{4,m}^{c(s)} + T_{6,m}^{c(s)} = 8\tilde{T}_{2,m}^{c(s)} - 4\tilde{T}_{4,m}^{c(s)} \qquad (n = 2),$$

$$-\underline{5T_{1,m}^{c(s)}} + 6T_{3,m}^{c(s)} - 4T_{5,m}^{c(s)} + T_{7,m}^{c(s)} = -4\tilde{T}_{1,m}^{c(s)} + 8\tilde{T}_{3,m}^{c(s)} - 4\tilde{T}_{5,m}^{c(s)} \qquad (n = 3).$$

From Eq. (30a), $T_{n,m}^{c(s)}$ for $m = 0$ is obtained. From Eqs. (30d), the following linear simultaneous equations for $m \geq 3$ are derived:

$$\mathbf{C}_{\text{n\_odd},m} \begin{bmatrix} T_{1,m}^{c(s)} \\ T_{3,m}^{c(s)} \\ T_{5,m}^{c(s)} \\ T_{7,m}^{c(s)} \\ \vdots \end{bmatrix} = \mathbf{D}_{\text{n\_odd},m} \begin{bmatrix} \tilde{T}_{1,m}^{c(s)} \\ \tilde{T}_{3,m}^{c(s)} \\ \tilde{T}_{5,m}^{c(s)} \\ \tilde{T}_{7,m}^{c(s)} \\ \vdots \end{bmatrix}, \quad \mathbf{C}_{\text{n\_even},m} \begin{bmatrix} T_{2,m}^{c(s)} \\ T_{4,m}^{c(s)} \\ T_{6,m}^{c(s)} \\ T_{8,m}^{c(s)} \\ \vdots \end{bmatrix} = \mathbf{D}_{\text{n\_even},m} \begin{bmatrix} \tilde{T}_{2,m}^{c(s)} \\ \tilde{T}_{4,m}^{c(s)} \\ \tilde{T}_{6,m}^{c(s)} \\ \tilde{T}_{8,m}^{c(s)} \\ \vdots \end{bmatrix}, \qquad (31)$$

where the matrices $\mathbf{C}_{\text{n\_odd},m}$ and $\mathbf{C}_{\text{n\_even},m}$ are penta-diagonal. From Eqs. (30b,c), the equations similar to Eq. (31) for $m = 1$ and even $m$ ($\geq 2$) with tri-diagnoal matrices $\mathbf{C}_{\text{n\_odd},m}$ and $\mathbf{C}_{\text{n\_even},m}$ are derived. By using Eq. (31), the expansion coefficients $T_{n,m}^{c(s)}$ are calculated from $\tilde{T}_{n,m}^{c(s)}$. A penta-diagonal matrix $\mathbf{C}$ can be LU decomposed as

$$\mathbf{C} = \mathbf{LU}, \quad \mathbf{L} \equiv \begin{bmatrix} * & 0 & 0 & 0 & 0 & 0 & \cdots & 0 \\ * & * & 0 & 0 & 0 & 0 & \cdots & 0 \\ * & * & * & 0 & 0 & 0 & \cdots & 0 \\ 0 & * & * & * & 0 & 0 & \cdots & 0 \\ & & & \vdots & & & & \\ 0 & 0 & \cdots & 0 & * & * & * & 0 \\ 0 & 0 & \cdots & 0 & 0 & * & * & * \end{bmatrix}, \quad \mathbf{U} \equiv \begin{bmatrix} 1 & * & * & 0 & 0 & 0 & \cdots & 0 \\ 0 & 1 & * & * & 0 & 0 & \cdots & 0 \\ 0 & 0 & 1 & * & * & 0 & \cdots & 0 \\ 0 & 0 & 0 & 1 & * & * & \cdots & 0 \\ & & & \vdots & & & & \\ 0 & 0 & \cdots & 0 & 0 & 0 & 1 & * \\ 0 & 0 & \cdots & 0 & 0 & 0 & 0 & 1 \end{bmatrix}. \qquad (32)$$

To solve $\mathbf{LU}x = b$, we solve $\mathbf{L}y = b$ with forward substitution first, and then solve $\mathbf{U}x = y$ with backward substitution. There are also other methods to solve Eq. (31). For example, the method using LU decomposition considering penta-diagonal matrices as $2 \times 2$ block tri-diagonal matrices makes SIMD operations more effective. The method using cyclic reduction for block tri-diagonal matrices (e.g., Gander and Golub, 1997) is suitable for vectorization and parallelization. The calculation with these methods for each $m$ requires $O(N)$ operations. The simultaneous equations with tri-diagonal matrices

**C** can be solved in a similar way. Therefore, the calculation of $T_{n,m}^{c(s)}$ for all $m$ and $n$ with Eq. (30) requires only $O(N^2)$ operations.

### 3.4 Comparison of new DFS with SH

Here we compare the new DFS method with the SH method to see the difference between them. In the SH method, $T_m^c(\theta)$
and $T_m^s(\theta)$ in Eq. (8) are expanded with the associated Legendre functions $P_{n,m}(\theta)$ as

$$T_m^{c(s)}(\theta) \cong T_m^{c(s),SH,N}(\theta) \equiv \sum_{n=m}^{N} T_{n,m}^{c(s),SH} P_{n,m}(\theta), \tag{33}$$

where $m \geq 0$. The functions $P_{n,m}(\theta)$ satisfy the following orthogonality relations for each $m$:

$$\int_0^\pi P_{n,m}(\theta) P_{n',m}(\theta) \sin\theta \, d\theta = \begin{cases} 1 \text{ (or 2)} & \text{for } n = n', \\ 0 & \text{for } n \neq n'. \end{cases} \tag{34}$$

By the modified Robert expansion (Merilees, 1973a; Orszag, 1974), the associated Legendre functions $P_{n,m}(\theta)$ are expressed as

$$P_{n,m}(\theta) = \sum_{\substack{l=0 \\ \text{when } n-|m|-l \text{ is even}}}^{n-|m|} a_{n,m,l} \sin^{|m|}\theta \cos l\theta. \tag{35}$$

Conversely, the functions $\sin^{|m|}\theta \cos(n-|m|)\theta$ $(n \geq |m|)$ can be expressed as the linear combination of $P_{l,m}(\theta)$ $(l = |m|, \dots, n)$. Substituting Eq. (35) into Eq. (33) gives

$$T_m^{c(s),SH,N}(\theta) = \sum_{n=0}^{N-m} T_{n,m}^{c(s),SH'} \sin^m\theta \cos n\theta, \tag{36}$$

where $m \geq 0$. Equation (36) is similar to Eq. (10) in the following sense: the basis functions for $m = 0$ and $m = 1$ in Eq. (36) are the same as Eq. (11). The basis functions $\sin^2\theta \cos n\theta$ $(n = 0, \dots, N-2)$ for $m = 2$ and $\sin^3\theta \cos n\theta$ $(n = 0, \dots, N-3)$ for $m = 3$ in Eq. (36) are the linear combinations of $\sin\theta \sin n\theta$ $(n = 1, \dots, N-1)$ and $\sin^2\theta \sin n\theta$ $(n = 1, \dots, N-2)$ in Eq. (11), respectively (see Eq. (A2a)), and vice versa. The basis functions for $m \geq 4$ in Eq. (36) are different from Eq. (11). The number of expansion coefficients in Eq. (33) or Eq. (36) in the SH method is smaller than in Eq. (10) in the new DFS method for each $m \geq 4$. From Eqs. (8) and (33), the number of expansion coefficients $T_{n,m}^{c,SH}$ in the SH model is about $N^2/2$ when $M = N$. This triangular truncation used in the SH method gives a uniform resolution over the sphere. From Eqs. (8) and (10), the number of the expansion coefficients $T_{n,m}^c$ in the DFS method is about $N^2$ when $M = N$. This rectangular truncation used in the DFS model gives almost the same resolution as the grid spacing of the regular longitude–latitude grids. Therefore, the zonal Fourier filter (see Appendix F) is used in the DFS model to give a more uniform resolution.

We compare the method used to calculate the expansion coefficients in the new DFS method with that in the SH method. The SH expansion coefficients $T_{n,m}^{\mathrm{c(s)},\mathrm{SH}}$ in Eq. (33) are calculated from $T_m^{\mathrm{c(s)}}(\theta)$ by the forward Legendre transform as

$$T_{n,m}^{\mathrm{c(s)},SH} = \int_0^{\pi} P_{n,m}(\theta) T_m^{\mathrm{c(s)}}(\theta) \sin \theta \, d\theta, \tag{37}$$

where Gaussian quadrature or Clenshaw–Curtis quadrature (e.g., Hotta and Ujiie, 2018) is usually used for integration. They can also be calculated using the same equations as Eq. (37) except that $T_m^{\mathrm{c(s)}}(\theta)$ are replaced with $\tilde{T}_m^{\mathrm{c(s)},N}(\theta)$ (e.g., Sneeuw and Bun, 1996), although the values of $T_{n,m}^{\mathrm{c(s)},\mathrm{SH}}$ calculated from $\tilde{T}_m^{\mathrm{c(s)},N}(\theta)$ are different from those calculated from $T_m^{\mathrm{c(s)}}(\theta)$. In the new DFS method, the values of $T_{n,m}^{\mathrm{c(s)}}$ calculated from $\tilde{T}_m^{\mathrm{c(s)},N}(\theta)$ in Eq. (29) are the same as those calculated from $\tilde{T}_m^{\mathrm{c(s)}}(\theta)$ in Eq. (28) (See Eq. (D4) in Appendix D).

Equation (37) can be derived using the least-squares method that minimizes the error $E^{\mathrm{SH}}$ (the squared $L_2$ norm of the residual):

$$E^{\mathrm{SH}} \equiv \frac{1}{4\pi} \int_0^{2\pi} \int_0^{\pi} R^{\mathrm{SH}}(\lambda, \theta)^2 \sin \theta \, d\theta d\lambda, \tag{38}$$

where the residual $R^{\mathrm{SH}}(\lambda, \theta)$ is

$$R^{\mathrm{SH}}(\lambda, \theta) \equiv \left( \sum_{m=0}^{M} T_m^{\mathrm{c},\mathrm{SH},N}(\theta) \cos m\lambda + \sum_{m=1}^{M} T_m^{\mathrm{s},\mathrm{SH},N}(\theta) \sin m\lambda \right) - T(\lambda, \theta). \tag{39}$$

From Eqs. (38), (39) and (33), and the equations $\partial E^{\mathrm{SH}} / \partial T_{n,m}^{\mathrm{c},\mathrm{SH}} = 0$ and $\partial E^{\mathrm{SH}} / \partial T_{n,m}^{\mathrm{s},\mathrm{SH}} = 0$ used in the least-squares method to minimize $E^{\mathrm{SH}}$, we derive

$$\int_0^{2\pi} \int_0^{\pi} P_{n,m}(\theta) \cos m\lambda \, R^{\mathrm{SH}}(\lambda, \theta) \sin \theta \, d\theta d\lambda = 0, \tag{40a}$$

$$\int_0^{2\pi} \int_0^{\pi} P_{n,m}(\theta) \sin m\lambda \, R^{\mathrm{SH}}(\lambda, \theta) \sin \theta \, d\theta d\lambda = 0. \tag{40b}$$

Equation (40) is the same as the equation obtained using the Galerkin method. From Eqs. (40), (33), (34), (9) and (A3), we derive Eq. (37).

In Eqs. (37) and (38), the latitudinal weight $\sin \theta$ appears, unlike in Eqs. (24) and (28), which is another difference between the SH and the new DFS methods. In the DFS method, the constant latitudinal weight is used in Eq. (24), although the latitudinal area weight described below in Appendix G is usually used as the latitudinal weight at the grid points, for example, for the calculation of the global mean.

When calculating the coefficients $T_{n,m}^{\mathrm{c(s)}}$ in Eq. (10), we can also consider the least-squares method, not using $E$ in Eq. (24) but using $E'$ with latitudinal weight $\sin \theta$ like Eq. (38). However, minimizing $E'$ derives the simultaneous equations for calculating $T_{n,m}^{\mathrm{c(s)}}$ with dense matrices, which leads to $O(N^3)$ operations. When using $E$, the simultaneous equations with penta-diagonal or tri-diagonal matrices require only $O(N^2)$ operations. Therefore, we choose to use $E$ instead of $E'$.

The new DFS meridional basis functions $S_{n,m}(\theta)$ for each $m$ are not orthogonal but independent. Therefore, by using Gram-Schmidt orthogonalization, the basis functions can be converted to orthogonalized basis functions $S_{n,m}^{\mathrm{O}}(\theta)$, which satisfy

$$\frac{1}{\pi}\int_0^{\pi} S_{n,m}^{\mathrm{O}}(\theta)S_{n',m}^{\mathrm{O}}(\theta)\,d\theta = \begin{cases} 1 & \text{for } n = n', \\ 0 & \text{for } n \neq n'. \end{cases} \tag{41}$$

5   This is similar to Eq. (34), but the latitudinal weight is constant. $T_m^{\mathrm{c(s)}}(\theta)$ in Eq. (8) are expanded with $S_{n,m}^{\mathrm{O}}(\theta)$ as

$$T_m^{\mathrm{c(s)}}(\theta) \cong T_m^{\mathrm{c(s),O},N}(\theta) \equiv \sum_{n=N_{\min,m}}^{N_{\max,m}} T_{n,m}^{\mathrm{c(s),O}} S_{n,m}^{\mathrm{O}}(\theta). \tag{42}$$

By using the least squares method or the Galerkin method with Eq. (42), we obtain the same equations as Eqs. (24)–(29) except that $T_m^{\mathrm{c(s)},N}(\theta)$ and $S_{n,m}(\theta)$ are replaced with $T_m^{\mathrm{c(s),O},N}(\theta)$ and $S_{n,m}^{\mathrm{O}}(\theta)$ respectively. From Eq. (29) with $T_m^{\mathrm{c(s)},N}(\theta)$ and $S_{n,m}(\theta)$ replaced by $T_m^{\mathrm{c(s),O},N}(\theta)$ and $S_{n,m}^{\mathrm{O}}(\theta)$, and Eqs. (41) and (42), we derive

$$T_{n,m}^{\mathrm{c(s),O}} = \frac{1}{\pi}\int_0^{\pi} S_{n,m}^{\mathrm{O}}(\theta)\tilde{T}_m^{\mathrm{c(s)},N}(\theta)\,d\theta. \tag{43}$$

Thus, $T_{n,m}^{\mathrm{c(s),O}}$ and $T_m^{\mathrm{c(s),O},N}(\theta)$ in Eqs. (43) and (42) are calculated uniquely. This unique solution $T_m^{\mathrm{c(s),O},N}(\theta)$ is the same as $T_m^{\mathrm{c(s)},N}(\theta)$ in Eq. (29) obtained by the least-squares method with the non-orthogonal basis functions $S_{n,m}(\theta)$, because $S_{n,m}^{\mathrm{O}}(\theta)$ $\left(n = N_{\min,m}, \dots, N_{\max,m}\right)$ are the linear combination of $S_{n,m}(\theta)$ $\left(n = N_{\min,m}, \dots, N_{\max,m}\right)$ for each $m$, and vice versa.

**3.5 New basis functions for a wind vector**

15   The velocity potential $\chi$ and the stream function $\psi$ can be converted into the wind vector components $u$ and $v$ using the equations

$$u = \frac{1}{a\cos\phi}\frac{\partial\chi}{\partial\lambda} - \frac{1}{a}\frac{\partial\psi}{\partial\phi} = \frac{1}{a\sin\theta}\frac{\partial\chi}{\partial\lambda} + \frac{1}{a}\frac{\partial\psi}{\partial\theta}, \tag{44a}$$

$$v = \frac{1}{a\cos\phi}\frac{\partial\psi}{\partial\lambda} + \frac{1}{a}\frac{\partial\chi}{\partial\phi} = \frac{1}{a\sin\theta}\frac{\partial\psi}{\partial\lambda} - \frac{1}{a}\frac{\partial\chi}{\partial\theta}, \tag{44b}$$

where $u = a\cos\phi\, d\lambda/dt$ is the zonal wind, and $v = a\,d\phi/dt$ is the meridional wind. The scalar variables $\chi$ and $\psi$ are
20   expanded like Eqs. (8) and (10) as

$$\begin{bmatrix} \chi(\lambda,\theta) \\ \psi(\lambda,\theta) \end{bmatrix} \cong \sum_{m=0}^{M} \begin{bmatrix} \chi_m^{\mathrm{c}}(\theta) \\ \psi_m^{\mathrm{c}}(\theta) \end{bmatrix} \cos m\lambda + \sum_{m=1}^{M} \begin{bmatrix} \chi_m^{\mathrm{s}}(\theta) \\ \psi_m^{\mathrm{s}}(\theta) \end{bmatrix} \sin m\lambda, \tag{45}$$

$$\begin{bmatrix} \chi_m^{\mathrm{c(s)}}(\theta) \\ \psi_m^{\mathrm{c(s)}}(\theta) \end{bmatrix} \cong \begin{bmatrix} \chi_m^{\mathrm{c(s)},N}(\theta) \\ \psi_m^{\mathrm{c(s)},N}(\theta) \end{bmatrix} \equiv \sum_{n=N_{\min,m}}^{N_{\max,m}} \begin{bmatrix} \chi_{n,m}^{\mathrm{c(s)}} \\ \psi_{n,m}^{\mathrm{c(s)}} \end{bmatrix} S_{n,m}(\theta), \tag{46}$$

The truncated variables $\psi^{N,M}(\lambda,\theta)$ and $\chi^{N,M}(\lambda,\theta)$ are defined as

$$\begin{bmatrix} \chi^{N,M}(\lambda,\theta) \\ \psi^{N,M}(\lambda,\theta) \end{bmatrix} \equiv \sum_{m=0}^{M} \begin{bmatrix} \chi_m^{c,N}(\theta) \\ \psi_m^{c,N}(\theta) \end{bmatrix} \cos m\lambda + \sum_{m=1}^{M} \begin{bmatrix} \chi_m^{s,N}(\theta) \\ \psi_m^{s,N}(\theta) \end{bmatrix} \sin m\lambda, \tag{47}$$

From Eqs. (44)–(47), the equations for the wind vector components $u^{N,M}(\lambda,\theta)$ and $v^{N,M}(\lambda,\theta)$ are derived as

$$u^{N,M}(\lambda,\theta) \equiv \frac{1}{a\sin\theta}\frac{\partial \chi^{N,M}(\lambda,\theta)}{\partial \lambda} + \frac{1}{a}\frac{\partial \psi^{N,M}(\lambda,\theta)}{\partial \theta} = \sum_{m=0}^{M} u_m^{c,N}(\theta)\cos m\lambda + \sum_{m=1}^{M} u_m^{s,N}(\theta)\sin m\lambda, \tag{48a}$$

$$u_m^{c,N}(\theta) \equiv \frac{m\chi_m^{s,N}(\theta)}{a\sin\theta} + \frac{1}{a}\frac{\partial \psi_m^{c,N}(\theta)}{\partial \theta} = \sum_{n=N_{\min,m}}^{N_{\max,m}} \left( \chi_{n,m}^{s}\frac{mS_{n,m}(\theta)}{a\sin\theta} + \psi_{n,m}^{c}\frac{1}{a}\frac{\partial S_{n,m}(\theta)}{\partial \theta} \right), \tag{48b}$$

$$u_m^{s,N}(\theta) \equiv -\frac{m\chi_m^{c,N}(\theta)}{a\sin\theta} + \frac{1}{a}\frac{\partial \psi_m^{s,N}(\theta)}{\partial \theta} = \sum_{n=N_{\min,m}}^{N_{\max,m}} \left( -\chi_{n,m}^{c}\frac{mS_{n,m}(\theta)}{a\sin\theta} + \psi_{n,m}^{s}\frac{1}{a}\frac{\partial S_{n,m}(\theta)}{\partial \theta} \right), \tag{48c}$$

$$v^{N,M}(\lambda,\theta) \equiv \frac{1}{a\sin\theta}\frac{\partial \psi^{N,M}(\lambda,\theta)}{\partial \lambda} - \frac{1}{a}\frac{\partial \chi^{N,M}(\lambda,\theta)}{\partial \theta} = \sum_{m=0}^{M} v_m^{c,N}(\theta)\cos m\lambda + \sum_{m=1}^{M} v_m^{s,N}(\theta)\sin m\lambda, \tag{49a}$$

$$v_m^{c,N}(\theta) \equiv \frac{m\psi_m^{s,N}(\theta)}{a\sin\theta} - \frac{1}{a}\frac{\partial \chi_m^{c,N}(\theta)}{\partial \theta} = \sum_{n=N_{\min,m}}^{N_{\max,m}} \left( \psi_{n,m}^{s}\frac{mS_{n,m}(\theta)}{a\sin\theta} - \chi_{n,m}^{c}\frac{1}{a}\frac{\partial S_{n,m}(\theta)}{\partial \theta} \right), \tag{49b}$$

$$v_m^{s,N}(\theta) \equiv -\frac{m\psi_m^{c,N}(\theta)}{a\sin\theta} - \frac{1}{a}\frac{\partial \chi_m^{s,N}(\theta)}{\partial \theta} = \sum_{n=N_{\min,m}}^{N_{\max,m}} \left( -\psi_{n,m}^{c}\frac{mS_{n,m}(\theta)}{a\sin\theta} - \chi_{n,m}^{s}\frac{1}{a}\frac{\partial S_{n,m}(\theta)}{\partial \theta} \right). \tag{49c}$$

The vector $(u^{N,M}, v^{N,M})$ in Eqs. (48) and (49) can also be represented as

$$\left( u^{N,M}(\lambda,\theta), v^{N,M}(\lambda,\theta) \right) = \sum_{m=0}^{M} \sum_{n=N_{\min,m}}^{N_{\max,m}} \left( \chi_{n,m}^{c}\mathbf{V}_{n,m}^{1} + \chi_{n,m}^{s}\mathbf{V}_{n,m}^{2} + \psi_{n,m}^{c}\mathbf{V}_{n,m}^{3} + \psi_{n,m}^{s}\mathbf{V}_{n,m}^{4} \right), \tag{50}$$

where we define the new DFS vector basis functions $\mathbf{V}_{n,m}^{1}$, $\mathbf{V}_{n,m}^{2}$, $\mathbf{V}_{n,m}^{3}$ and $\mathbf{V}_{n,m}^{4}$ as

$$\mathbf{V}_{n,m}^{1}(\lambda,\theta) \equiv \left( -\frac{mS_{n,m}(\theta)}{a\sin\theta}\sin m\lambda, -\frac{1}{a}\frac{\partial S_{n,m}(\theta)}{\partial \theta}\cos m\lambda \right), \tag{51a}$$

$$\mathbf{V}_{n,m}^{2}(\lambda,\theta) \equiv \left( \frac{mS_{n,m}(\theta)}{a\sin\theta}\cos m\lambda, -\frac{1}{a}\frac{\partial S_{n,m}(\theta)}{\partial \theta}\sin m\lambda \right), \tag{51b}$$

$$\mathbf{V}_{n,m}^{3}(\lambda,\theta) \equiv \left( \frac{1}{a}\frac{\partial S_{n,m}(\theta)}{\partial \theta}\cos m\lambda, -\frac{mS_{n,m}(\theta)}{a\sin\theta}\sin m\lambda \right), \tag{51c}$$

$$\mathbf{V}_{n,m}^{4}(\lambda,\theta) \equiv \left( \frac{1}{a}\frac{\partial S_{n,m}(\theta)}{\partial \theta}\sin m\lambda, \frac{mS_{n,m}(\theta)}{a\sin\theta}\cos m\lambda \right). \tag{51d}$$

From Eqs. (48), (49), and (16)–(21), we obtain

$$\begin{bmatrix} u_m^{c(s),N}(\theta) \\ v_m^{c(s),N}(\theta) \end{bmatrix} = \begin{cases} \sum_{n=1}^{N} \begin{bmatrix} u_{n,m}^{c(s)} \\ v_{n,m}^{c(s)} \end{bmatrix} \sin n\theta & \text{for } m = 0, \\[2ex] \sum_{n=0}^{N} \begin{bmatrix} u_{n,m}^{c(s)} \\ v_{n,m}^{c(s)} \end{bmatrix} \cos n\theta & \text{for } m = 1, \\[2ex] \sum_{n=1}^{N} \begin{bmatrix} u_{n,m}^{c(s)} \\ v_{n,m}^{c(s)} \end{bmatrix} \sin n\theta & \text{for even } m \geq 2, \\[2ex] \sum_{n=0}^{N} \begin{bmatrix} u_{n,m}^{c(s)} \\ v_{n,m}^{c(s)} \end{bmatrix} \cos n\theta \left( = \sum_{n=1}^{N-1} \begin{bmatrix} u_{n,m}^{c(s)\prime} \\ v_{n,m}^{c(s)\prime} \end{bmatrix} \sin\theta \sin n\theta \right) & \text{for odd } m \geq 3, \end{cases} \tag{52}$$

where

$$u_{n,m}^c = \frac{1}{a}\left[-n\psi_{n,m}^c\right] \qquad (n = 1, \dots, N) \quad \text{for } m = 0, \tag{53a}$$

$$u_{n,m}^c = \frac{1}{a}\left[m\chi_{n,m}^s + \frac{n(\psi_{n-1,m}^c - \psi_{n+1,m}^c)}{2}\right] \qquad (n = 0, \dots, N) \quad \text{for } m = 1, \tag{53b}$$

$$\text{except for } u_{1,m}^c = \frac{1}{a}\left[m\chi_{1,m}^s + \frac{2\psi_{0,m}^c - \psi_{2,m}^c}{2}\right] \qquad (n = 1),$$

$$u_{n,m}^c = \frac{1}{a}\left[m\chi_{n,m}^s + \frac{n(\psi_{n-1,m}^c - \psi_{n+1,m}^c)}{2}\right] \qquad (n = 1, \dots, N) \quad \text{for even } m \geq 2, \tag{53c}$$

$$u_{n,m}^c = \frac{1}{a}\left[\frac{m(-\chi_{n-1,m}^s + \chi_{n+1,m}^s)}{2} + \frac{n(-\psi_{n-2,m}^c + 2\psi_{n,m}^c - \psi_{n+2,m}^c)}{4}\right] \quad (n = 0, \dots, N) \ \text{for odd } m \geq 3 \tag{53d}$$

$$\text{except for } u_{1,m}^c = \frac{1}{a}\left[\frac{m\chi_{2,m}^s}{2} + \frac{(3\psi_{1,m}^c - \psi_{3,m}^c)}{4}\right] \qquad (n = 1).$$

The equations for $u_{n,m}^s$ are the same as Eqs. (53b–d), except that $u_{n,m}^c$, $\chi_{n,m}^s$, and $\psi_{n,m}^c$ are replaced with $u_{n,m}^s$, $-\chi_{n,m}^c$, and $\psi_{n,m}^s$, respectively. The equations for $v_{n,m}^c$ are the same as Eqs. (53a–d), except that $u_{n,m}^c$, $\chi_{n,m}^s$, and $\psi_{n,m}^c$ are replaced with $v_{n,m}^c$, $\psi_{n,m}^s$, and $-\chi_{n,m}^c$, respectively. The equations for $v_{n,m}^s$ are the same as Eqs. (53b–d), except that $u_{n,m}^c$, $\chi_{n,m}^s$, and $\psi_{n,m}^c$ are replaced with $v_{n,m}^s$, $-\psi_{n,m}^c$, and $-\chi_{n,m}^s$, respectively.

From Eqs. (52) and (53), it can be seen that $u_m^{c,N}(\theta)$, $u_m^{s,N}(\theta)$, $v_m^{c,N}(\theta)$, and $v_m^{s,N}(\theta)$ at the poles are finite for $m = 1$ and zero for $m \neq 1$. Moreover, the following relations are satisfied for $m = 1$:

$$u_{m=1}^{c,N}(\theta) = -v_{m=1}^{s,N}(\theta)\left(= \frac{1}{a}\sum_{n=0}^{N-1}(\chi_{n,m=1}^s + \psi_{n,m=1}^c)\right) \quad \text{at } \theta = 0 \text{ (North Pole)}, \tag{54a}$$

$$u_{m=1}^{s,N}(\theta) = v_{m=1}^{c,N}(\theta)\left(= \frac{1}{a}\sum_{n=0}^{N-1}(-\chi_{n,m=1}^c + \psi_{n,m=1}^s)\right) \quad \text{at } \theta = 0 \text{ (North Pole)}, \tag{54b}$$

$$u_{m=1}^{c,N}(\theta) = v_{m=1}^{s,N}(\theta)\left(= \frac{1}{a}\sum_{n=0}^{N-1}(-1)^n(\chi_{n,m=1}^s - \psi_{n,m=1}^c)\right) \text{ at } \theta = \pi \text{ (South Pole)}, \tag{54c}$$

$$u_{m=1}^{s,N}(\theta) = -v_{m=1}^{c,N}(\theta) \left( = \frac{1}{a} \sum_{n=0}^{N-1} (-1)^n \left( -\chi_{n,m=1}^{c} - \psi_{n,m=1}^{s} \right) \right) \text{ at } \theta = \pi \text{ (South Pole).} \qquad (54d)$$

Thus, it is guaranteed that the wind vector $(u^{N,M}, v^{N,M})$ in Eqs. (48) and (49) is continuous at the poles.

### 3.6 New method to calculate expansion coefficients for a wind vector

We propose a new method that calculates the expansion coefficients $\chi_{n,m}^{c}$, $\chi_{n,m}^{s}$, $\psi_{n,m}^{c}$ and $\psi_{n,m}^{s}$ in Eqs. (48)–(50) using the least-squares method to minimize the error of $u^{N,M}(\lambda, \theta)$ and $v^{N,M}(\lambda, \theta)$ from $u(\lambda, \theta)$ and $v(\lambda, \theta)$ due to the meridional wavenumber truncation. First, the wind vector components $u$ and $v$ are expanded zonally as

$$\begin{bmatrix} u(\lambda, \theta) \\ v(\lambda, \theta) \end{bmatrix} \cong \sum_{m=0}^{M} \begin{bmatrix} u_m^c(\theta) \\ v_m^c(\theta) \end{bmatrix} \cos m\lambda + \sum_{m=1}^{M} \begin{bmatrix} u_m^s(\theta) \\ v_m^s(\theta) \end{bmatrix} \sin m\lambda, \qquad (55)$$

where $u_m^{c(s)}(\theta)$ and $v_m^{c(s)}(\theta)$ are calculated from $u(\lambda, \theta)$ and $v(\lambda, \theta)$ by the forward Fourier transform as

$$\begin{bmatrix} u_m^c(\theta) \\ v_m^c(\theta) \end{bmatrix} = \frac{a}{2\pi} \int_0^{2\pi} \cos m\lambda \begin{bmatrix} u(\lambda, \theta) \\ v(\lambda, \theta) \end{bmatrix} d\lambda, \quad a \equiv \begin{cases} 1 \text{ for } m = 0 \\ 2 \text{ for } m \geq 1, \end{cases} \qquad (56a)$$

$$\begin{bmatrix} u_m^s(\theta) \\ v_m^s(\theta) \end{bmatrix} = \frac{1}{\pi} \int_0^{2\pi} \sin m\lambda \begin{bmatrix} u(\lambda, \theta) \\ v(\lambda, \theta) \end{bmatrix} d\lambda. \qquad (56b)$$

The variables $u_m^{c(s)}(\theta)$ and $v_m^{c(s)}(\theta)$ are meridionally expanded as

$$\begin{bmatrix} u_m^{c(s)}(\theta) \\ v_m^{c(s)}(\theta) \end{bmatrix} \cong \begin{bmatrix} \tilde{u}_m^{c(s),N}(\theta) \\ \tilde{v}_m^{c(s),N}(\theta) \end{bmatrix} \equiv \begin{cases} \sum_{n=1}^{N} \begin{bmatrix} \tilde{u}_{n,m}^{c(s)} \\ \tilde{v}_{n,m}^{c(s)} \end{bmatrix} \sin n\theta & \text{for even } m, \\ \sum_{n=0}^{N} \begin{bmatrix} \tilde{u}_{n,m}^{c(s)} \\ \tilde{v}_{n,m}^{c(s)} \end{bmatrix} \cos n\theta & \text{for odd } m, \end{cases} \qquad (57)$$

where $\tilde{u}_{n,m}^{c(s)}$ and $\tilde{v}_{n,m}^{c(s)}$ are calculated from $u_m^{c(s)}(\theta)$ and $v_m^{c(s)}(\theta)$ by the forward discrete cosine or sine transform (See Appendix B).

Next, $\chi_{n,m}^{c}$, $\chi_{n,m}^{s}$, $\psi_{n,m}^{c}$, and $\psi_{n,m}^{s}$ are calculated to minimize the following error $F$ (the squared $L_2$ norm of the residual vector):

$$F \equiv \frac{1}{2\pi^2} \int_0^{2\pi} \int_0^{\pi} \left( R^u(\lambda, \theta)^2 + R^v(\lambda, \theta)^2 \right) d\theta \, d\lambda, \qquad (58)$$

where the residual vector $\left( R^u(\lambda, \theta), R^v(\lambda, \theta) \right)$ is defined as

$$R^u(\lambda, \theta) \equiv u^{N,M}(\lambda, \theta) - u(\lambda, \theta), \qquad (59a)$$

$$R^v(\lambda, \theta) \equiv v^{N,M}(\lambda, \theta) - v(\lambda, \theta). \qquad (59b)$$

From Eqs. (58), (59), and the equations $\partial F / \partial \chi_{m,n}^{c} = 0$, $\partial F / \partial \chi_{n,m}^{s} = 0$, $\partial F / \partial \psi_{n,m}^{c} = 0$, and $\partial F / \partial \psi_{n,m}^{s} = 0$ used in the least-squares method, we obtain

$$\frac{1}{2\pi^2}\int_0^{2\pi}\int_0^{\pi}\left[\frac{\partial u^{N,M}(\lambda,\theta)}{\partial\chi_{n,m}^c}R^u(\lambda,\theta)+\frac{\partial v^{N,M}(\lambda,\theta)}{\partial\chi_{n,m}^c}R^v(\lambda,\theta)\right]d\theta\,d\lambda=0,\qquad(60a)$$

$$\frac{1}{2\pi^2}\int_0^{2\pi}\int_0^{\pi}\left[\frac{\partial u^{N,M}(\lambda,\theta)}{\partial\chi_{n,m}^s}R^u(\lambda,\theta)+\frac{\partial v^{N,M}(\lambda,\theta)}{\partial\chi_{n,m}^s}R^v(\lambda,\theta)\right]d\theta\,d\lambda=0,\qquad(60b)$$

$$\frac{1}{2\pi^2}\int_0^{2\pi}\int_0^{\pi}\left[\frac{\partial u^{N,M}(\lambda,\theta)}{\partial\psi_{n,m}^c}R^u(\lambda,\theta)+\frac{\partial v^{N,M}(\lambda,\theta)}{\partial\psi_{n,m}^c}R^v(\lambda,\theta)\right]d\theta\,d\lambda=0,\qquad(60c)$$

$$\frac{1}{2\pi^2}\int_0^{2\pi}\int_0^{\pi}\left[\frac{\partial u^{N,M}(\lambda,\theta)}{\partial\psi_{n,m}^s}R^u(\lambda,\theta)+\frac{\partial v^{N,M}(\lambda,\theta)}{\partial\psi_{n,m}^s}R^v(\lambda,\theta)\right]d\theta\,d\lambda=0.\qquad(60d)$$

5  From Eq. (50), we derive

$$\left(\frac{\partial u^{N,M}(\lambda,\theta)}{\partial\chi_{n,m}^c},\frac{\partial v^{N,M}(\lambda,\theta)}{\partial\chi_{n,m}^c}\right)=\mathbf{V}_{n,m}^1(\lambda,\theta),\qquad(61a)$$

$$\left(\frac{\partial u^{N,M}(\lambda,\theta)}{\partial\chi_{n,m}^s},\frac{\partial v^{N,M}(\lambda,\theta)}{\partial\chi_{n,m}^s}\right)=\mathbf{V}_{n,m}^2(\lambda,\theta),\qquad(61b)$$

$$\left(\frac{\partial u^{N,M}(\lambda,\theta)}{\partial\psi_{n,m}^c},\frac{\partial v^{N,M}(\lambda,\theta)}{\partial\psi_{n,m}^c}\right)=\mathbf{V}_{n,m}^3(\lambda,\theta),\qquad(61c)$$

$$\left(\frac{\partial u^{N,M}(\lambda,\theta)}{\partial\psi_{n,m}^s},\frac{\partial v^{N,M}(\lambda,\theta)}{\partial\psi_{n,m}^s}\right)=\mathbf{V}_{n,m}^4(\lambda,\theta).\qquad(61d)$$

10  Equations (60) and (61) show that the residual vector $\left(R^u(\lambda,\theta),R^v(\lambda,\theta)\right)$ is orthogonal to each of the vector basis function, which means that Eq. (60) is the same as the equation obtained by the Galerkin method. From Eqs. (60), (61), (51), (48a), (49a), (56), (A3) and (D6), we derive

$$\frac{1}{\pi}\int_0^{\pi}\left[-\frac{mS_{n,m}(\theta)}{a\sin\theta}\left(u_m^{s,N}(\theta)-\tilde{u}_m^{s,N}(\theta)\right)-\frac{1}{a}\frac{\partial S_{n,m}(\theta)}{\partial\theta}\left(v_m^{c,N}(\theta)-\tilde{v}_m^{c,N}(\theta)\right)\right]d\theta=0,\qquad(62a)$$

$$\frac{1}{\pi}\int_0^{\pi}\left[\frac{mS_{n,m}(\theta)}{a\sin\theta}\left(u_m^{c,N}(\theta)-\tilde{u}_m^{c,N}(\theta)\right)-\frac{1}{a}\frac{\partial S_{n,m}(\theta)}{\partial\theta}\left(v_m^{s,N}(\theta)-\tilde{v}_m^{s,N}(\theta)\right)\right]d\theta=0,\qquad(62b)$$

$$\frac{1}{\pi}\int_0^{\pi}\left[\frac{1}{a}\frac{\partial S_{n,m}(\theta)}{\partial\theta}\left(u_m^{c,N}(\theta)-\tilde{u}_m^{c,N}(\theta)\right)-\frac{mS_{n,m}(\theta)}{a\sin\theta}\left(v_m^{s,N}(\theta)-\tilde{v}_m^{s,N}(\theta)\right)\right]d\theta=0,\qquad(62c)$$

$$\frac{1}{\pi}\int_0^{\pi}\left[\frac{1}{a}\frac{\partial S_{n,m}(\theta)}{\partial\theta}\left(u_m^{s,N}(\theta)-\tilde{u}_m^{s,N}(\theta)\right)+\frac{mS_{n,m}(\theta)}{a\sin\theta}\left(v_m^{c,N}(\theta)-\tilde{v}_m^{c,N}(\theta)\right)\right]d\theta=0.\qquad(62d)$$

By substituting Eqs. (52) and (57) into Eq. (62a), the following equations for $\chi_{n,m}^c$ and $\psi_{n,m}^s$ are derived as shown in Appendix H.

For $m=0$,

$$\frac{1}{a}\left[n\chi_{n,m}^c\right]=\tilde{v}_{n,m}^c\qquad(1\leq n\leq N).\qquad(63a)$$

The coefficient $\chi_{m=0,n=0}^c$ is determined so that the global means of $\chi$ are zero. See Eq. (G1) about the calculation of the global mean.

For $m = 1$,

$$\frac{1}{a}\left[-(n-1)^2\chi_{n-2,m}^{c} - 2m\psi_{n-1,m}^{s} + (4m^2 + 2n^2 + 2)\chi_{n,m}^{c} - 2m\psi_{n+1,m}^{s} - (n+1)^2\chi_{n+2,m}^{c}\right]$$

$$= 2(n-1)\tilde{v}_{n-1,m}^{c} - 4m\tilde{u}_{n,m}^{s} - 2(n+1)\tilde{v}_{n+1,m}^{c} \quad (0 \le n \le N-1), \qquad (63b)$$

with the exception of the following underlined values:

$$\frac{1}{a}\left[(\underline{8m^2 + 4})\chi_{0,m}^{c} - \underline{4}m\psi_{1,m}^{s} - \underline{2}\chi_{2,m}^{c}\right] = -\underline{8}m\tilde{u}_{0,m}^{s} - \underline{4}\tilde{v}_{1,m}^{c} \quad (n = 0),$$

$$\frac{1}{a}\left[-\underline{4}m\psi_{0,m}^{s} + (4m^2 + 4)\chi_{1,m}^{c} + \cdots\right] = \cdots \quad (n = 1),$$

$$\frac{1}{a}\left[-\underline{2}\chi_{0,m}^{c} - 2m\psi_{1,m}^{s} + \cdots\right] = \cdots \quad (n = 2).$$

For even $m \ge 2$,

$$\frac{1}{a}\left[-(n-1)^2\chi_{n-2,m}^{c} - 2m\psi_{n-1,m}^{s} + (4m^2 + 2n^2 + 2)\chi_{n,m}^{c} - 2m\psi_{n+1,m}^{s} - (n+1)^2\chi_{n+2,m}^{c}\right]$$

$$= 2(n-1)\tilde{v}_{n-1,m}^{c} - 4m\tilde{u}_{n,m}^{s} - 2(n+1)\tilde{v}_{n+1,m}^{c} \quad (1 \le n \le N-1), \qquad (63c)$$

with no exception.

For odd $m \ge 3$,

$$\frac{1}{a}\left[(n-2)^2\chi_{n-4,m}^{c} + 2m\psi_{n-3,m}^{s} + (-4m^2 - 4n^2 + 8n - 8)\chi_{n-2,m}^{c} - 2m\psi_{n-1,m}^{s} + (8m^2 + 6n^2 + 8)\chi_{n,m}^{c}\right.$$

$$\left. -2m\psi_{n+1,m}^{s} + (-4m^2 - 4n^2 - 8n - 8)\chi_{n+2,m}^{c} + 2m\psi_{n+3,m}^{s} + (n+2)^2\chi_{n+4,m}^{c}\right]$$

$$= 4(n-2)\tilde{v}_{n-2,m}^{c} - 8m\tilde{u}_{n-1,m}^{s} - 8n\tilde{v}_{n,m}^{c} + 8m\tilde{u}_{n+1,m}^{s} + 4(n+2)\tilde{v}_{n+2,m}^{c} \quad (1 \le n \le N-2), \qquad (63d)$$

with the exception of the following underlined values:

$$\frac{1}{a}\left[(\underline{12m^2 + 18})\chi_{1,m}^{c} - \underline{4}m\psi_{2,m}^{s} + (-4m^2 - \underline{21})\chi_{3,m}^{c} + \cdots\right] = -\underline{16}m\tilde{u}_{0,m}^{s} - \underline{12}\tilde{v}_{1,m}^{c} + \cdots \quad (n = 1),$$

$$\frac{1}{a}\left[-\underline{4}m\psi_{1,m}^{s} + (8m^2 + 32)\chi_{2,m}^{c} + \cdots\right] = \cdots \quad (n = 2),$$

$$\frac{1}{a}\left[(-4m^2 - \underline{21})\chi_{1,m}^{c} - 2m\psi_{2,m}^{s} + \cdots\right] = \cdots \quad (n = 3).$$

Similarly, from Eq. (62b), we derive the same equations as Eqs. (63b–d), except that $\chi^{c}$, $\psi^{s}$, $\tilde{v}^{c}$, and $\tilde{u}^{s}$ are replaced with $\chi^{s}$, $-\psi^{c}$, $\tilde{v}^{s}$, and $-\tilde{u}^{c}$, respectively. From Eq. (62c), we derive the same equations as Eqs. (63a–d), except that $\chi^{c}$, $\psi^{s}$, $\tilde{v}^{c}$, and $\tilde{u}^{s}$ are replaced with $-\psi^{c}$, $\chi^{s}$, $\tilde{u}^{c}$, and $-\tilde{v}^{s}$, respectively. From Eq. (62d), we derive the same equations as Eqs. (63b–d), except that $\chi^{c}$, $\psi^{s}$, $\tilde{v}^{c}$, and $\tilde{u}^{s}$ are replaced with $\psi^{s}$, $\chi^{c}$, $-\tilde{u}^{s}$, and $-\tilde{v}^{c}$, respectively.

Eq. (63a) is easily solved. From Eqs. (63d), and from the same equations as Eqs. (63d), except that $\chi^{c}$, $\psi^{s}$, $\tilde{v}^{c}$, and $\tilde{u}^{s}$ are replaced with $\psi^{s}$, $\chi^{c}$, $-\tilde{u}^{s}$, and $-\tilde{v}^{c}$, respectively, we derive the following linear simultaneous equations for $m \ge 3$:

$$\mathbf{E}_m \begin{bmatrix} \chi_{1,m}^{\mathrm{c}} \\ \psi_{2,m}^{\mathrm{s}} \\ \chi_{3,m}^{\mathrm{c}} \\ \psi_{4,m}^{\mathrm{s}} \\ \vdots \end{bmatrix} = \mathbf{F}_m \begin{bmatrix} \tilde{u}_{0,m}^{\mathrm{s}} \\ \tilde{v}_{1,m}^{\mathrm{c}} \\ \tilde{u}_{2,m}^{\mathrm{s}} \\ \tilde{v}_{3,m}^{\mathrm{c}} \\ \vdots \end{bmatrix}, \qquad \mathbf{E}_m \begin{bmatrix} \psi_{1,m}^{\mathrm{s}} \\ \chi_{2,m}^{\mathrm{c}} \\ \psi_{3,m}^{\mathrm{s}} \\ \chi_{4,m}^{\mathrm{c}} \\ \vdots \end{bmatrix} = \mathbf{F}_m \begin{bmatrix} -\tilde{v}_{0,m}^{\mathrm{c}} \\ -\tilde{u}_{1,m}^{\mathrm{s}} \\ -\tilde{v}_{2,m}^{\mathrm{c}} \\ -\tilde{u}_{3,m}^{\mathrm{s}} \\ \vdots \end{bmatrix}, \tag{64}$$

where the matrices $\mathbf{E}_m$ are nine-diagonal. From Eqs. (63b,c), we derive the equations similar to Eq. (64) for $m = 1$ and even $m\ (\geq 2)$ with penta-diagonal matrices $\mathbf{E}_m$. The simultaneous equations with nine-diagonal or penta-diagonal matrices $\mathbf{E}_m$ can be solved in a similar way to Eq. (31), and the expansion coefficients $\chi_{n,m}^{\mathrm{c}}$ and $\psi_{n,m}^{\mathrm{s}}$ in Eq. (64) can be calculated efficiently. From the same equations as Eqs. (63b–d), except that $\chi^{\mathrm{c}}$, $\psi^{\mathrm{s}}$, $\tilde{v}^{\mathrm{c}}$, and $\tilde{u}^{\mathrm{s}}$ are replaced with $\chi^{\mathrm{s}}$, $-\psi^{\mathrm{c}}$, $\tilde{v}^{\mathrm{s}}$, and $-\tilde{u}^{\mathrm{c}}$, respectively, and the same equations as Eqs. (63b–d), except that $\chi^{\mathrm{c}}$, $\psi^{\mathrm{s}}$, $\tilde{v}^{\mathrm{c}}$, and $\tilde{u}^{\mathrm{s}}$ are replaced with $-\psi^{\mathrm{c}}$, $\chi^{\mathrm{s}}$, $\tilde{u}^{\mathrm{c}}$, and $-\tilde{v}^{\mathrm{s}}$, respectively, the simultaneous equations similar to Eq. (64) are also derived. Thus, the expansion coefficients $\chi_{n,m}^{\mathrm{c}}$, $\chi_{n,m}^{\mathrm{s}}$, $\psi_{n,m}^{\mathrm{c}}$, and $\psi_{n,m}^{\mathrm{s}}$ are calculated from $\tilde{u}_{n,m}^{\mathrm{c}}$, $\tilde{u}_{n,m}^{\mathrm{s}}$, $\tilde{v}_{n,m}^{\mathrm{c}}$, and $\tilde{v}_{n,m}^{\mathrm{s}}$ using Eqs. (63a–d) and the similar equations. The expansion coefficients $u_{n,m}^{\mathrm{c}}$, $u_{n,m}^{\mathrm{s}}$, $v_{n,m}^{\mathrm{c}}$, and $v_{n,m}^{\mathrm{s}}$ are calculated from $\chi_{n,m}^{\mathrm{c}}$, $\chi_{n,m}^{\mathrm{s}}$, $\psi_{n,m}^{\mathrm{c}}$, and $\psi_{n,m}^{\mathrm{s}}$ using Eq. (53) for $u_{n,m}^{\mathrm{c}}$ and the similar equations for $u_{n,m}^{\mathrm{s}}$, $v_{n,m}^{\mathrm{c}}$, and $v_{n,m}^{\mathrm{s}}$.

This method to calculate the DFS expansion coefficients of $\chi$ and $\psi$ from $u$ and $v$ using the least-squares method (or the Galerkin method with the DFS vector basis functions) is similar to the vector harmonic transform method (Browning et al., 1989; Swarztrauber, 1993), where the SH expansion coefficients of the divergence $D = \nabla^2 \chi$ and the vorticity $\zeta = \nabla^2 \psi$ are calculated from the grid-point values of $u$ and $v$ using the Galerkin spectral method with the orthogonal vector SH basis functions.

## 3.7 Laplacian operator and Poisson equation

The calculation of the Laplacian operator and the Poisson equation in the new DFS method is described here. In the equation

$$g(\lambda, \theta) = \nabla^2 f(\lambda, \theta) = \frac{1}{a^2}\left[\frac{1}{\sin^2\theta}\frac{\partial^2 f}{\partial\lambda^2} + \frac{1}{\sin\theta}\frac{\partial}{\partial\theta}\left(\sin\theta\frac{\partial f}{\partial\theta}\right)\right], \tag{65}$$

where $\nabla^2$ is the Laplacian operator, the variables $f$ and $g$ are expanded zonally like Eq. (8) as

$$\begin{bmatrix} f(\lambda, \theta) \\ g(\lambda, \theta) \end{bmatrix} \cong \sum_{m=0}^{M} \begin{bmatrix} f_m^{\mathrm{c}}(\theta) \\ g_m^{\mathrm{c}}(\theta) \end{bmatrix} \cos m\lambda + \sum_{m=1}^{M} \begin{bmatrix} f_m^{\mathrm{s}}(\theta) \\ g_m^{\mathrm{s}}(\theta) \end{bmatrix} \sin m\lambda. \tag{66}$$

The variables $f_m^{\mathrm{c}}(\theta)$, $f_m^{\mathrm{s}}(\theta)$, $g_m^{\mathrm{c}}(\theta)$, and $g_m^{\mathrm{s}}(\theta)$ are expanded meridionally like Eq. (10) as

$$\begin{bmatrix} f_m^{\mathrm{c(s)}}(\theta) \\ g_m^{\mathrm{c(s)}}(\theta) \end{bmatrix} \cong \begin{bmatrix} f_m^{\mathrm{c(s)},N}(\theta) \\ g_m^{\mathrm{c(s)},N}(\theta) \end{bmatrix} \equiv \sum_{n=N_{\min,m}}^{N_{\max,m}} \begin{bmatrix} f_{n,m}^{\mathrm{c(s)}} \\ g_{n,m}^{\mathrm{c(s)}} \end{bmatrix} S_{n,m}(\theta). \tag{67}$$

We define the truncated variables $f^{N,M}(\theta)$ and $g^{N,M}(\theta)$ as

$$\begin{bmatrix} f^{N,M}(\lambda, \theta) \\ g^{N,M}(\lambda, \theta) \end{bmatrix} \equiv \sum_{m=0}^{M} \begin{bmatrix} f_m^{\mathrm{c},N}(\theta) \\ g_m^{\mathrm{c},N}(\theta) \end{bmatrix} \cos m\lambda + \sum_{m=1}^{M} \begin{bmatrix} f_m^{\mathrm{s},N}(\theta) \\ g_m^{\mathrm{s},N}(\theta) \end{bmatrix} \sin m\lambda. \tag{68}$$

From Eqs. (65) and (68), we obtain

$$\nabla^2 f^{N,M}(\lambda, \theta) = \sum_{m=0}^{M} \frac{1}{a^2} \left[ \frac{-m^2}{\sin^2 \theta} f_m^{c,N}(\theta) + \frac{1}{\sin \theta} \frac{\partial}{\partial \theta} \left( \sin \theta \frac{\partial f_m^{c,N}(\theta)}{\partial \theta} \right) \right] \cos m\lambda$$

$$+ \sum_{m=1}^{M} \frac{1}{a^2} \left[ \frac{-m^2}{\sin^2 \theta} f_m^{s,N}(\theta) + \frac{1}{\sin \theta} \frac{\partial}{\partial \theta} \left( \sin \theta \frac{\partial f_m^{s,N}(\theta)}{\partial \theta} \right) \right] \sin m\lambda. \tag{69}$$

Here we use the Galerkin method to calculate the Laplacian operator and the Poisson equation, and obtain

$$\frac{1}{2\pi^2} \int_0^{2\pi} \int_0^{\pi} S_{n,m}(\theta) \begin{bmatrix} \cos m\lambda \\ \sin m\lambda \end{bmatrix} R^g(\lambda, \theta) d\theta d\lambda = 0, \tag{70}$$

where the residual

$$R^g(\lambda, \theta) \equiv g^{N,M}(\lambda, \theta) - \nabla^2 f^{N,M}(\lambda, \theta) \tag{71}$$

is orthogonal to each of the new DFS basis functions $S_{m,n}(\theta) \cos m\lambda$ and $S_{m,n}(\theta) \sin m\lambda$.

We can also use the least-squares method instead of the Galerkin method so that the following error $H$ (the squared $L_2$ norm of the residual) is minimized:

$$H \equiv \frac{1}{2\pi^2} \int_0^{2\pi} \int_0^{\pi} R^g(\lambda, \theta)^2 d\theta \, d\lambda. \tag{72}$$

When calculating $g$ by applying the Laplacian operator to a given $f$, $g_{n,m}^c$ and $g_{n,m}^s$ can also be calculated from $\partial H/\partial g_{n,m}^c$ and $\partial H/\partial g_{n,m}^s$ using the least-squares method. The equations $\partial H/\partial g_{n,m}^c$ and $\partial H/\partial g_{n,m}^s$ give the equivalent equations to Eq. (70). When calculating $f$ from a given $g$ in the Poisson equation, $f_{n,m}^c$ and $f_{n,m}^s$ can also be calculated from $\partial H/\partial f_{n,m}^c$ and $\partial H/\partial f_{n,m}^s$ using the least-squares method. However, the equations derived from $\partial H/\partial f_{n,m}^c$ and $\partial H/\partial f_{n,m}^s$ are different from Eq. (70). If we use different equations for calculating $g$ from $f$ and $f$ from $g$, the original values are changed when calculating $g$ from $f$ followed by calculating $f$ from $g$, which may be not good for numerical stability. Therefore, we use Eq. (70) obtained with the Galerkin method for calculating both $g$ from $f$ and $f$ from $g$. Generally, it cannot be said that the least-squares method is superior to the Galerkin method or vice versa, and here we choose to use the Galerkin method because of the reason described above.

From Eqs. (68)–(71) and Eq. (A3) we derive

$$\int_0^{\pi} S_{n,m}(\theta) \left\{ g_m^{c(s),N}(\theta) - \frac{1}{a^2} \left[ \frac{-m^2}{\sin^2 \theta} f_m^{c(s),N}(\theta) + \frac{1}{\sin \theta} \frac{\partial}{\partial \theta} \left( \sin \theta \frac{\partial f_m^{c(s),N}(\theta)}{\partial \theta} \right) \right] \right\} d\theta = 0, \tag{73}$$

For $m = 0$, we calculate $g_{n,m}^{c(s)}$ by using

$$g_m^{c(s),N}(\theta) = \frac{1}{a^2} \left[ \frac{-m^2}{\sin^2 \theta} f_m^{c(s),N}(\theta) + \frac{1}{\sin \theta} \frac{\partial}{\partial \theta} \left( \sin \theta \frac{\partial f_m^{c(s),N}(\theta)}{\partial \theta} \right) \right], \tag{74}$$

instead of Eq. (73) following Yee (1981) and Cheong (2000a) for ease of calculation. For $0 \le m \le 3$, the exact solutions of $g_{n,m}^{c(s)}$ can be obtained from Eq. (74) because the new DFS meridional basis functions for $0 \le m \le 3$ are the linear combination of the associated Legendre functions for $0 \le m \le 3$ and vice versa as described in Sect. 3.4.

For $m = 0$, by substituting Eq. (67) into Eq. (74) multiplied by $\sin^2 \theta$, transforming using Eqs. (A2d) and (A5b), and comparing both sides of the equation, we obtain

$$-g_{n-2,m}^{c(s)} + 2g_{n,m}^{c(s)} - g_{n+2,m}^{c(s)} = \frac{1}{a^2}\left[(n-1)(n-2)f_{n-2,m}^{c(s)} - 2n^2 f_{n,m}^{c(s)} + (n+1)(n+2)f_{n+2,m}^{c(s)}\right] \quad (0 \le n \le N), \quad (75a)$$

except for the following underlined values:

$$\underline{1}g_{1,m}^{c(s)} - g_{3,m}^{c(s)} = \cdots \qquad (n = 1),$$

$$-\underline{2}g_{0,m}^{c(s)} + 2g_{2,m}^{c(s)} - g_{4,m}^{c(s)} = \cdots \qquad (n = 2).$$

For $m = 1$, by substituting Eqs. (67) into Eq. (73) and using Eqs. (A2d), (A4a) and (A5b), we obtain

$$-g_{n-2,m}^{c(s)} + 2g_{n,m}^{c(s)} - g_{n+2,m}^{c(s)} = \frac{1}{a^2}\left[(n-1)n f_{n-2,m}^{c(s)} - (2n^2 + 4m^2)f_{n,m}^{c(s)} + (n+1)n f_{n+2,m}^{c(s)}\right] \quad (0 \le n \le N - 1), \quad (75b)$$

except for the following underlined values:

$$\underline{1}g_{1,m}^{c(s)} - g_{3,m}^{c(s)} = \cdots \qquad (n = 1),$$

$$-\underline{2}g_{0,m}^{c(s)} + 2g_{2,m}^{c(s)} - g_{4,m}^{c(s)} = \frac{1}{a^2}\left[\underline{4}f_{0,m}^{c(s)} + \cdots\right] \qquad (n = 2).$$

For even $m \ge 2$, by substituting Eqs. (67) into Eq. (73) and using Eqs. (A2c), (A4b) and (A5d), we obtain

$$-g_{n-2,m}^{c(s)} + 2g_{n,m}^{c(s)} - g_{n+2,m}^{c(s)} = \frac{1}{a^2}\left[(n-1)n f_{n-2,m}^{c(s)} - (2n^2 + 4m^2)f_{n,m}^{c(s)} + (n+1)n f_{n+2,m}^{c(s)}\right] \quad (1 \le n \le N - 1), \quad (75c)$$

except for the following underlined values:

$$\underline{3}g_{1,m}^{c(s)} - g_{3,m}^{c(s)} = \cdots \qquad (n = 1).$$

For odd $m \ge 3$, by substituting Eqs. (67) into Eq. (73) and using Eqs. (A2c,e), (A4b) and (A5d), we obtain

$$g_{n-4,m}^{c(s)} - 4g_{n-2,m}^{c(s)} + 6g_{n,m}^{c(s)} - 4g_{n+2,m}^{c(s)} + g_{n+4,m}^{c(s)}$$

$$= \frac{1}{a^2}\left[-(n-2)(n-1)f_{n-4,m}^{c(s)} + (4n^2 - 6n + 4 + 4m^2)f_{n-2,m}^{c(s)} - (6n^2 + 4 + 8m^2)f_{n,m}^{c(s)}\right.$$

$$\left. +(4n^2 + 6n + 4 + 4m^2)f_{n+2,m}^{c(s)} - (n+2)(n+1)f_{n+4,m}^{c(s)}\right] \qquad (1 \le n \le N - 2), \quad (75d)$$

except for the following underlined values:

$$\underline{10}g_{1,m}^{c(s)} - \underline{5}g_{3,m}^{c(s)} + g_{5,m}^{c(s)} = \frac{1}{a^2}\left[-(\underline{12} + \underline{12}m^2)f_{1,m}^{c(s)} + \cdots\right] \qquad (n = 1),$$

$$\underline{5}g_{2,m}^{c(s)} - 4g_{4,m}^{c(s)} + g_{6,m}^{c(s)} = \cdots \qquad (n = 2),$$

$$-\underline{5}g_{1,m}^{c(s)} + 6g_{3,m}^{c(s)} - 4g_{5,m}^{c(s)} + g_{7,m}^{c(s)} = \frac{1}{a^2}\left[(\underline{24} + 4m^2)f_{1,m}^{c(s)} + \cdots\right] \qquad (n = 3).$$

From Eq. (75), we obtain the following two linear simultaneous equations with tri-diagonal or penta-diagonal matrices:

$$\mathbf{A}_{n\_even,m}\boldsymbol{g}_{n\_even,m}^{c(s)} = \mathbf{B}_{n\_even,m}\boldsymbol{f}_{n\_even,m}^{c(s)} \quad , \quad \mathbf{A}_{n\_odd,m}\boldsymbol{g}_{n\_odd,m}^{c(s)} = \mathbf{B}_{n\_odd,m}\boldsymbol{f}_{n\_odd,m}^{c(s)} \quad , \tag{76}$$

where $\boldsymbol{g}_{n\_even,m}^{c(s)}$ and $\boldsymbol{g}_{n\_odd,m}^{c(s)}$ are the vectors whose components are $g_{n,m}^{c(s)}$ ($n$ is even) and $g_{n,m}^{c(s)}$ ($n$ is odd), respectively, and $\boldsymbol{f}_{n\_even,m}^{c(s)}$ and $\boldsymbol{f}_{n\_odd,m}^{c(s)}$ are the vectors whose components are $f_{n,m}^{c(s)}$ ($n$ is even) and $f_{n,m}^{c(s)}$ ($n$ is odd), respectively; $\mathbf{A}_{n\_even,m}$, $\mathbf{B}_{n\_even,m}$, $\mathbf{A}_{n\_odd,m}$ and $\mathbf{B}_{n\_odd,m}$ are tri-diagonal or penta-diagonal matrices. $\boldsymbol{g}_{m,n\_even}^{c(s)}$ and $\boldsymbol{g}_{m,n\_odd}^{c(s)}$ are calculated by

$$\boldsymbol{g}_{n\_even,m}^{c(s)} = \mathbf{A}_{n\_even,m}^{-1}\mathbf{B}_{n\_even,m}\boldsymbol{f}_{n\_even,m}^{c(s)} \quad , \quad \boldsymbol{g}_{n\_odd,m}^{c(s)} = \mathbf{A}_{n\_odd,m}^{-1}\mathbf{B}_{n\_odd,m}\boldsymbol{f}_{n\_odd,m}^{c(s)} \quad , \tag{77}$$

which can be solved efficiently as in Eq. (31). We have verified that all the eigenvalues of the matrices $\mathbf{A}_{n\_even,m}^{-1}\mathbf{B}_{n\_even,m}$ and $\mathbf{A}_{n\_odd,m}^{-1}\mathbf{B}_{n\_odd,m}$ are negative real numbers for several truncation wavenumbers $M$ and $N$, but we have not yet proved that this is true for all truncation wavenumbers.

In the Poisson equation, $f$ is calculated from given $g$ in Eq. (65). We calculate $f$ from $g$ by the reverse calculation of $g$ from $f$ in Eq. (77). That is, we calculate $f$ from $g$ by

$$\boldsymbol{f}_{n\_even,m}^{c(s)} = \mathbf{B}_{n\_even,m}^{-1}\mathbf{A}_{n\_even,m}\boldsymbol{g}_{n\_even,m}^{c(s)} \quad , \quad \boldsymbol{g}_{n\_odd,m}^{c(s)} = \mathbf{B}_{n\_odd,m}^{-1}\mathbf{A}_{n\_odd,m}\boldsymbol{f}_{n\_odd,m}^{c(s)} \quad , \tag{78}$$

except when $m = 0$ and $n$ is even. For $m = 0$, $f_{n=0,m=0}^{c}$ disappears in Eq. (75a). The coefficients $f_{n,m=0}^{c}$ (even $n \geq 2$) are calculated from $g_{n,m=0}^{c}$ (even $n \geq 2$) by using Eq. (75a). The value $f_{n=0,m=0}^{c}$ is calculated from $f_{n,m=0}^{c}$ (even $n \geq 2$) so that the global mean of $f$ is zero using Eq. (G1).

In Eq. (65), the global mean of $g$ must be zero because the global mean of the right-hand side of Eq. (65) is zero. Before calculating $f$ from a given $g$ in the Poisson equation, we should subtract the global mean from $g$ (Cheong 2000b). See Eq. (G1) about the calculation of the global mean.

## 3.8 The Helmholtz equation

The Helmholtz equation is

$$f - \varepsilon\nabla^2 f = \left\{1 - \varepsilon\frac{1}{a^2}\left[\frac{1}{\sin^2\theta}\frac{\partial^2}{\partial\lambda^2} + \frac{1}{\sin\theta}\frac{\partial}{\partial\theta}\left(\sin\theta\frac{\partial}{\partial\theta}\right)\right]\right\}f = g, \tag{79}$$

where $f$ is calculated from given $g$. From Eq. (76), the Poisson equation in Eq. (65) is represented as

$$\mathbf{A}\boldsymbol{g} = \mathbf{B}\boldsymbol{f}, \tag{80}$$

where the subscripts n_even, n_odd and $m$, and the superscripts c and s are omitted. Similarly, by using the Galerkin method, Eq. (79) is represented as

$$\mathbf{A}\boldsymbol{f} - \varepsilon\mathbf{B}\boldsymbol{f} = \mathbf{A}\boldsymbol{g}. \tag{81}$$

From Eq. (81), $\boldsymbol{f}$ is calculated from $\boldsymbol{g}$ by

$$\boldsymbol{f} = (\mathbf{A} - \varepsilon\mathbf{B})^{-1}\mathbf{A}\boldsymbol{g}. \tag{82}$$

Since $\mathbf{A} - \varepsilon\mathbf{B}$ is a penta-diagonal or tri-diagonal matrix, Eq. (82) can be efficiently solved as in Eq. (31).

Similarly, the Helmholtz-like equation

$$f - \varepsilon\nabla^2 f = \nabla^2 g \tag{83}$$

is represented as

$$\mathbf{A}f - \varepsilon\mathbf{B}f = \mathbf{B}g. \tag{84}$$

From Eq. (84), $f$ is calculated from $g$ by

$$f = (\mathbf{A} - \varepsilon\mathbf{B})^{-1}\mathbf{B}g. \tag{85}$$

## 3.9 Horizontal diffusion

The horizontal diffusion is calculated in the similar way as in Cheong et al. (2004). Here we describe how to calculate fourth-order diffusion. Higher-order diffusion can be calculated similarly.

The equation for fourth-order hyperdiffusion is

$$f + \varepsilon\nabla^4 f = g, \tag{86}$$

where $f$ is calculated from $g$. Equation (86) can be converted into

$$\left(1 + i\sqrt{\varepsilon}\nabla^2\right)\left(1 - i\sqrt{\varepsilon}\nabla^2\right)f = g, \tag{87}$$

where $i = \sqrt{-1}$. The calculation of Eq. (86) is accomplished by successive calculations of the following Helmholtz equations:

$$\left(1 + i\sqrt{\varepsilon}\nabla^2\right)f' = g, \tag{88a}$$

$$\left(1 - i\sqrt{\varepsilon}\nabla^2\right)f = f', \tag{88b}$$

which are represented as

$$\left(\mathbf{A} + i\sqrt{\varepsilon}\mathbf{B}\right)f' = \mathbf{A}g. \tag{89a}$$

$$\left(\mathbf{A} - i\sqrt{\varepsilon}\mathbf{B}\right)f = \mathbf{A}f'. \tag{89b}$$

From Eqs. (89), we obtain the equation to calculate $f$ from $g$ as

$$f = \left(\mathbf{A} - i\sqrt{\varepsilon}\mathbf{B}\right)^{-1}\mathbf{A}\left(\mathbf{A} + i\sqrt{\varepsilon}\mathbf{B}\right)^{-1}\mathbf{A}g. \tag{90}$$

Here, $\mathbf{A} - i\sqrt{\varepsilon}\mathbf{B}$ and $\mathbf{A} + i\sqrt{\varepsilon}\mathbf{B}$ are complex matrices and $f$ and $g$ are real column vectors. For efficient computation, two real column vectors can be combined into one complex column vector (Cheong et al., 2004); for example, $f = f^c - if^s$ and $g = g^c - ig^s$, where the superscript c indicates the zonal cosine component, and the superscript s indicates the zonal sine component.

## 3.10 Essential summary (cook book) of the new DFS method

The essential summary for a scalar variable:

1. Define DFS expansion for a scalar variable with zonal expansion in Eq. (8) and meridional expansion in Eq. (10).

2. For the inverse transform from spectral space to grid point space,

    a) Calculate the coefficients $T_{n,m}^{c(s)'}$ from $T_{n,m}^{c(s)}$ by using Eqs. (14).

    b) Calculate $T_{m}^{c(s),N}(\theta)$ in Eq. (13) from $T_{n,m}^{c(s)'}$ by inverse cosine and sine Fourier transforms in Appendix B.

c) Calculate the grid point values $T^{N,M}(\lambda, \theta)$ in Eq. (15) from $T_m^{\text{c(s)},N}(\theta)$ by inverse Fourier transform.

3. For the forward transform from grid point space to spectral space,

a) Calculate $T_m^{\text{c(s)}}(\theta)$ in Eq. (9) from the grid point values $T(\lambda, \theta)$ by forward Fourier transform.

b) Calculate the coefficients $\tilde{T}_{n,m}^{\text{c(s)}}$ in Eq. (23) from $T_m^{\text{c(s)}}(\theta)$ by forward cosine and sine transforms in Appendix B.

c) Calculate the coefficients $T_{n,m}^{\text{c(s)}}$ from $\tilde{T}_{n,m}^{\text{c(s)}}$ by using Eqs. (30) and (31). Here, the coefficients $T_{n,m}^{\text{c(s)}}$ are calculated so that Eq. (29) derived from the Galerkin method (or the least-squares method) is satisfied.

The essential summary for a vector variable:

1. Represent DFS expansion for a vector variable by Eq. (50).

2. For the inverse transform from spectral space to grid point space,

a) Calculate the coefficients $u_{n,m}^{\text{c(s)}}$ and $v_{n,m}^{\text{c(s)}}$ from $\chi_{n,m}^{\text{c(s)}}$ and $\psi_{n,m}^{\text{c(s)}}$ by using Eq. (53) and the similar equations.

b) Calculate $u_m^{\text{c(s)},N}(\theta)$ and $v_m^{\text{c(s)},N}(\theta)$ in Eq. (52) from $u_{n,m}^{\text{c(s)}}$ and $v_{n,m}^{\text{c(s)}}$ by inverse cosine and sine transforms in Appendix B.

c) Calculate the grid point values $u^{N,M}(\lambda, \theta)$ and $v^{N,M}(\lambda, \theta)$ in Eqs. (48a) and (49a) from $u_m^{\text{c(s)},N}(\theta)$ and $v_m^{\text{c(s)},N}(\theta)$ by inverse Fourier transform.

3. For the forward transform from grid point space to spectral space,

a) Calculate $u_m^{\text{c(s)}}(\theta)$ and $v_m^{\text{c(s)}}(\theta)$ in Eq. (56) from the grid point values $u(\lambda, \theta)$ and $v(\lambda, \theta)$ by forward Fourier transform.

b) Calculate the coefficients $\tilde{u}_{n,m}^{\text{c(s)}}$ and $\tilde{v}_{n,m}^{\text{c(s)}}$ in Eq. (57) from the grid point values $u_m^{\text{c(s)}}(\theta)$ and $v_m^{\text{c(s)}}(\theta)$ by forward cosine and sine transforms in Appendix B.

c) Calculate the coefficients $\chi_{n,m}^{\text{c(s)}}$ and $\psi_{n,m}^{\text{c(s)}}$ from $\tilde{u}_{n,m}^{\text{c(s)}}$ and $\tilde{v}_{n,m}^{\text{c(s)}}$ by using Eqs. (63) and (64). Here, $\chi_{n,m}^{\text{c(s)}}$ and $\psi_{n,m}^{\text{c(s)}}$ are calculated so that Eq. (62) derived from the Galerkin method (or the least-squares method) is satisfied.

## 4 The error due to meridional wavenumber truncation in DFS expansion methods

Here we examine the error due to the meridional wavenumber truncation when the same continuity conditions at the poles as Eq. (3) are satisfied. In the DFS method of Orszag (1974) using Eq. (2), only $f_{N-1,m}$ and $f_{N,m}$ are modified to satisfy Eq. (4) equivalent to Eq. (3). In the old DFS method using Eq. (6), which is proposed in Cheong (2000a, 2000b) and used in Yoshimura and Matsumura (2005), the DFS meridional basis functions automatically satisfy the pole conditions in Eq. (3) for even $m$, but not for odd $m$. In the new DFS method using Eqs. (10)–(12), the DFS meridional basis functions automatically satisfy the condition in Eq. (3) for both even and odd $m$. We examine the error due to the wavenumber truncation in these DFS methods while comparing it with the SH method.

Figure 2 shows the error due to the wavenumber truncation. The number of latitudinal grid points is $J = 64$. The initial values of $F_m(\theta_j)$ are set to one at the grid points north of 30°N (except for the North pole), and zero at the grid points south

of 30°N. Grid [0] is used in the DFS methods, and the Gaussian grid is used in the SH method. There are no grid points at the poles. Since the values at the poles are zero due to the pole conditions in Eq. (3), the initial values abruptly change around the North pole. The initial values are meridionally transformed from grid space to spectral space (forward transform), truncated with $N = 42$, and then transformed back from spectral space to grid space (inverse transform) to obtain the truncated reconstruction of $F_m(\theta_j)$.

In the DFS method of Orszag, a very large error occurs, especially for odd $|m|$ ($\geq 3$) (Fig. 2c), when $f_{N-1,m}$ and $f_{N,m}$ are modified to satisfy the pole conditions in Eq. (4). Dividing $F_m(\theta_j)$ by $\sin\theta$ before the forward Fourier cosine transform for odd $m$ also contributes to the large error.

In the old DFS method, large high wavenumber oscillations appear for even $m$ ($\neq 0$) in Fig. 2a. Although the basis functions for even $m$ ($\neq 0$) in the old DFS method are the same as those in the new method, the expansion coefficients are calculated differently in the two methods. In the old DFS method, the simple meridional truncation with $N < J$ after the forward Fourier sine transform of a variable divided by $\sin\theta$ causes the large high-wavenumber oscillations. The large oscillations appear especially when the initial values abruptly change around the poles. In the case shown in Fig. 2, the initial values at the grid points near the North Pole are one, but the value at the North Pole abruptly becomes zero due to the pole conditions of Eq. (3). The result in the old DFS method for odd $|m|$ ($\geq 3$) is not shown in Fig. 2c because the method does not satisfy the condition of Eq. (3) for odd $m$.

In the new DFS method, the usual small oscillations from the Gibbs phenomenon appear in Fig. 2. The error is small because the expansion coefficients are calculated using the least-squares method (or the Galerkin method) to minimize the error. Because of this, the truncation with arbitrary $N < J$ does not cause large oscillations in the new DFS method. The values for even $m$ ($\geq 2$) and odd $m$ ($\geq 3$) in the new DFS method are similar to those for $m = 2$ and $m = 3$ in the SH method, respectively. In the SH method, when $m$ is large, the values become close to zero at high latitudes.

When using the basis functions of Orszag in Eq. (2), we can also obtain results equivalent to the new DFS method by calculating the expansion coefficients using the least-squares method with Lagrange multipliers in order to minimize the error while satisfying the pole conditions in Eq. (4).

Figure 3a shows the same figure as Fig. 2a except for $N = 63$. In the old DFS method using Eq. (6), we set $N = 63$ for $m = 0$, and $N' = 64$ for $m \neq 0$. Because $N' = J$ for even $m$ ($\geq 2$), the forward transform followed by the inverse transform does not change the initial values at the grid points, and the oscillations do not appear in the old DFS method. For this reason, Yoshimura and Matsumura (2005) and Yoshimura (2012) set $N' = J$ for even $m$ ($\geq 2$) to improve stability. However, there is a problem with the latitudinal derivative in the old DFS method even when $N' = J$ for even $m$ ($\geq 2$). Fig. 3b is the same as Fig. 3a except that it also shows the values between grid points calculated from the expansion coefficients by using Eq. (6) or Eq. (10). The large oscillations appear in the old DFS method with Grid [0], and it makes the latitudinal derivative at the grid points unrealistically large. In the new DFS method with the least-squares method, the large oscillations do not appear.

## 5 Tests of the DFS methods with the Laplacian operator and the Helmholtz equation

We examine the accuracy of the old and new DFS methods for the Laplacian operator in Eq. (65) and the Helmholtz equation

$$(1 - \varepsilon \nabla^2)f = h. \tag{91}$$

Here, we give the function $f$ as

$$f = \begin{cases} \dfrac{H}{4}\left(1 + \cos\dfrac{\pi r}{R}\right)^2 & \text{if } r < R, \\ 0 & \text{if } r \geq R, \end{cases} \tag{92}$$

$$r = a\cos^{-1}[\sin\phi_c \sin\phi + \cos\phi_c \cos\phi \cos(\lambda - \lambda_c)], \tag{93}$$

where $H = 1000$, $R = a/3$, $\phi$ is latitude, $\lambda$ is longitude, $a$ is the radius of the earth and $r$ is the distance between $(\lambda, \phi)$ and the center $(\lambda_c, \phi_c) = (3\pi/2, \pi/2 - 0.05)$. The function $f$ is similar to the cosine bell in the Williamson test case 1 (Williamson et al., 1992), but $(1 + \cos\pi r/R)$ is squared so that the second derivative of $f$ is continuous. To easily calculate the exact values of $\nabla^2 f$, the center is temporarily set to the North Pole, that is, $(\lambda_c, \phi_c) = (0, \pi/2)$ and $r = a\cos^{-1}[\sin\phi] = a\theta$, where $\theta$ is colatitude. At this time, $g$ is calculated as follows:

$$g = \nabla^2 f = \frac{1}{a^2}\left[\frac{1}{\sin^2\theta}\frac{\partial^2 f}{\partial\lambda^2} + \frac{1}{\sin\theta}\frac{\partial}{\partial\theta}\left(\sin\theta\frac{\partial f}{\partial\theta}\right)\right]$$

$$= -\frac{\cos\theta}{\sin\theta}\frac{H}{2a^2}\frac{\pi a}{R}\left[\left(1 + \cos\frac{\pi r}{R}\right)\sin\frac{\pi r}{R}\right] + \frac{H}{2a^2}\left(\frac{\pi a}{R}\right)^2\left[\sin^2\frac{\pi r}{R} - \left(1 + \cos\frac{\pi r}{R}\right)\cos\frac{\pi r}{R}\right]. \tag{94}$$

Equation (94) is satisfied at any position of the center. The function $h$ in Eq. (91) is calculated by

$$h = (1 - \varepsilon\nabla^2)f = f - \varepsilon g, \tag{95}$$

where $\varepsilon = 0.01a^2$, and $f$ and $g$ are given by Eqs. (92) and (94).

To examine the accuracy for the Laplacian operator, $f$ is given by (92), and $\nabla^2 f$ is calculated from $f$ with the old DFS method (Cheong 2000a), the new DFS method (See Sect. 3.7) and the SH method. The calculated values are compared with the exact values of $\nabla^2 f$ in Eq. (94). Here, the exact values of $\nabla^2 f$ are truncated by the forward transform followed by the inverse transform in order to see the error that does not include the error due to inability to resolve at the resolution. Table 1 shows the normalized $L_2$ error between the calculated values and the exact values, which is normalized by the $L_2$ norm of the exact values. The differences in error between the methods are small, but the results of the SH method are a little better than the old and new DFS methods. Table 2 shows the global mean of calculated $\nabla^2 f$. The exact value of the global mean of $\nabla^2 f$ is zero. In Table 2, the global means calculated with each method are very close to zero. The global means of $\nabla^2 f$ in the DFS methods using Grid [1] and Grid [-1] are not as close to zero as those in the DFS methods using Grid [0] and the SH method. This seems to be because the accuracy of the meridional discrete cosine and sine transforms in the DFS methods using Grid [1] and Grid [-1] is not as good as that in the DFS methods using Grid [0].

To examine the accuracy of the solution of the Helmholtz equation, $h$ is given in Eq. (95) and the Helmholtz equation in Eq. (91) is solved with the old DFS method (Cheong 2000a), the new DFS method (See Sect. 3.8) and the SH method. The

calculated values are compared with the exact solution $f$ in Eq. (92). The exact values of $f$ are also truncated as described above. Table 3 shows the normalized $L_2$ error between the calculated values and the exact values. The differences in error between the methods are small, and which is better depends on the resolution and the arrangement of the grid points.

## 6 Evaluation of the DFS methods using shallow water test cases

We ran the Williamson test cases 1, 2, 5 and 6 (Williamson et al., 1992), and the Galewsky test case (Galewsky et al., 2004) in the model using the new DFS method described in Sect. 3, the model using the old DFS method of Yoshimura and Matsumura (2005), and the model using the SH method. By comparing the results of these model, we evaluated the old and new DFS methods.

### 6.1 Shallow water equations on a sphere

The prognostic equations of the shallow water model on a sphere are

$$\frac{d\boldsymbol{v}}{dt} = -2(\boldsymbol{\Omega} \times \boldsymbol{v})_{\mathrm{H}} - g\nabla h, \tag{96}$$

$$\frac{d(h - h_{\mathrm{s}})}{dt} = -(h - h_{\mathrm{s}})\nabla \cdot \boldsymbol{v}, \tag{97}$$

where $t$ is time, $\boldsymbol{v}$ is the horizontal wind vector, $h$ is the height, $h_{\mathrm{s}}$ is the surface height, $g$ is the acceleration due to gravity, $\boldsymbol{\Omega}$ is the 3-dimensional angular velocity of the earth's rotation, and the subscript H indicates the horizontal component.

Equation (96) is converted for the advective treatment of the Coriolis term (Temperton, 1997) into

$$\frac{d(\boldsymbol{v} + 2\boldsymbol{\Omega} \times \boldsymbol{r})}{dt} = -g\nabla h, \tag{98}$$

where $\boldsymbol{r}$ is the 3-dimensional position vector from the Earth's center. Equation (97) is converted for the spatially averaged Eulerian treatment of mountains (Ritchie and Tanguay, 1996) into

$$\frac{dh}{dt} = -(h - h_{\mathrm{s}})\nabla \cdot \boldsymbol{v} + \boldsymbol{v} \cdot \nabla h_{\mathrm{s}}. \tag{99}$$

Equations (98) and (99) are integrated in time using a two-time-level semi-implicit semi-Lagrangian scheme (See Appendix I).

### 6.2 Models

We ran the shallow water test cases in the semi-implicit semi-Lagrangian shallow water model or the Eulerian advection model (See Sect. 6.3) using the new DFS method (hereafter the new DFS model). We also ran the same test cases in the
model using the old DFS method of Yoshimura and Matsumura (2005) with the basis functions of Cheong (2000a, 2000b) (hereafter the old DFS model), and in the model using the SH method (hereafter the SH model) for comparison. The new DFS model was run for each of Grid [0], [1], and [−1]. In the old DFS model, Grid [0] is used. In the SH model, the

Gaussian grid is used. We use a regular longitude-latitude grid, not a reduced grid. We use the timestep $\Delta t = 3600$ s at about 300 km resolution with $J^0 = 64$, $\Delta t = 1800$ s at about 120 km resolution with $J^0 = 160$, $\Delta t = 1200$ s at about 60 km resolution with $J^0 = 320$, $\Delta t = 600$ s at about 20 km resolution with $J^0 = 960$, and $\Delta t = 90$ s at about 1.3 km resolution with $J^0 = 15360$, where $J^0$ is the number of latitudinal grid points in Grid [0]. The number of latitudinal grid points $J$ is $J^0$

in Grid [0] (and in the Gaussian grid), $J^0 + 1$ in Grid [1], and $J^0 - 1$ in Grid [−1] (See Sect. 2). The number of longitudinal grid points $I$ is $2J^0$. The meridional truncation wavenumber $N$ and the zonal wavenumber $M$ are set to be equal. In the Eulerian advection model, shorter timesteps are used as shown in Sect. 6.3. Horizontal diffusion is not used in all test cases. The zonal Fourier filter described in Appendix F is used in the DFS models. We have confirmed that numerical instability occurs in some test cases in the old DFS shallow water model without the zonal Fourier filter, but stable integration is

possible in all test cases shown here in the new DFS semi-Lagrangian shallow water model, even without the zonal Fourier filter. In the new DFS Eulerian advection model, the zonal Fourier filter is necessary (See Sect. 6.3).

     The zonal Fourier transforms in all the models and the meridional Fourier cosine and sine transforms in the DFS models are calculated using the Netlib BIHAR library, which includes a double precision version of the Netlib FFTPACK library (Swarztrauber, 1982). The meridional Legendre transform in the SH model is calculated using the ISPACK library (Ishioka,

2018), which adopts on-the-fly computation of the associated Legendre functions. We use the ISPACK library's optimization option for Intel AVX512, which is highly optimized by using assembly language together with Fortran.

## 6.3 Williamson test case 1

The Williamson test case 1 simulates a cosine-bell advection. In the semi-Lagrangian models, the advection is calculated in the semi-Lagrangian scheme and the horizontal derivatives calculated from the expansion coefficients are not used for the

advection calculation. Therefore, we also use the Eulerian scheme here to simulate the advection in the DFS and SH models to test the expansion methods. The advection equation is

$$\frac{dh}{dt} = \frac{\partial h}{\partial t} + \boldsymbol{v} \cdot \nabla h. \tag{100}$$

In the Eulerian models, the advection term $\boldsymbol{v} \cdot \nabla h$ is evaluated using the spectral transform method. The advection equation is integrated by the leap-frog scheme with the Robert-Asselin time filter (Robert, 1966; Asselin, 1972) with a coefficient of

0.05. The horizontal diffusion is not used, but the zonal Fourier filter is used in the old and new DFS methods. In Eq. (F1), the value $M_0 = 20$ is used in the DFS semi-Lagrangian shallow water models. However, the larger the value $M_0$ is, the higher the longitudinal resolution around the pole is. Because of this, when the Eulerian scheme is used and $M_0$ is large, a timestep must be very short due to the CFL condition. Therefore $M_0$ should be as small as possible. We have tested $M_0 = 0$, but this degrades the result of the Williamson test case 1. We have also tested $M_0 = 1$ and this result is good. Therefore, we

use $M_0 = 1$ in the Eulerian models.

     Figure 4 shows the predicted height after a 12-day integration in the Williamson test case 1 when using the Eulerian advection models at the resolution $J^0 = 64$. The meridional truncation wavenumber $N$ and the zonal truncation wavenumber

$M$ are set as $N = M = 42 \cong 2J^0/3$ because the 2/3 rule (Orszag, 1971) is used in order to avoid aliasing in the nonlinear advection term. The timestep is 30 minutes. The angle between the solid body rotation and the polar axis $\alpha$ is $\pi/2 - 0.05$. The results for DFS [0], DFS [1], DFS [−1] and SH are very similar. Instability occurs in the old DFS model without horizontal diffusion. This is probably because of the appearance of high-wavenumber oscillations due to the wavenumber truncation with $N \cong 2J^0/3$ for even $m$ ($\neq 0$) in the old DFS method, as shown in Sect. 4. Table 4 shows the normalized $L_2$ errors of the predicted height after a 12-day integration when using the Eulerian advection models. The timesteps are 30, 15, 7.5, and 2.5 minutes at the resolution $J^0 = 64, 160, 320$ and $960$ ($N = 42, 106, 213$ and $639$), respectively. The errors are very close between the models at each resolution. At the resolution $N = 639$, the new DFS model without horizontal diffusion is unstable when the timestep is 200 seconds. The SH model without horizontal diffusion is stable when the timestep is 240 seconds and unstable when the timestep is 300 seconds. One reason for this difference in timestep is probably that the longitudinal resolution near the poles is higher in the new DFS model with $M_0 = 1$ than in the SH model. When the fourth order horizontal diffusion in Eq. (86) with $\varepsilon = a^4 \Delta t/(7.2 \times 3600 \times 107^2 N^2)$ is used, the both new DFS and SH models are stable when the timestep is 240 seconds and are unstable when the timestep is 300 seconds. The old DFS model is unstable even when the same fourth order horizontal diffusion is used. Higher-order horizontal diffusion, which effectively smooths out the high wavenumber components, is necessary to stabilize the Eulerian old DFS model (Cheong, 2000b; Cheong et al., 2002).

Table 5 shows the same as Table 4 except for using the semi-Lagrangian scheme. In the semi-Lagrangian models, the forward transform followed by the inverse transform are executed at every timestep, but the expansion coefficients are not used for the advection calculation. The timesteps are the same as described in Sect. 6.2. The errors are very close between the models. At the resolution $J^0 = 64$, the errors in the semi-Lagrangian models are larger than those in the Eulerian models, but at the resolutions $J^0 = 160, 320$ and $960$, the errors in the semi-Lagrangian models are smaller than those in the Eulerian models.

The conservation of mass in the Williamson test case 1 was also examined, and the results are shown in Sect. S2 in the supplement.

## 6.4 Williamson test case 2

The Williamson test case 2 simulates a steady state non-linear zonal geostrophic flow. In this test case, the angle between the solid body rotation and the polar axis $\alpha$ is given, and the zonal and meridional components of $2\boldsymbol{\Omega} \times \boldsymbol{r}$ become

$$2\boldsymbol{\Omega} \times \boldsymbol{r} = (2\Omega a[\cos\theta \cos\alpha + \cos\lambda \sin\theta \sin\alpha], -2\Omega a \sin\lambda \sin\alpha). \tag{101}$$

Figure 5 shows the time series of forecast errors of the height for a 5-day integration in the Williamson test case 2 with $\alpha = \pi/2 - 0.05$ in the models at the resolution $J^0 = 64$ and $N = 63$ (DFS) or $N = 62$ (SH), using no horizontal diffusion. The normalized $L_1$, $L_2$, and $L_\infty$ errors are almost the same between the new DFS models using Grids [0], [1] and [−1], the

old DFS model, and the SH model. Table 6 shows the normalized $L_2$ errors of the predicted height after a 5-day integration. The errors are almost the same between the old DFS, new DFS and SH models at each resolution.

The conservation of mass, energy and vorticity in the Williamson test cases 2, 5 and 6 was also examined, and the results are shown in Sect. S2 in the supplement.

## 6.5 Williamson test case 5

The Williamson test case 5 simulates zonal flow over an isolated mountain. Figure 6 shows the predicted height after a 15-day integration in Williamson test case 5 with $h_0 = 5960$ m. The result of the high-resolution SH model at the resolution $J = 960$ and $N = 958$ is regarded as the reference solution. Horizontal diffusion is not used. The errors with respect to the reference solution are almost the same for the new DFS models, the old DFS model, and the SH model at the resolution $J^0 = 64$. Table 7 shows the normalized $L_2$ errors of the predicted height after a 15-day integration. The errors are almost the same between the old DFS, new DFS and SH models at each resolution. The errors do not decrease when the resolution increases, which is different from the results in the other test cases. This may be because the mountain topography is not a differentiable function, and the mountain is added impulsively on to an initially balanced flow (Galewsky et al. 2004).

Figure 7 shows the longitudinal distributions of meridional wind at the grid points near the South Pole after a 15-day integration in the old and new DFS models using Grid [0] at the resolutions $J^0 = 64$ and $J^0 = 960$. While the zonal wavenumber 1 component is dominant in the new DFS model at the resolution $J^0 = 64$ and $N = 63$, high zonal wavenumber noise appears in the old DFS model at the same resolution. One possible reason is that the latitudinal derivative at the grid points can be unrealistically large in the old DFS method even when $N' = J^0$ for even $m$ ($\geq 2$) as described in Sect. 4 (Fig. 3b). The new DFS expansion method with the least-squares method does not have this problem. By using the new expansion method with the least-squares method, the high zonal wavenumber noise does not appear even in the model that uses the same DFS basis functions as in Eq. (11) except that the basis function for odd $m$ ($\geq 3$) is $\sin\theta \cos n\theta$ instead of $\sin^2\theta \sin n\theta$. In the old DFS model at high resolution with $J^0 = 960$ and $N = 959$, the high wavenumber noise is not seen in Fig. 7. The higher the resolution, the smaller the high wavenumber noise becomes.

Figure 8 shows the kinetic energy spectra of the horizontal winds (Lambert, 1984) after a 15-day integration in Williamson test case 5. The kinetic energy spectra in the DFS models are calculated from the SH expansion coefficients, which are obtained by firstly calculating the Gaussian grid point values from the DFS coefficients using Eq. (10) for the new DFS method and Eq. (6) for the old DFS method, and secondly calculating the SH expansion coefficients from the Gaussian grid point values by using a forward Legendre transform. In the old DFS model with $J^0 = 64$ and $N = 63$, the high wavenumber components are larger than in the other models, which is related to the high wavenumber noise near the South Pole in Fig. 7. In the old DFS model with $J^0 = 960$, the high wavenumber components are a little larger than in the other models, but the differences are slight.

Figure 9 shows the predicted height after a 15-day integration in Williamson test case 5, which is the same as Fig. 6 except for the truncation wavenumber $N = 42 \cong 2J^0/3$. In our semi-implicit semi-Lagrangian models, we usually use $N$ satisfying $N \cong J^0 - 1$, which is called linear truncation. However, here $N$ is determined to satisfy $N \cong 2J^0/3$ to eliminate aliasing errors with quadratic nonlinearity, which is called quadratic truncation. When using the quadratic truncation, the new DFS models with Grids [0], [1], and [−1] are stable without horizontal diffusion, but the old DFS model without strong high-order horizontal diffusion is unstable. The numerical instability in the old DFS model probably occurs because of the high-wavenumber oscillations due to the quadratic wavenumber truncation for even $m$ ($\neq 0$) (See Sect. 4) as in the Williamson test case 1 with the Eulerian model. The results of the new DFS models are almost the same as those of the SH model. Table 8 is the same as Table 7 except for $N \cong 2J^0/3$. The results of the new DFS models and the SH model with $N \cong 2J^0/3$ in Table 8 are very similar to those with $N \cong J^0 - 1$ in Table 7 when $J^0$ is the same.

Figure 10 shows the kinetic energy spectrum of the horizontal winds after a 15-day integration in Williamson test case 5, which is the same as Fig. 8 except for the truncation wavenumber $N \cong 2J^0/3$. At the resolution $J^0 = 64$ and $N = 42$, the high wavenumber components are a little larger in the SH model than in the new DFS model. At the resolution $J^0 = 960$ and $N = 639$, very small oscillations appear in the high wavenumber region in the SH model, but not in the new DFS models. In the SH model, the wind components $u$ and $v$ divided by $\sin\theta$ are transformed from grid space to spectral space (Ritchie, 1988; Temperton, 1991), which seems to reduce the accuracy and cause the small oscillations in the high wavenumber region. Another way to transform $u$ and $v$ from grid space to spectral space in the SH model is to use the vector harmonic transform (see Sect. 3.6). This way is algebraically equivalent to the way dividing $u$ and $v$ by $\sin\theta$ (Temperton, 1991), but avoids dividing $u$ and $v$ by $\sin\theta$ and provides the remarkable stability and accuracy (Swarztrauber, 2004). This way is similar to the new DFS expansion method for $u$ and $v$ using the least-squares method described in Sect. 3.6, and probably eliminates the small oscillations in the SH model. Alternatively, using $D$ and $\zeta$ instead of $u$ and $v$ as prognostic variables may eliminate the small oscillations.

**6.6 Williamson test case 6**

Figure 11 shows the predicted height after a 14-day integration in Williamson test case 6. The error is similar between the old and new DFS models using Grid [0] and the SH model. The error in the new DFS model using Grid [1] is the smallest. This is probably because Grid [1] has grid points at the poles, where the minimum height exists, and on the equator, where the maximum height exists. The error in the new DFS model using Grid [−1] is the second smallest. This is probably because Grid [−1] has grid points on the equator. Table 9 shows the normalized $L_2$ errors of the predicted height after a 14-day integration. The error in the new DFS model using Grid [1] is the smallest, and that in the new DFS model using Grid [−1] is the second smallest, at each resolution. The errors in the old and new DFS models using Grid [0] and in the SH model are very close.

### 6.7 Galewsky test case

The Galewsky test case simulates a barotropically unstable mid-latitude jet. Figure 12 shows the predicted vorticity after a 6-day integration in the Galewsky test case for the models at 1.3 km resolution with $J^0 = 15360$ and the quadratic truncation $N = 10239$, without horizontal diffusion. The result in the new DFS model using Grid [0] is almost the same as in the SH model. The old DFS model is unstable for the same reason as that shown in Sect. 6.5 (Fig. 9). Figure 13 shows the kinetic energy spectrum of horizontal winds after a 6-day integration in the Galewsky test case. The results are almost the same for the DFS models using Grid [0], [1] and [−1], and the SH model, but very small oscillations appear near the truncation wavenumber in the SH model. This is probably for the same reason as in Williamson test case 5 in Fig. 10.

The results of the Galewsky-like test case using the north-south symmetric initial conditions are shown in Sect. S3 in the supplement.

### 6.8 Elapsed time

Figure 14 shows the elapsed time for the 15-day integration in the Williamson test case 5 in the SH model and the new DFS model using Grid [0] at 20 km resolution with $J^0 = 960$ and $N = 958$ (SH) or $N = 959$ (DFS), and that for the 6-day integration in the Galewsky test case at 1.3 km resolution with $J^0 = 15360$ and $N = 10239$. We use one node (with two Intel Xeon Gold 6248 CPUs with 20 cores per CPU) of the FUJITSU Server PRIMERGY CX2550 M5 in the MRI. The source code written in Fortran is compiled with the Intel compiler. OpenMP parallelization is used, but MPI parallelization is not used. The elapsed time in the SH model is larger than in the DFS model, although the Legendre transform used in the SH model is highly optimized for Intel AVX512. The higher the resolution, the larger is the difference of the elapsed time between the models. This is because the Legendre transform used in the SH model requires $O(N^3)$ operations while the Fourier cosine and sine transforms used in the DFS model require only $O(N^2 \log N)$ operations. If the fast Legendre transform, which requires only $N^2 (\log N)^3$ operation, is used instead of the usual Legendre transform in the SH model, the difference of the elapsed time between the models will be reduced at high resolutions. We have not tested the fast Legendre transform yet because we do not have subroutines for the fast Legendre transform.

### 7 Conclusions and perspectives

We have developed the new DFS method to improve the numerical stability of the DFS model, which has the following two improvements:

1. A new expansion method with the least-squares method is used to calculate the expansion coefficients so that the error due to the meridional wavenumber truncation is minimized. The method also avoids dividing by $\sin \theta$ before taking the forward Fourier cosine or sine transform.

2. New DFS basis functions are used, which guarantees that not only scalar variables, but also vector variables and the gradient of scalar variables, are continuous at the poles.

The equations obtained with the least-squares method are equivalent to those obtained with the Galerkin method. We also use the Galerkin method to solve partial differential equations such as the Poisson equation, the Helmholtz equation, and the shallow water equations.

To test the new DFS method, we conducted experiments for the Williamson test cases 2, 5 and 6, and the Galewsky test case in the semi-implicit semi-Lagrangian shallow water models using the new DFS method with the three types of equally spaced latitudinal grids with or without the poles. We also ran the Williamson test case 1, which simulates a cosine-bell advection, in the Eulerian and semi-Lagrangian advection models. We compared the results between the new DFS models using the new DFS method, the old DFS model using the method of Yoshimura and Matsumura (2005) with the basis functions of Cheong (2000a, 2000b), and the SH model.

The high zonal wavenumber noise of the meridional wind appears near the poles in the old DFS model, but not in the new DFS models in the Williamson test case 5. One possible reason is that the latitudinal derivative at the grid points can be unrealistically large in the old DFS method even when the truncation wavenumber $N'$ for even $m$ ($\neq 0$) is equal to the number of latitudinal grid points $J$, while the new DFS expansion method with the least-squares method does not have this problem. In the old DFS model, $N' < J$ for even $m$ ($\neq 0$) causes numerical instability. In the new DFS model, an arbitrary meridional wavenumber truncation $N$ ($< J$) can be used without the stability problem because the error due to meridional wavenumber truncation is small when using the new DFS expansion method with the least-squares method. This is one of the merits of the new DFS method because the quadratic truncation ($N \cong 2J/3$) or the cubic truncation ($N \cong J/2$) is usually used in the Eulerian model and is also becoming to be used in the semi-Lagrangian model instead of the linear truncation ($N \cong J - 1$) for stability and efficiency at high resolutions (Wedi, 2014; Hotta and Ujiie, 2018; Dueben et al., 2020). We have also confirmed that in the new DFS model, stable integration is possible in all test cases shown here even without using the zonal Fourier filter unlike in the old DFS model. Thus, the numerical stability of the semi-implicit semi-Lagrangian model using the new DFS method is very good. In the Williamson test cases 1, the Eulerian advection model using the new DFS method also gives stable results without horizontal diffusion but with a zonal Fourier filter. The Eulerian advection model using the old DFS method is unstable without horizontal diffusion or with the weak fourth-order horizontal diffusion. In the old DFS model, the use of the semi-Lagrangian scheme is important for numerical stability. On the other hand, the advection model using the new DFS method is stable, even when the Eulerian scheme is used instead of the semi-Lagrangian scheme.

The results of the new DFS model are almost the same as the SH model. But in the SH shallow water model without horizontal diffusion, very small oscillations appear in the high wavenumber region of the kinetic energy spectrum in some cases, unlike in the new DFS model. This seems to be because the wind components $u$ and $v$ divided by $\sin\theta$ are transformed from grid space to spectral space in the SH model. The small oscillations with the SH model can probably be eliminated by using the vector harmonic transform, which is similar to the new DFS expansion method for $u$ and $v$ using the

least-squares method and avoids dividing $u$ and $v$ by $\sin\theta$. Alternatively, using divergence and vorticity instead of $u$ and $v$ as prognostic variables may eliminate the small oscillations.

The elapsed time in the new DFS model is shorter than in the SH model especially at high resolution because the Fourier transform requires only $O(N^2 \log N)$ operations, and the Legendre transform in the SH model requires $O(N^3)$ operations. We have executed our shallow water models on Intel CPUs. The execution on GPUs is one important topic, but we have not tested our models on GPUs because the execution on GPUs is not an easy task. MPI parallelization is another important topic. However, in our shallow water models, we use only OpenMP parallelization, not MPI parallelization for the simplicity of the source code.

We developed hydrostatic and nonhydrostatic global atmospheric models using the old DFS method (Yoshimura and Matsumura, 2005; Yoshimura, 2012) and conducted typhoon prediction experiments in the nonhydrostatic global atmospheric model using the old DFS method in the Global 7 km mesh nonhydrostatic Model Intercomparison Project for improving TYphoon forecast (TYMIP-G7; Nakano et al., 2017). We have already developed a nonhydrostatic (or hydrostatic) atmospheric model using the new DFS method, where both OpenMP and MPI parallelization are used. We will describe the nonhydrostatic DFS model and the MPI parallelization in another paper after improving the nonhydrostatic dynamical core as needed.

Supplement. The supplement related to this article is available online at ...

Code availability. The source code of the DFS and SH shallow water models is available in the supplement to the article and is licensed under a Creative Commons Attribution-NonCommercial-ShareAlike 4.0 International (CC BY-NC-SA 4.0) license. These models utilize the Netlib BIHAR library and the ISPACK library. The Netlib BIHAR library is available at https://www.netlib.org/bihar/ and is also included in the supplement. The ISPACK library is available at https://www.gfd-dennou.org/arch/ispack/.

Data availability. The results of model experiments are available at https://climate.mri-jma.go.jp/pub/archives/Yoshimura_DFS_SW_Testcase/.

## Appendix A: Trigonometric identities

We list here the trigonometric identities used in transforming the expressions in this paper. The following identities are satisfied:

$$\sin n\theta \cos n'\theta = \frac{1}{2}[\sin(n+n')\theta + \sin(n-n')\theta], \qquad \text{(A1a)}$$

$$\cos n\theta \sin n'\theta = \frac{1}{2}[\sin(n+n')\theta - \sin(n-n')\theta], \tag{A1b}$$

$$\cos n\theta \cos n'\theta = \frac{1}{2}[\cos(n+n')\theta + \cos(n-n')\theta], \tag{A1c}$$

$$\sin n\theta \sin n'\theta = \frac{1}{2}[-\cos(n+n')\theta + \cos(n-n')\theta]. \tag{A1d}$$

From Eq. (A1), the following identities are derived:

$$\sin\theta \cos n\theta = \frac{1}{2}[\sin(n+1)\theta - \sin(n-1)\theta], \tag{A2a}$$

$$\sin\theta \sin n\theta = \frac{1}{2}[-\cos(n+1)\theta + \cos(n-1)\theta], \tag{A2b}$$

$$\sin^2\theta \sin n\theta = \frac{1}{4}[-\sin(n-2)\theta + 2\sin n\theta - \sin(n+2)\theta], \tag{A2c}$$

$$\sin^2\theta \cos n\theta = \frac{1}{4}[-\cos(n-2)\theta + 2\cos n\theta - \cos(n+2)\theta], \tag{A2d}$$

$$\sin^4\theta \sin n\theta = \frac{1}{16}[\sin(n-4)\theta - 4\sin(n-2)\theta + 6\sin n\theta - 4\sin(n+2)\theta + \sin(n+4)\theta]. \tag{A2e}$$

From Eq. (A1), the following orthogonal relations in longitude are derived:

$$\int_0^{2\pi} \cos m\lambda \cos m'\lambda \, d\lambda = \begin{cases} 2\pi & \text{for } m = m' = 0, \\ \pi & \text{for } m = m' \neq 0, \\ 0 & \text{for } m \neq m', \end{cases} \tag{A3a}$$

$$\int_0^{2\pi} \cos m\lambda \sin m'\lambda \, d\lambda = 0, \tag{A3b}$$

$$\int_0^{2\pi} \sin m\lambda \sin m'\lambda \, d\lambda = \begin{cases} \pi & \text{for } m = m' \neq 0, \\ 0 & \text{for } m \neq m'. \end{cases} \tag{A3c}$$

Similarly, from Eq. (A1), the following orthogonal relations in latitude are derived:

$$\int_0^{\pi} \cos n\theta \cos n'\theta \, d\theta = \begin{cases} \pi & \text{for } n = n' = 0, \\ \frac{1}{2}\pi & \text{for } n = n' \neq 0, \\ 0 & \text{for } n \neq n', \end{cases} \tag{A4a}$$

$$\int_0^{\pi} \sin n\theta \sin n'\theta \, d\theta = \begin{cases} \frac{1}{2}\pi & \text{for } n = n' \neq 0, \\ 0 & \text{for } n \neq n'. \end{cases} \tag{A4b}$$

By using Eqs. (A1) and (A2), the following relations are derived:

$$\frac{\partial}{\partial\theta}(\sin^l\theta \cos n\theta) = \frac{n+l}{2}\sin^{l-1}\theta \cos(n+1)\theta - \frac{n-l}{2}\sin^{l-1}\theta \cos(n-1)\theta, \tag{A5a}$$

$$\sin\theta \frac{\partial}{\partial\theta}\left[\sin\theta \frac{\partial}{\partial\theta}(\sin^l\theta \cos n\theta)\right] = \frac{(n+l)(n+l+1)}{4}\sin^l\theta \cos(n+2)\theta$$
$$-\frac{2n^2 - 2l^2 + 2l}{4}\sin^l\theta \cos n\theta + \frac{(n-l)(n-l-1)}{4}\sin^l\theta \cos(n-2)\theta, \tag{A5b}$$

$$\frac{\partial}{\partial \theta}(\sin^l \theta \sin n\theta) = \frac{n+l}{2}\sin^{l-1}\theta\sin(n+1)\theta - \frac{n-l}{2}\sin^{l-1}\theta\sin(n-1)\theta, \tag{A5c}$$

$$\sin\theta\frac{\partial}{\partial\theta}\left[\sin\theta\frac{\partial}{\partial\theta}(\sin^l\theta\sin n\theta)\right] = \frac{(n+l)(n+l+1)}{4}\sin^l\theta\sin(n+2)\theta$$

$$-\frac{2n^2-2l^2+2l}{4}\sin^l\theta\sin n\theta + \frac{(n-l)(n-l-1)}{4}\sin^l\theta\sin(n-2)\theta, \tag{A5d}$$

**Appendix B: Discrete Fourier cosine and sine transforms in latitude**

5 Forward discrete Fourier cosine and sine transforms are performed in Eqs. (23) and (57), and inverse discrete Fourier cosine and sine transforms are performed in Eqs. (13), (52), in the latitudinal direction. The calculation of the discrete cosine and sine transforms in Grids [0], [1], and [−1] is shown below. Here, $g(\theta_j)$ and $h(\theta_j)$ are grid-point values, $\theta_j$ is the colatitude defined in Eq. (7), and $g_n$ and $h_n$ are expansion coefficients.

When using Grid [0], inverse and forward discrete cosine transforms are performed as

$$g(\theta_j) = \sum_{n=0}^{J^0-1} g_n \cos n\theta_j, \tag{B1a}$$

$$g_n = \frac{b}{J^0}\sum_{j=0}^{J^0-1} g(\theta_j)\cos n\theta_j, \quad b \equiv \begin{cases} 1 & \text{for } n=0 \\ 2 & \text{for } 1 \le n \le J^0-1. \end{cases} \tag{B1b}$$

When using Grid [0], inverse and forward discrete sine transforms are performed as

$$h(\theta_j) = \sum_{n=1}^{J^0} h_n \sin n\theta_j, \tag{B2a}$$

$$h_n = \frac{b}{J^0}\sum_{j=0}^{J^0-1} h(\theta_j)\sin n\theta_j, \quad b \equiv \begin{cases} 1 & \text{for } n=J^0 \\ 2 & \text{for } 1 \le n \le J^0-1. \end{cases} \tag{B2b}$$

15 When using Grid [1], inverse and forward discrete cosine transforms are performed as

$$g(\theta_j) = \sum_{n=0}^{J^0} g_n \cos n\theta_j, \tag{B3a}$$

$$g_n = \frac{b}{J^0}\sum_{j=0}^{J^0} c\, g(\theta_j)\cos n\theta_j,$$

$$b \equiv \begin{cases} 1 & \text{for } n=0,J^0 \\ 2 & \text{for } 1 \le n \le J^0-1, \end{cases} \quad c \equiv \begin{cases} 1/2 & \text{for } j=0,J^0 \\ 1 & \text{for } 1 \le j \le J^0-1. \end{cases} \tag{B3b}$$

When using Grid [1], inverse and forward discrete sine transforms are performed as

$$h(\theta_j) = \sum_{n=1}^{J^0-1} h_n \sin n\theta_j, \quad h(\theta_0) = h(\theta_{J^0}) = 0, \tag{B4a}$$

$$h_n = \frac{2}{J^0} \sum_{j=1}^{J^0-1} h(\theta_j) \sin n\theta_j \quad (1 \le n \le J^0 - 1). \tag{B4b}$$

Grid $[-1]$ is the same as Grid $[1]$, except that there are no grid points at the North and South poles. The zonal wavenumber components of scalar variables at the poles are zero except for $m = 0$ (See Eqs. (10) and (11)), and those of vector variables at the poles are zero except for $m = 1$ (See Eq. (52)). When we use Grid $[-1]$ and the values at the poles are known to be zero, forward and inverse discrete cosine transforms can be performed using Eq. (B3) and forward and inverse discrete sine transforms can be performed using Eq. (B4) in the same way as for Grid $[1]$. When we use Grid $[-1]$ and the values at the poles are unknown (i.e., the zonal wavenumber components of scalar variables for $m = 0$, and those of vector variables for $m = 1$), the inverse discrete cosine transform can be performed like Eq. (B3a) as

$$g(\theta_j) = \sum_{n=0}^{J^0-2} g_n \cos n\theta_j, \tag{B5}$$

where $n$ is from $0$ to $J^0 - 2 (= J - 1)$ because the number of the meridional grid points is $J^0 - 1 (= J)$ in Grid $[-1]$. However, the forward discrete cosine transform cannot be performed like Eq. (B3b). We can calculate the expansion coefficients $g_n$ from $g(\theta_j)$ in the following way. Eq. (B5) is multiplied by $\sin \theta_j$, and we define $\hat{g}(\theta_j)$ as

$$\hat{g}(\theta_j) \equiv g(\theta_j) \sin \theta_j = \sum_{n=0}^{J^0-2} g_n \sin \theta_j \cos n\theta_j. \tag{B6}$$

We can expand $\hat{g}(\theta_j)$ as

$$\hat{g}(\theta_j) = \sum_{n=1}^{J^0-1} \hat{g}_n \sin n\theta_j. \tag{B7}$$

The expansion coefficients $\hat{g}_n$ can be obtained from $\hat{g}(\theta_j)$ in the same way as in Eq. (B4b) by forward discrete sine transform:

$$\hat{g}_n = \frac{2}{J^0} \sum_{j=1}^{J^0-1} \hat{g}(\theta_j) \sin n\theta_j. \tag{B8}$$

From Eqs. (B6) and (B7), we obtain

$$\sum_{n=0}^{J^0-2} g_n \sin \theta \cos n\theta = \sum_{n=1}^{J^0-1} \hat{g}_n \sin n\theta, \tag{B9}$$

By using Eq. (A2a), we obtain

$$\sum_{n=0}^{J^0-2} g_n \sin\theta \cos n\theta = \left(g_0 - \frac{g_2}{2}\right)\sin\theta + \sum_{n=2}^{J^0-3}\left(\frac{g_{n-1}}{2} - \frac{g_{n+1}}{2}\right)\sin n\theta + \frac{g_{J^0-3}}{2}\sin(J^0-2)\theta + \frac{g_{J^0-2}}{2}\sin(J^0-1)\theta. \quad \text{(B10)}$$

By substituting Eq. (B10) into Eq. (B9) and comparing the left and right sides of the equation, we obtain

$$\hat{g}_n = \begin{cases} g_0 - \dfrac{g_2}{2} & \text{for } n = 1, \\ \dfrac{g_{n-1}}{2} - \dfrac{g_{n+1}}{2} & \text{for } 2 \le n \le J^0 - 3, \\ \dfrac{g_{J^0-3}}{2} & \text{for } n = J^0 - 2, \\ \dfrac{g_{J^0-2}}{2} & \text{for } n = J^0 - 1. \end{cases} \quad \text{(B11)}$$

We can calculate $\hat{g}(\theta_j)$ from $g(\theta_j)$ using Eq. (B6), calculate $\hat{g}_n$ from $\hat{g}(\theta_j)$ using Eq. (B8), and calculate $g_n$ from $\hat{g}_n$ using

Eq. (B11).

## Appendix C: The upper limit of the meridional truncation wavenumber $N$

In the new DFS method, the meridional truncation wavenumber $N$ is used for the new DFS meridional basis functions in Eq. (12), and for the discrete cosine or sine transform of a scalar variable (Eqs. (13) and (23)), derivatives of a scalar variable (Eqs. (18) and (20)) and a wind vector (Eqs. (52) and (57)). In Grid [0], the upper limit of $N$ is $J^0 - 1$ for each $m$ because the

discrete cosine transform in Eq. (B1), where the maximum value of $n$ is $J^0 - 1$, is used for a scalar variable when $m$ is even, and for vector components when $m$ is odd. In Grid [1], the upper limit of $N$ is $J^0 - 1$ for each $m$ because the discrete sine transform in Eq. (B4), where the maximum value of $n$ is $J^0 - 1$, is used for a scalar variable when $m$ is odd, and for vector components when $m$ is even. In Grid [$-1$], the upper limit of $N$ is $J^0 - 1$ for $m \ge 2$ because of the same reason as in Grid [1]. However, for $m = 0$ or 1 in Grid [$-1$], the upper limit of $N$ is $J^0 - 2$ because the discrete cosine transform in Eq. (B5),

where the maximum value of $n$ is $J^0 - 2$, is used for a scalar variable when $m = 0$, and for vector components when $m = 1$. Thus, the upper limit of $N$ is $J^0 - 1$, except that the upper limit of $N$ for $m = 0$ or 1 in Grid [$-1$] is $J^0 - 2$. For example, in the model using the new DFS method with Grid [$-1$] at the resolution $J^0 = 64$ and $N = 63$, we set $N = 63$ for $m \ge 2$ but $N = 62$ for $m = 0$ or 1.

## Appendix D: Equations for the derivation of Eqs. (29) and (62)

$\tilde{T}_{n,m}^{c(s)}$ in Eq. (23) is calculated by the forward Fourier cosine or sine transform as

$$\tilde{T}_{n,m}^{c(s)} = \begin{cases} \dfrac{b}{\pi}\displaystyle\int_0^\pi \cos n\theta\, T_m^{c(s)}(\theta)d\theta\,, \quad b \equiv \begin{cases} 1 & \text{for } n = 0 \\ 2 & \text{for } n \ne 0, \end{cases} & \text{for even } m, \\ \dfrac{2}{\pi}\displaystyle\int_0^\pi \sin n\theta\, T_m^{c(s)}(\theta)d\theta & \text{for odd } m. \end{cases} \quad \text{(D1)}$$

The equations for the forward discrete Fourier cosine or sine transform are described in Appendix B. From Eq. (23) and (A4),

$$\frac{b}{\pi} \int_0^\pi \cos n\theta \, \tilde{T}_m^{c(s),N}(\theta) d\theta = \tilde{T}_{n,m}^{c(s)} \quad (n = 0, \dots, N) \text{ for even } m, \tag{D2a}$$

$$\frac{2}{\pi} \int_0^\pi \sin n\theta \, \tilde{T}_m^{c(s),N}(\theta) d\theta = \tilde{T}_{n,m}^{c(s)} \quad (n = 1, \dots, N) \text{ for odd } m \tag{D2b}$$

are also derived. From Eqs. (D1) and (D2),

$$\int_0^\pi \cos n\theta \, T_m^{c(s)}(\theta) d\theta = \int_0^\pi \cos n\theta \, \tilde{T}_m^{c(s),N}(\theta) d\theta \quad (n = 0, \dots, N) \text{ for even } m, \tag{D3a}$$

$$\int_0^\pi \sin n\theta \, T_m^{c(s)}(\theta) d\theta = \int_0^\pi \sin n\theta \, \tilde{T}_m^{c(s),N}(\theta) d\theta \quad (n = 1, \dots, N) \text{ for odd } m \tag{D3b}$$

are satisfied. From Eqs. (D3), (11), (12) and (A2a–c), we derive

$$\int_0^\pi S_{n,m}(\theta) T_m^{c(s)}(\theta) d\theta = \int_0^\pi S_{n,m}(\theta) \tilde{T}_m^{c(s),N}(\theta) d\theta \quad (n = N_{\min,m}, \dots, N_{\max,m}). \tag{D4}$$

From Eqs. (28) and (D4), we derive Eq. (29).

We can also derive the following equations from Eq. (57) in the similar way to the derivation of (D3):

$$\int_0^\pi \sin n\theta \, u_m^{c(s)}(\theta) d\theta = \int_0^\pi \sin n\theta \, \tilde{u}_m^{c(s),N}(\theta) d\theta \quad (n = 1, \dots, N) \text{ for even } m, \tag{D5a}$$

$$\int_0^\pi \cos n\theta \, u_m^{c(s)}(\theta) d\theta = \int_0^\pi \cos n\theta \, \tilde{u}_m^{c(s),N}(\theta) d\theta \quad (n = 0, \dots, N) \text{ for odd } m. \tag{D5b}$$

From Eqs. (D5), (11), (12), and (A2a–c), we derive

$$\int_0^\pi \frac{m S_{n,m}(\theta)}{\sin \theta} u_m^{c(s)}(\theta) d\theta = \int_0^\pi \frac{m S_{n,m}(\theta)}{\sin \theta} \tilde{u}_m^{c(s),N}(\theta) d\theta \quad (n = N_{\min,m}, \dots, N_{\max,m}), \tag{D6a}$$

$$\int_0^\pi \frac{\partial S_{n,m}(\theta)}{\partial \theta} u_m^{c(s)}(\theta) d\theta = \int_0^\pi \frac{\partial S_{n,m}(\theta)}{\partial \theta} \tilde{u}_m^{c(s),N}(\theta) d\theta \quad (n = N_{\min,m}, \dots, N_{\max,m}). \tag{D6b}$$

We can also derive the same equations as Eq. (D6) except that $u$ is replaced with $v$. Equation (D6) are used to derive Eq. (62).

**Appendix E: Derivation of Eq. (30) from Eq. (29)**

Here we derive Eq. (30d) for odd ($m \geq 3$) from Eq. (29). Eqs. (30b,c) can also be derived similarly. By using Eqs. (10), (11), (23) and (A2c,e), Eq. (29) is converted as follows:

$$(\text{l. h. s of Eq. (29) for odd } m \geq 3) = \int_0^\pi S_{n,m}(\theta) T_m^{c,N}(\theta) d\theta = \int_0^\pi \sin^2 \theta \sin n\theta \sum_{n'=1}^{N-2} T_{n',m}^c \sin^2 \theta \sin n'\theta \, d\theta$$

$$= \int_0^\pi \sin n\theta \sum_{n'=1}^{N-2} \frac{T_{n',m}^c}{16} [\sin(n'-4)\theta - 4\sin(n'-2)\theta + 6\sin n'\theta - 4\sin(n'+2)\theta + \sin(n'+4)\theta] \, d\theta$$

$$= \int_0^\pi \sin n\theta \left[ \frac{10T_{1,m}^c - 5T_{3,m}^c + T_{5,m}^c}{16} \sin\theta + \frac{5T_{2,m}^c - 4T_{4,m}^c + T_{6,m}^c}{16} \sin 2\theta + \frac{-5T_{1,m}^c + 6T_{3,m}^c - 4T_{5,m}^c + T_{7,m}^c}{16} \sin 3\theta \right.$$

$$\left. + \sum_{n'=4}^{N+2} \frac{T_{n'-4,m}^c - 4T_{n'-2,m}^c + 6T_{n',m}^c - 4T_{n'+2,m}^c + T_{n'+4,m}^c}{16} \sin n'\theta \right] d\theta, \tag{E1}$$

$$(\text{r.h.s of Eq. (29) for odd } m \geq 3) = \int_0^\pi S_{n,m}(\theta) \tilde{T}_m^{c(s),N}(\theta) d\theta = \int_0^\pi \sin^2\theta \sin n\theta \sum_{n'=1}^{N} \tilde{T}_{n',m}^c \sin n'\theta \, d\theta$$

$$= \int_0^\pi \sin n\theta \sum_{n'=1}^{N} \frac{\tilde{T}_{n',m}^c}{4} \left[ -\sin(n'-2)\theta + 2\sin n'\theta - \sin(n'+2)\theta \right] d\theta$$

$$= \int_0^\pi \sin n\theta \left[ \frac{3\tilde{T}_{1,m}^c - \tilde{T}_{3,m}^c}{4} \sin\theta + \frac{2\tilde{T}_{2,m}^c - \tilde{T}_{4,m}^c}{4} \sin 2\theta + \sum_{n'=3}^{N+2} \frac{-\tilde{T}_{n'-2,m}^c + 2\tilde{T}_{n',m}^c - \tilde{T}_{n'+2,m}^c}{4} \sin n'\theta \right] d\theta, \tag{E2}$$

where $1 \leq n \leq N - 2$. From Eqs. (29), (E1), (E2) and (A4b), Eq. (30d) are derived.

**Appendix F: Zonal Fourier filter**

In a regular longitude–latitude grid, the longitudinal grid spacing becomes narrow at high latitudes. In DFS methods, the zonal Fourier filter (Merilees 1974; Boer and Steinberg 1975; Cheong 2000a), which filters out the high zonal wavenumber components at high latitudes, is usually used to obtain a more uniform resolution. The use of a reduced grid (Hortal and Simmons, 1991; Juang, 2004; Miyamoto, 2006; Malardel, 2016) has a similar effect to the zonal Fourier filter. In our atmospheric model using the old DFS method (Yoshimura, 2012), we use the reduced grid of Miyamoto (2006).

In this study, we use the regular longitude–latitude grid with the zonal Fourier filter, not the reduced grid, for the simplicity of the source code. We set the largest zonal wavenumber $M_f$ at each colatitude $\theta_j$ as

$$M_f(\theta_j) = \min(M, M_0 + M\sin(\theta_j)). \tag{F1}$$

The values of $T_m^c(\theta_j)$ and $T_m^s(\theta_j)$ in Eq. (8) are set to zero for $m > M_f(\theta_j)$ during the spectral transform. We use the value $M_0 = 20$ in the DFS shallow water model to make the resolution similar to that in the reduced grid of Miyamoto (2006). In the DFS Eulerian advection model, we use the value $M_0 = 1$ as described in Sect. 6.3.

**Appendix G: Calculation of global mean and latitudinal area weight**

The global mean value of $T^{N,M}(\lambda, \theta)$ in Eq. (15) can be calculated in spectral space by the following equation (Cheong 2000a):

$$G = \frac{1}{4\pi} \int_0^{2\pi} \int_0^\pi \left( \sum_{m=0}^{M} T_m^{c,N}(\theta) \cos m\lambda + \sum_{m=1}^{M} T_m^{s,N}(\theta) \sin m\lambda \right) \sin\theta \, d\theta d\lambda$$

$$= \frac{1}{2} \int_0^\pi \sum_{n=0}^N T_{n,m=0}^c \cos n\theta \sin\theta \, d\theta = \sum_{\substack{n=0 \\ \text{when } n \text{ is even}}}^N \frac{T_{n,m=0}^c}{1-n^2}, \tag{G1}$$

where Eq. (A2a) is used.

The latitudinal area weight at each colatitude $\theta_j$ is calculated as follows:

1. The latitudinal distribution of $T_{m=0}^{c\,(j)}(\theta_k)$ for each $j$ is given as

$$T_{m=0}^{c\,(j)}(\theta_k) = \begin{cases} 1 & \text{for } k = j \\ 0 & \text{for } k \neq j, \end{cases} \tag{G2}$$

where $0 \leq k \leq J - 1$ in Grid [0] and Grid [1], and $1 \leq k \leq J$ in Grid [-1] (See Sect. 2).

2. From $T_{m=0}^{c\,(j)}(\theta_k)$, the meridional expansion coefficients $T_{n,m=0}^{c\,(j)}$ ($0 \leq n \leq N$) are calculated by forward discrete cosine transform described in Appendix B.

3. The value of $G$ calculated from $T_{n,m=0}^{c\,(j)}$ using Eq. (G1) is considered as the latitudinal area weight $w(\theta_j)$ at colatitude $\theta_j$.

In Grid [0] and Grid[1], the distribution of $w(\theta_j)$ is smooth. However, in Grid [−1], the distribution of $w(\theta_j)$ is not smooth because of the irregularity with Grid [−1] (See Eqs. (B5)–(B11) in Appendix B).

The latitudinal area weight $w(\theta_j)$ is used, for example, to calculate the global mean in the grid space.

**Appendix H: Derivation of Eq. (63) from Eq. (62)**

Here we describe the derivation of Eq. (63d) for odd $m$ ($\geq 3$) from Eq. (62a). Eqs. (63b,c) can also be derived similarly. By using Eqs. (52), (57), (11), (A2b,c), and the same equations as Eq. (53) except that $u_{n,m}^c$, $\chi_{n,m}^s$, and $\psi_{n,m}^c$ are replaced with $u_{n,m}^s$, $-\chi_{n,m}^c$, and $\psi_{n,m}^s$, respectively, and the same equations as Eq. (53) except that $u_{n,m}^c$, $\chi_{n,m}^s$, and $\psi_{n,m}^c$ are replaced with $v_{n,m}^c$, $\psi_{n,m}^s$, and $-\chi_{n,m}^c$, respectively, Eq. (62a) is converted into the following equation for odd $m \geq 3$:

$$\int_0^\pi \left\{ -m \frac{-\cos(n+1)\theta + \cos(n-1)\theta}{2} \left[ \frac{-m\chi_{1,m}^c}{2a} - \tilde{u}_{0,m}^s + \left( \frac{-m\chi_{2,m}^c}{2a} + \frac{3\psi_{1,m}^s - \psi_{3,m}^s}{4a} - \tilde{u}_{1,m}^s \right) \cos\theta \right. \right.$$

$$\left. + \sum_{n'=2}^N \left( \frac{m(\chi_{n'-1,m}^c - \chi_{n'+1,m}^c)}{2a} + \frac{n(-\psi_{n'-2,m}^s + 2\psi_{n',m}^s - \psi_{n'+2,m}^s)}{4a} - \tilde{u}_{n',m}^s \right) \cos n'\theta \right]$$

$$- \left( -\frac{n-2}{4}\cos(n-2)\theta + \frac{2n}{4}\cos n\theta - \frac{n+2}{4}\cos(n+2)\theta \right) \left[ \frac{m\psi_{1,m}^s}{2a} - \tilde{v}_{0,m}^c + \left( \frac{m\psi_{2,m}^s}{2a} + \frac{-3\chi_{1,m}^c + \chi_{3,m}^c}{4a} - \tilde{v}_{1,m}^c \right) \cos\theta \right.$$

$$\left. \left. + \sum_{n'=2}^N \left( \frac{m(-\psi_{n'-1,m}^s + \psi_{n'+1,m}^s)}{2a} + \frac{n'(\chi_{n'-2,m}^c - 2\chi_{n',m}^c + \chi_{n'+2,m}^c)}{4a} - \tilde{v}_{n',m}^c \right) \cos n'\theta \right] \right\} d\theta = 0. \tag{H1}$$

When $n \geq 4$, by using Eq. (A4a), Eq. (H1) can be converted into

$$\int_0^\pi \left\{ \frac{m}{2}\cos(n+1)\theta \left( \frac{m(\chi_{n,m}^c - \chi_{n+2,m}^c)}{2a} + \frac{(n+1)(-\psi_{n-1,m}^s + 2\psi_{n+1,m}^s - \psi_{n+3,m}^s)}{4a} - \tilde{u}_{n+1,m}^s \right) \cos(n+1)\theta \right.$$

$$-\frac{m}{2}\cos(n-1)\theta\left(\frac{m(\chi_{n-2,m}^{c}-\chi_{n,m}^{c})}{2a}+\frac{(n-1)(-\psi_{n-3,m}^{s}+2\psi_{n-1,m}^{s}-\psi_{n+1,m}^{s})}{4a}-\tilde{u}_{n-1,m}^{s}\right)\cos(n-1)\theta$$

$$+\frac{n-2}{4}\cos(n-2)\theta\left(\frac{m(-\psi_{n-3,m}^{s}+\psi_{n-1,m}^{s})}{2a}+\frac{(n-2)(\chi_{n-4,m}^{c}-2\chi_{n-2,m}^{c}+\chi_{n,m}^{c})}{4a}-\tilde{v}_{n-2,m}^{c}\right)\cos(n-2)\theta$$

$$-\frac{2n}{4}\cos n\theta\left(\frac{m(-\psi_{n-1,m}^{s}+\psi_{n+1,m}^{s})}{2a}+\frac{n(\chi_{n-2,m}^{c}-2\chi_{n,m}^{c}+\chi_{n+2,m}^{c})}{4a}-\tilde{v}_{n,m}^{c}\right)\cos n\theta$$

$$+\frac{n+2}{4}\cos(n+2)\theta\left(\frac{m(-\psi_{n+1,m}^{s}+\psi_{n+3,m}^{s})}{2a}+\frac{(n+2)(\chi_{n,m}^{c}-2\chi_{n+2,m}^{c}+\chi_{n+4,m}^{c})}{4a}-\tilde{v}_{n+2,m}^{c}\right)\cos(n+2)\theta\Big\}\,d\theta=0.$$

(H2)

From Eq. (H2) and (A4a), Eq. (63d) for $n \geq 4$ is derived. Equation (63d) for $n \leq 3$ can also be derived from (H1) and (A4a).

## Appendix I: Two-time-level semi-implicit semi-Lagrangian scheme for time integration

A two-time-level semi-implicit semi-Lagrangian scheme (e.g., Temperton et al., 2001) and the Stable Extrapolation Two-Time-Level Scheme (SETTLS; Hortal, 2002) are adopted to discretize the shallow water equations in Eqs. (98) and (99) in

time as

$$\frac{(\mathbf{v}+2\mathbf{\Omega}\times\mathbf{r})^{+}-(\mathbf{v}+2\mathbf{\Omega}\times\mathbf{r})_{D}^{0}}{\Delta t}=-\frac{g(\nabla h_{D}^{(+)}+\nabla h^{0})}{2}+\beta_{\mathbf{v}}\frac{g(\nabla h_{D}^{(+)}+\nabla h^{0})}{2}-\beta_{\mathbf{v}}\frac{g(\nabla h_{D}^{0}+\nabla h^{+})}{2},\qquad (I1)$$

$$\frac{h^{+}-h_{D}^{0}}{\Delta t}=-\frac{[(h-h_{s})D]_{D}^{(+)}+[(h-h_{s})D]^{0}}{2}+\frac{[\mathbf{v}\cdot\nabla h_{s}]_{D}^{(+)}+[\mathbf{v}\cdot\nabla h_{s}]^{0}}{2}$$

$$+\beta_{h}\frac{[\bar{h}D]_{D}^{(+)}+[\bar{h}D]^{0}}{2}-\beta_{h}\frac{[\bar{h}D]_{D}^{0}+[\bar{h}D]^{+}}{2},\qquad (I2)$$

where

$$D\equiv\nabla\cdot\mathbf{v}=\frac{1}{a}\left[\frac{1}{\cos\phi}\frac{\partial u}{\partial\lambda}+\frac{1}{\cos\phi}\frac{\partial v\cos\phi}{\partial\phi}\right]\qquad (I3)$$

is horizontal divergence; $\Delta t$ is a timestep; the superscripts $-$, 0, and $+$ mean past time $(t-\Delta t)$, present time $(t)$, and future time $(t+\Delta t)$, respectively, and the superscript $(+)$ means future time $(t+\Delta t)$ extrapolated in time, for example, $h^{(+)}=2h^{0}-h^{-}$; the subscript D means the departure point, and the absence of the subscript D means the arrival point; $\bar{h}$ is a constant value of height for semi-implicit linear terms; $\beta_{\mathbf{v}}$ and $\beta_{h}$ are second-order decentering parameters (Yukimoto et al.,

2011). Using $\beta_{\mathbf{v}}$ and $\beta_{h}$ larger than 1.0 (e.g., 1.2) increases the effect of the semi-implicit scheme improving computational stability, but $\beta_{\mathbf{v}}=\beta_{h}=1.0$ is used here because $\bar{h}$ larger than $h$ is enough for stable calculations in the shallow water model. The departure point $\mathbf{x}_{D}$ is the upstream horizontal position from the arrival point $\mathbf{x}$ along the wind vector between present time $(t)$ and future time $(t+\Delta t)$. Here, the arrival point $\mathbf{x}$ is on a grid point, and the departure point $\mathbf{x}_{D}$ is not generally on a grid point. Since the right-hand sides of Eqs. (I1) and (I2) are the time average between present time $(t)$ and future time

$(t + \Delta t)$ and the spatial average between the departure point and the arrival point, these equations have second-order precision in time and space. In SETTLS, $\boldsymbol{x}_D$ is calculated using

$$\boldsymbol{x}_D = \boldsymbol{x} - \frac{\boldsymbol{v}_D^{(+)} + \boldsymbol{v}^0}{2}\Delta t. \tag{I4}$$

However, when $\Delta t$ is longer than 30 minutes, using $\boldsymbol{v}_D^{(+)}$ extrapolated in time to calculate $\boldsymbol{x}_D$ causes numerical instability in our experiments. To avoid instability when $\Delta t$ is 1 hour, here we use

$$\boldsymbol{x}_D = \boldsymbol{x} - \frac{\boldsymbol{v}_D^0 + \boldsymbol{v}'^+}{2}\Delta t, \tag{I5a}$$

$$\boldsymbol{v}'^+ \equiv \boldsymbol{v}_D^0 + (2\boldsymbol{\Omega} \times \boldsymbol{r})_D - 2\boldsymbol{\Omega} \times \boldsymbol{r} - \frac{g\left(\nabla h_D^{(+)} + \nabla h^0\right)}{2}\Delta t, \tag{I5b}$$

instead of Eq. (I4), where $\boldsymbol{v}'^+$ is a provisional future value obtained by discretizing Eq. (98) in an explicit semi-Lagrangian scheme. From Eq. (I5), we obtain

$$\boldsymbol{x}_D = \boldsymbol{x} - \Delta t\left[\left(\boldsymbol{v}^0 + \boldsymbol{\Omega} \times \boldsymbol{r} - \frac{g\Delta t\nabla h^{(+)}}{4}\right)_D - \boldsymbol{\Omega} \times \boldsymbol{r} - \frac{g\Delta t\nabla h^0}{4}\right]. \tag{I6}$$

This method using a provisional future value to calculate $\boldsymbol{x}_D$ is similar to the method in Gospodinov et al. (2001). Since the value with the subscript D depends on $\boldsymbol{x}_D$, $\boldsymbol{x}_D$ is calculated iteratively from Eq. (I6) (e.g., Ritchie et al., 1995; Temperton et al., 2001). Since $\boldsymbol{x}_D$ is not generally on the grid point, the value at $\boldsymbol{x}_D$ is calculated by spatial interpolation from nearby grid points. In the right-hand side of Eq. (I6), the value at $\boldsymbol{x}_D$ with the subscript D is calculated by third-order Lagrange interpolation.

Eqs. (I1) and (I2) are converted into

$$\boldsymbol{v}^+ + \frac{\beta_v\Delta t}{2}g\nabla h^+ = \boldsymbol{R}_v, \tag{I7a}$$

$$\boldsymbol{R}_v \equiv \left[\boldsymbol{v}^0 + 2\boldsymbol{\Omega} \times \boldsymbol{r} - \frac{\Delta t}{2}g\left(\nabla h^{(+)} - \beta_v\nabla h^{(+)} + \beta_v\nabla h^0\right)\right]_D - 2\boldsymbol{\Omega} \times \boldsymbol{r} - \frac{\Delta t}{2}g(\nabla h^0 - \beta_v\nabla h^0), \tag{I7b}$$

$$h^+ + \frac{\beta_h\Delta t}{2}\bar{h}D^+ = R_h, \tag{I8a}$$

$$R_h \equiv \left\{h^0 + \frac{\Delta t}{2}\left[-(h - h_s)D^{(+)} + \boldsymbol{v} \cdot \nabla h_s^{(+)} + \beta_h\bar{h}D^{(+)} - \beta_h\bar{h}D^0\right]\right\}_D$$

$$+ \frac{\Delta t}{2}\left[-(h - h_s)D^0 + \boldsymbol{v} \cdot \nabla h_s^0 + \beta_h\bar{h}D^0\right]. \tag{I8b}$$

In Eqs. (I7b) and (I8b), the values at $\boldsymbol{x}_D$ with the subscript D are calculated by fifth-order and third-order Lagrange interpolations, respectively, since high-order interpolation of wind vector components increases the accuracy of the model's results in our experiments. From Eq. (I7), we obtain

$$D^+ + \frac{\beta_v\Delta t}{2}g\nabla^2 h^+ = R_D, \tag{I9}$$

$$\zeta^+ = R_\zeta, \tag{I10}$$

where

$$\zeta \equiv \boldsymbol{k} \cdot \nabla \times \boldsymbol{v} = \frac{1}{a}\left[\frac{1}{\cos\phi}\frac{\partial v}{\partial\lambda} - \frac{1}{\cos\phi}\frac{\partial u\cos\phi}{\partial\phi}\right] \tag{I11}$$

is vorticity, $\boldsymbol{k} \equiv \boldsymbol{r}/|\boldsymbol{r}|$ is the vertical unit vector, $R_D \equiv \nabla \cdot \boldsymbol{R_v}$ and $R_\zeta \equiv \boldsymbol{k} \cdot \nabla \times \boldsymbol{R_v}$.

We calculate $h^+$ and $\boldsymbol{v}^+$ using the spectral transform method and the Galerkin method with the new DFS method as follows. (See Sect. 3.10 for the spectral transform with the new DFS method.)

1. The scalar variable $R_h$ is transformed from grid space to spectral space using Eqs. (9), (23), (30) and (31). The components of the vector variable $\boldsymbol{R_v} = (R_u, R_v)$ in grid space are transformed to $R_\chi$ and $R_\psi$ in spectral space using Eqs. (56), (57), (63) and (64), where $R_\chi$ and $R_\psi$ are the velocity potential and the stream function of $\boldsymbol{R_v}$, respectively.

2. $R_D$ and $R_\zeta$ are calculated by

$$R_D = \nabla^2 R_\chi, \tag{I12}$$

$$R_\zeta = \nabla^2 R_\psi, \tag{I13}$$

using Eqs. (75) and (77). $\zeta^+$ is obtained from $R_\zeta$ using Eq. (I10).

3. Equations (I8a) and (I12) are substituted into Eq. (I9) and we obtain

$$D^+ - \left(\frac{\Delta t}{2}\right)^2 \beta_{\mathbf{v}}\beta_h g\bar{h}\nabla^2 D^+ = \nabla^2\left(R_\chi - \frac{\Delta t}{2}\beta_{\mathbf{v}} g R_h\right). \tag{I14}$$

$D^+$ is calculated by solving the Helmholtz-like equation Eq. (I14) using Eqs. (83) and (85).

4. $h^+$ is calculated from $D^+$ and $R_h$ using Eq. (I8).

5. $\chi^+$ and $\psi^+$ are calculated from $D^+$ and $\zeta^+$ by solving the Poisson equations

$$\nabla^2\chi^+ = D^+, \tag{I15}$$

$$\nabla^2\psi^+ = \zeta^+, \tag{I16}$$

using Eqs. (75) and (78).

6. $\mathbf{v}^+ = (u^+, v^+)$ is calculated from $\chi^+$ and $\psi^+$ using Eq. (53) for $u_{n,m}^c$ and the similar equations for $u_{n,m}^s$, $v_{n,m}^c$, and $v_{n,m}^s$.

7. $u^+, v^+, h^+, D^+$, and $\nabla h^+$ in spectral space are transformed to grid space. $h^+$ and $D^+$ are transformed meridionally using Eqs. (14) and (13). $u^+$ and $v^+$ are transformed meridionally using Eq. (52). $\nabla h^+ = (h_\lambda^+, h_\theta^+)$ is transformed meridionally using Eqs. (18)–(21). $h_\lambda^+$ can also be calculated from $h_m^{+c,N}(\theta_j)$ and $h_m^{+s,N}(\theta_j)$ at the latitudinal grid points using Eq. (16), and additionally using Eq. (22) at the poles when using Grid [1], which is more efficient than using Eqs. (18) and (19) because the meridional inverse discrete cosine and sine transforms of $h_\lambda^+$ become unnecessary.

**Author Contributions.**

HY developed a new DFS method and a shallow water model using the method, conducted model experiments, analysed data, and wrote the paper.

**Competing Interests.**

The author declares that there is no conflict of interest.

**Acknowledgements.**

We are grateful to Keiichi Ishioka (Kyoto University), Tadashi Tsuyuki (MRI), Daisuke Hotta (MRI), Masashi Ujiie (JMA),
and other members of the MRI and JMA model development teams for their useful comments. We are also grateful to the referees and the editor for their comments that help to improve the paper quality. This work was supported by the Integrated Research Program for Advancing Climate Models (TOUGOU) Grant Number JPMXD0717935561 from the Ministry of Education, Culture, Sports, Science and Technology (MEXT), Japan.

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

**Table 1.** Normalized $L_2$ errors of Laplacian operator calculation ($\nabla^2 f$). We use the old DFS method with Grid [0], the new DFS methods with Grid [0], Grid [1] and Grid [-1], and the SH method. $J^0$ is the number of latitudinal grid points in Grid [0]. The truncation wavenumber $N \cong 2J^0/3$.

| Resolution\ Method | Old DFS [0] | DFS [0] | DFS [1] | DFS [−1] | SH |
|---|---|---|---|---|---|
| $J^0$=64, $N$=42 | 4.1208E−3 | 2.3019E−3 | 2.2530E−3 | 2.6281E−3 | 2.0927E−3 |
| $J^0$=160, $N$=106 | 2.2221E−4 | 2.3678E−4 | 2.3369E−4 | 2.3374E−4 | 2.1668E−4 |
| $J^0$=320, $N$=213 | 3.8070E−5 | 3.7931E−5 | 3.8752E−5 | 3.8740E−5 | 3.7565E−5 |
| $J^0$=960, $N$=639 | 2.4281E−6 | 3.5687E−6 | 3.5888E−6 | 3.5904E−6 | 2.3453E−6 |

**Table 2.** Same as Table 1 except that the global mean values of calculated $\nabla^2 f$ are shown.

| Resolution\ Method | Old DFS [0] | DFS [0] | DFS [1] | DFS [−1] | SH |
|---|---|---|---|---|---|
| $J^0$=64, $N$=42 | 3.4331E−26 | 2.2012E−26 | −1.2242E−25 | −6.4414E−25 | −3.8370E−27 |
| $J^0$=160, $N$=106 | −6.1392E−27 | 2.9404E−26 | 3.1530E−25 | −4.1152E−25 | 3.0050E−26 |
| $J^0$=320, $N$=213 | −2.9272E−26 | −4.4429E−28 | 1.3779E−24 | −1.0004E−24 | 3.3190E−26 |
| $J^0$=960, $N$=639 | −4.6309E−26 | −3.5020E−26 | 2.3521E-24 | 4.7404E−25 | 9.4697E−27 |

**Table 3.** Same as Table 1 except that $L_2$ errors of the solution of the Helmholtz equation are shown.

| Resolution\ Method | Old DFS [0] | DFS [0] | DFS [1] | DFS [−1] | SH |
|---|---|---|---|---|---|
| $J^0$=64, $N$=42 | 7.5000E−4 | 7.0729E−4 | 7.3360E−4 | 7.5868E−4 | 6.4564E−4 |
| $J^0$=160, $N$=106 | 1.7270E−5 | 1.7263E−5 | 1.5884E−5 | 1.5907E−5 | 3.0100E−5 |
| $J^0$=320, $N$=213 | 1.0970E−6 | 1.0965E−6 | 1.2557E−6 | 1.2602E−6 | 2.7348E−6 |
| $J^0$=960, $N$=639 | 4.3114E−8 | 4.3114E−8 | 3.8081E−8 | 3.8253E−8 | 3.7720E−8- |

**Table 4.** Normalized $L_2$ errors of the predicted height after a 12-day integration in the Williamson test case 1 when using the Eulerian advection models. The truncation wavenumber $N \cong 2J^0/3$.

| Resolution \ Model | Old DFS [0] | DFS [0] | DFS [1] | DFS [−1] | SH |
|---|---|---|---|---|---|
| $J^0$=64, $N$=42 | Unstable | 1.1557E−1 | 1.1559E−1 | 1.1559E−1 | 1.1554E−1 |
| $J^0$=160, $N$=106 | Unstable | 5.0956E−2 | 5.0954E−2 | 5.0954E−2 | 5.0955E−2 |
| $J^0$=320, $N$=213 | Unstable | 2.4619E−2 | 2.4619E−2 | 2.4619E−2 | 2.4619E−2 |
| $J^0$=960, $N$=639 | Unstable | 8.2424E−3 | 8.2424E−3 | 8.2424E−3 | 8.2424E−3 |

**Table 5.** Same as Table 4 except for using the semi-Lagrangian models and the truncation wavenumber $N \cong J^0 - 1$.

| Resolution \ Model | Old DFS [0] | DFS [0] | DFS [1] | DFS [−1] | SH |
|---|---|---|---|---|---|
| $J^0$=64, $N$=63 | 1.6782E−1 | 1.6782E−1 | 1.6795E−1 | 1.6849E−1 | 1.6464E−1 |
| $J^0$=160, $N$=159 | 2.0076E−2 | 2.0076E−2 | 2.0074E−2 | 2.0080E−2 | 1.9887E−2 |
| $J^0$=320, $N$=319 | 3.4033E−3 | 3.4033E−3 | 3.4029E−3 | 3.4033E−3 | 3.3855E−3 |
| $J^0$=960, $N$=959 | 2.1503E−4 | 2.1503E−4 | 2.1503E−4 | 2.1504E−4 | 2.1514E−4 |

**Table 6.** Same as Table 5 except for the errors after a 5-day integration in the Williamson test case 2

| Resolution \ Model | Old DFS [0] | DFS [0] | DFS [1] | DFS [−1] | SH |
|---|---|---|---|---|---|
| $J^0$=64, $N$=63 | 2.4468E−05 | 2.4468E−05 | 2.4453E−05 | 2.4434E−05 | 2.4147E−05 |
| $J^0$=160, $N$=159 | 1.3462E−06 | 1.3462E−06 | 1.3463E−06 | 1.3458E−06 | 1.3402E−06 |
| $J^0$=320, $N$=319 | 4.1918E−07 | 4.1918E−07 | 4.1918E−07 | 4.1916E−07 | 4.1927E−07 |
| $J^0$=960, $N$=959 | 1.1800E−07 | 1.1800E−07 | 1.1800E−07 | 1.1800E−07 | 1.1800E−07 |

**Table 7.** Same as Table 5 except for the errors after a 15-day integration in the Williamson test case 5. The result of the high-resolution SH model with $J^0 = 960$ and $N = 958$ is regarded as the reference solution.

| Resolution \ Model | Old DFS [0] | DFS [0] | DFS [1] | DFS [−1] | SH |
|---|---|---|---|---|---|
| $J^0$=64, $N$=63 | 8.2998E−4 | 8.2972E−4 | 8.2559E−4 | 8.2533E−4 | 8.2575E−4 |
| $J^0$=160, $N$=159 | 9.2568E−4 | 9.2569E−4 | 9.2571E−4 | 9.2607E−4 | 9.2578E−4 |
| $J^0$=320, $N$=319 | 8.3815E−4 | 8.3815E−4 | 8.3813E−4 | 8.3807E−4 | 8.3812E−4 |

**Table 8.** Same as Table 7 except for $N \cong 2J^0/3$.

| Resolution \ Model | Old DFS [0] | DFS [0] | DFS [1] | DFS [−1] | SH |
|---|---|---|---|---|---|
| $J^0$=64, $N$=42 | Unstable | 8.2985E−4 | 8.2555E−4 | 8.2545E−4 | 8.2587E−4 |
| $J^0$=160, $N$=106 | Unstable | 9.2571E−4 | 9.2573E−4 | 9.2571E−4 | 9.2584E−4 |
| $J^0$=320, $N$=259 | Unstable | 8.3814E−4 | 8.3813E−4 | 8.3812E−4 | 8.3812E−4 |

**Table 9.** Same as Table 7 except for the errors after a 14-day integration in the Williamson test case 6

| Resolution \ Model | Old DFS [0] | DFS [0] | DFS [1] | DFS [−1] | SH |
|---|---|---|---|---|---|
| $J^0$=64, $N$=63 | 1.0319E−2 | 1.0361E−2 | 7.2824E−3 | 8.7423E−3 | 1.0118E−2 |
| $J^0$=160, $N$=159 | 2.7830E−3 | 2.7830E−3 | 1.5615E−3 | 2.0704E−3 | 2.7766E−3 |
| $J^0$=320, $N$=319 | 9.3546E−4 | 9.3546E−4 | 5.6164E−4 | 6.8201E−4 | 9.3560E−4 |

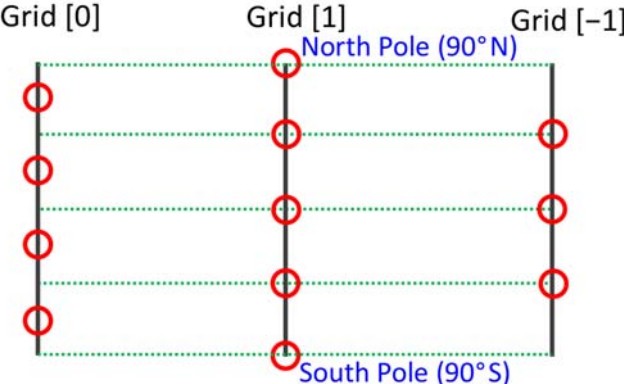

**Figure 1.** Grid [0], Grid[1], and Grid [−1] are three ways of arranging equally spaced latitudinal grid points. Red circles show the positions of the grid points when the grid interval $\Delta\theta = \pi/4$.

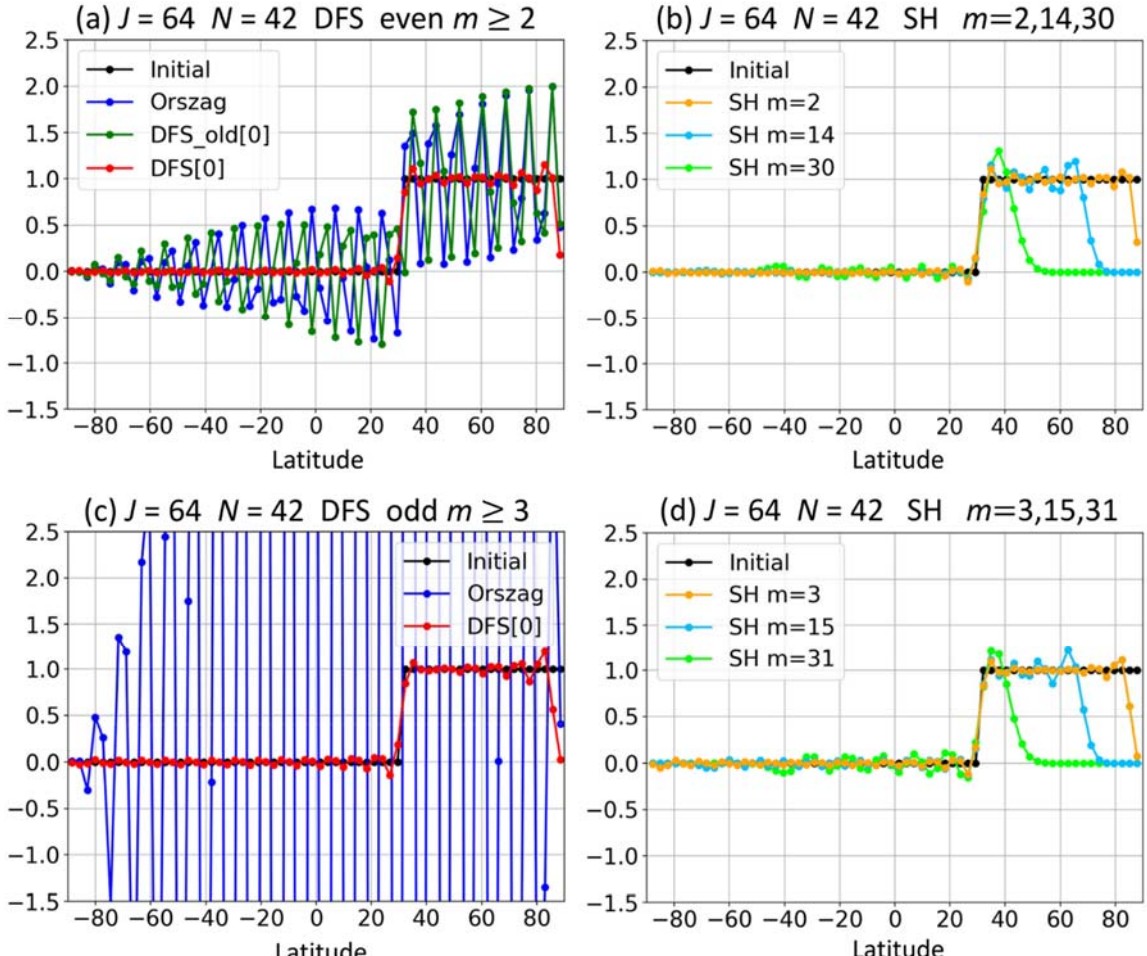

**Figure 2.** Change in values at the grid points due to the meridional wavenumber truncation. We use Grid [0] with the number of latitudinal grid points $J = 64$. Initial values (black) are meridionally transformed from grid space to spectral space, truncated with $N = 42$, and transformed back from spectral space to grid space. (a) Values for even $|m| \geq 2$ when using the DFS method of Orszag (blue), the old DFS method (green), and the new DFS method (red) with Grid [0]. (b) Values for $m = 2$ (orange), 14 (deep sky blue), 30 (lime) when using the SH expansion method with the gaussian grid. (c) Same as (a) except for the values for odd $|m| \geq 3$. (d) Same as (b) except for the values for $m = 3$ (orange), 15 (deep sky blue), 31 (lime).

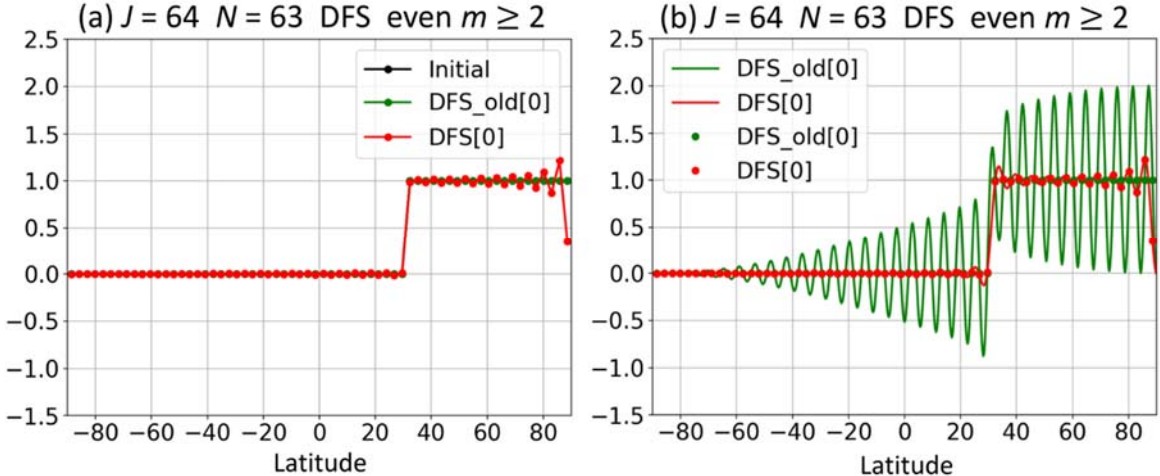

**Figure 3.** (a) Same as Fig 2(a) except for $N = 63$. (b) Same as (a) except that the values between grid points calculated from the expansion coefficients are also shown.

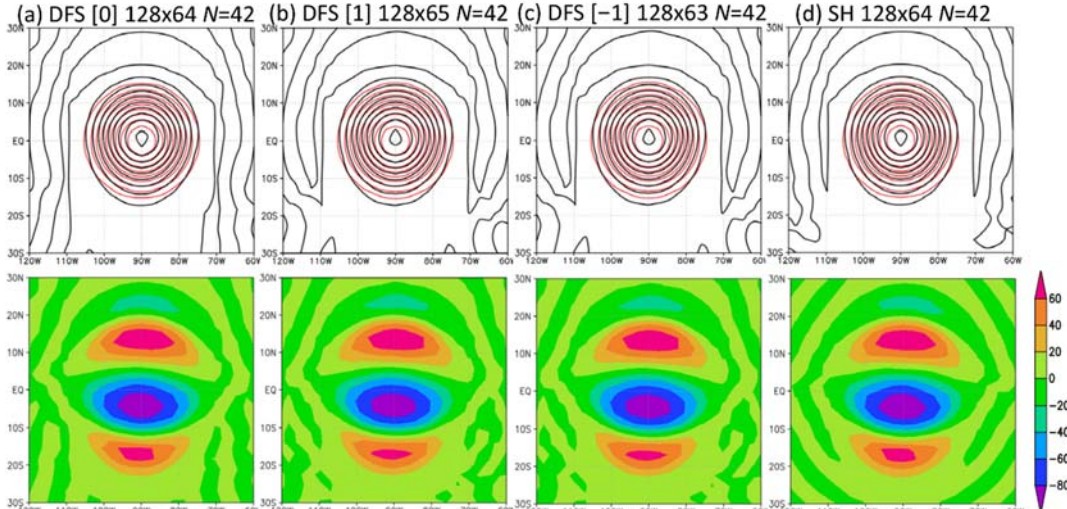

**Figure 4.** Predicted height (m) in the Eulerian models after a 12-day integration in Williamson test case 1. (a) New DFS model with Grid [0]. (b) New DFS model with Grid [1]. (c) New DFS model with Grid [−1]. (d) SH model. The number of longitudinal ($I$) and latitudinal ($J$) grid points is shown in the form $I \times J$. In the upper figures, the black contour shows the predicted height, and the red contour shows the reference solution. In the lower figures, color shading shows the difference between the predicted height and the reference solution.

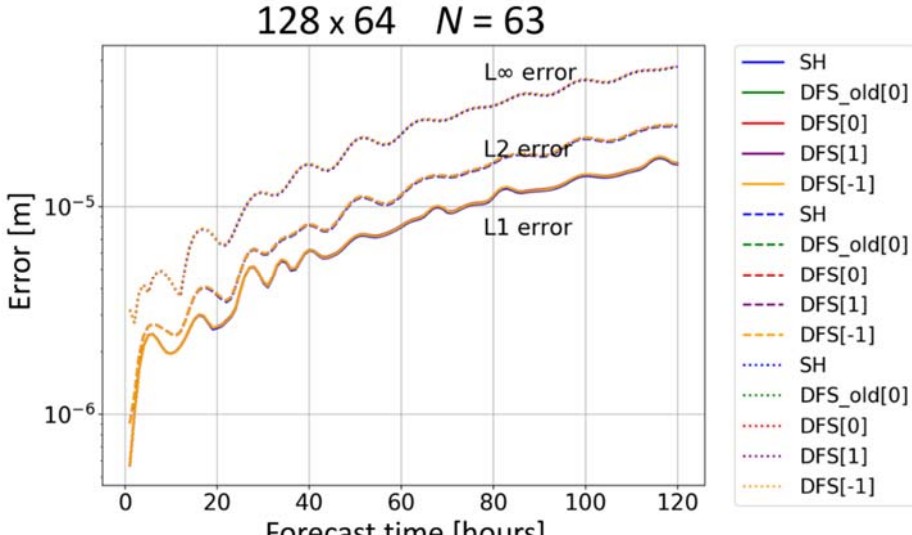

**Figure 5.** Time series of prediction error of height (m) for 5 days (120 hours) integration in Williamson test case 2 ($\alpha = \pi/2 - 0.05$). The number of longitudinal grid points $I = 128$. The number of latitudinal grid points in Grid [0] is $J^0 = 64$. The truncation wavenumber $N = 63$. Solid, dashed, and dotted lines represent normalized $L_1$, $L_2$, and $L_\infty$ errors, respectively. The colors blue, green, red, purple, and orange represent the models using SH, old DFS with Grid [0], new DFS with Grid [0], new DFS with Grid [1], and new DFS with Grid [−1], respectively.

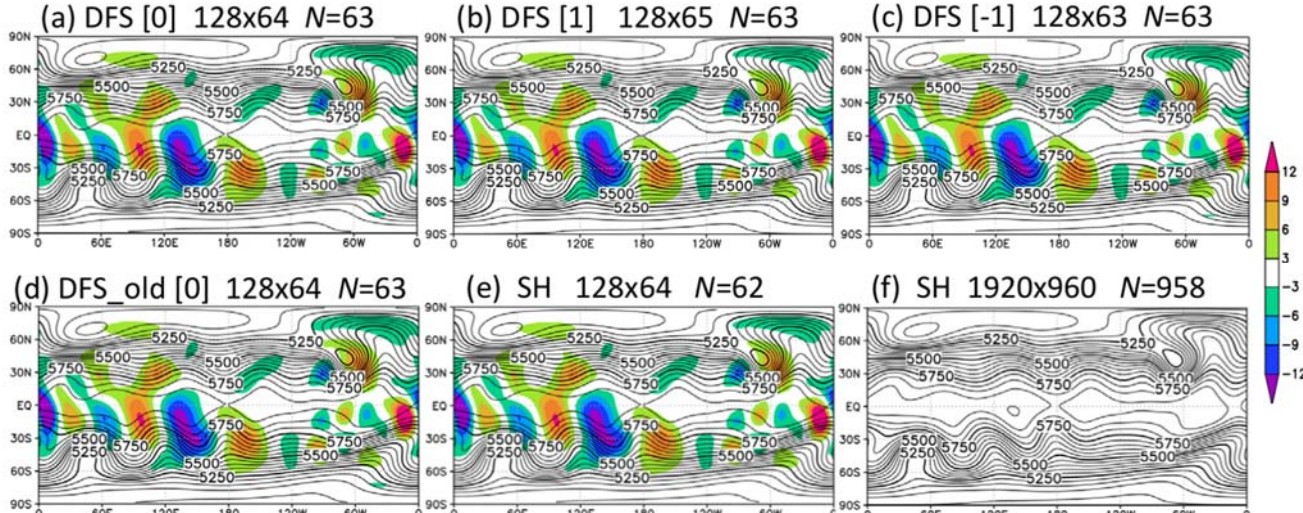

**Figure 6.** Predicted height (m) after a 15-day integration in Williamson test case 5. (a) New DFS model with Grid [0]. (b) New DFS model with Grid [1]. (c) New DFS model with Grid [−1]. (d) Old DFS model with Grid [0]. (e) SH model. (f) SH model at high resolution, which is regarded as the reference solution. The number of longitudinal (*I*) and latitudinal (*J*) grid points is shown in the form *I* × *J*. *N* is the truncation wavenumber. Color shading shows the error with respect to the reference solution.

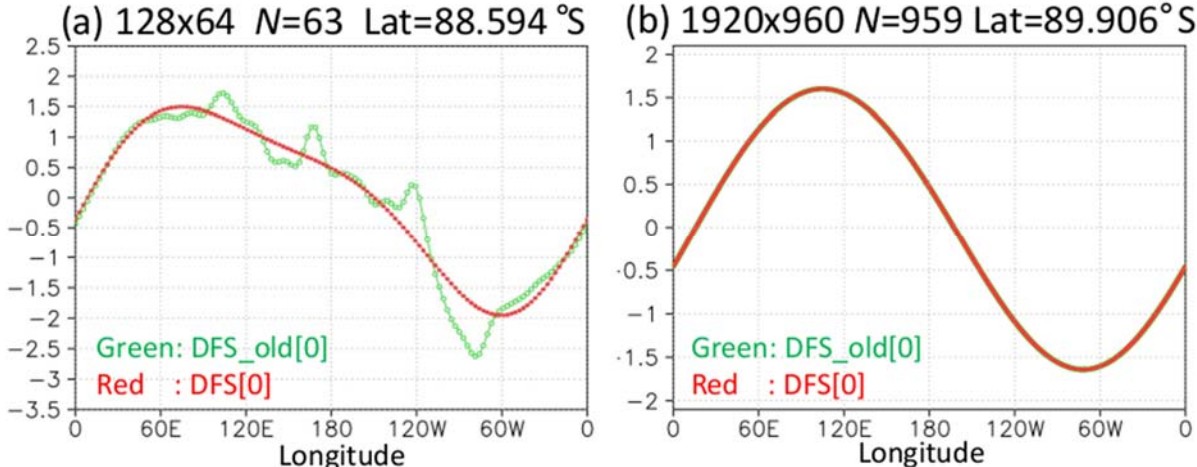

**Figure 7.** Longitudinal distributions of meridional wind $(\mathrm{m\,s^{-1}})$ at the grid points near the South Pole after a 15-day integration in Williamson test case 5. Results of the models using Grid [0] with (a) $I = 128, J^0 = 64$ and $N = 63$, and (b) $I = 1920, J^0 = 960$ and $N = 959$. Green (red) lines represent the old (new) DFS models.

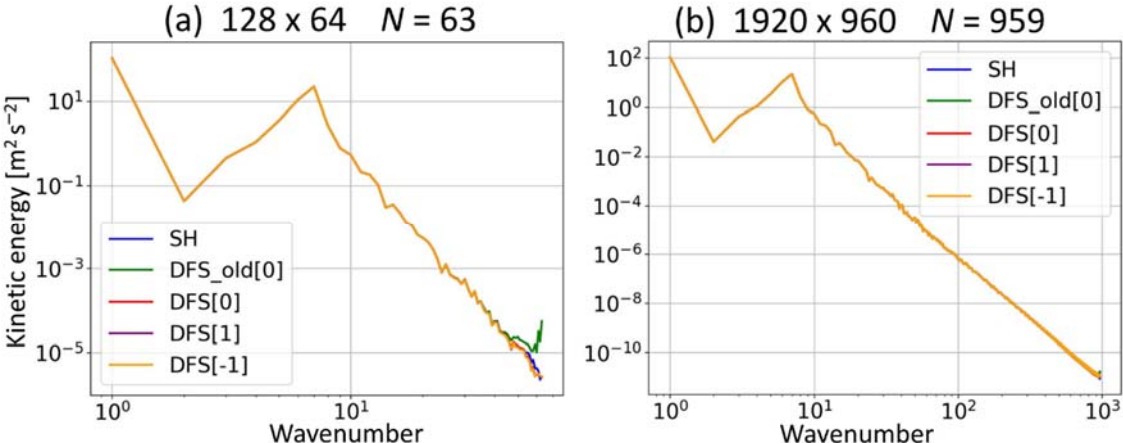

**Figure 8.** Kinetic energy spectrum of horizontal winds ($m^2s^{-2}$) after a 15-day integration in Williamson test case 5. Results of the models with (a) $I = 128, J^0 = 64$, and $N = 63$ (DFS) or $N = 62$ (SH), and (b) $I = 1920, J^0 = 960$ and $N = 959$ (DFS) or 958 (SH). The colors blue, green, red, purple, and orange represent the models using SH, old DFS with Grid [0], new DFS with Grid [0], new DFS with Grid [1], and new DFS with Grid [−1], respectively.

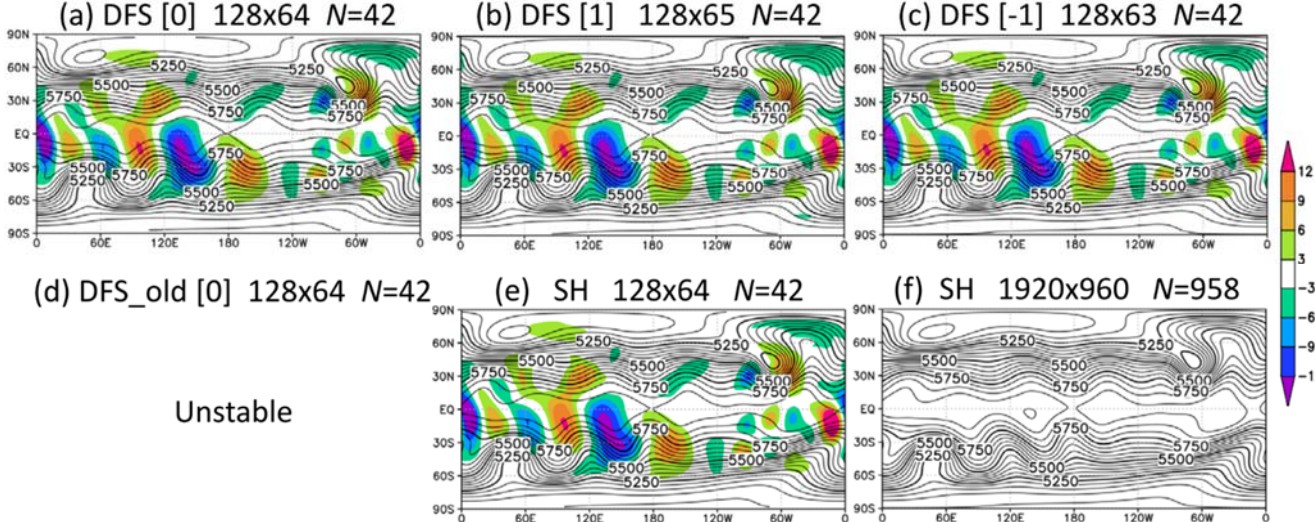

**Figure 9.** Same as Fig. 6, except with truncation wavenumber $N$.

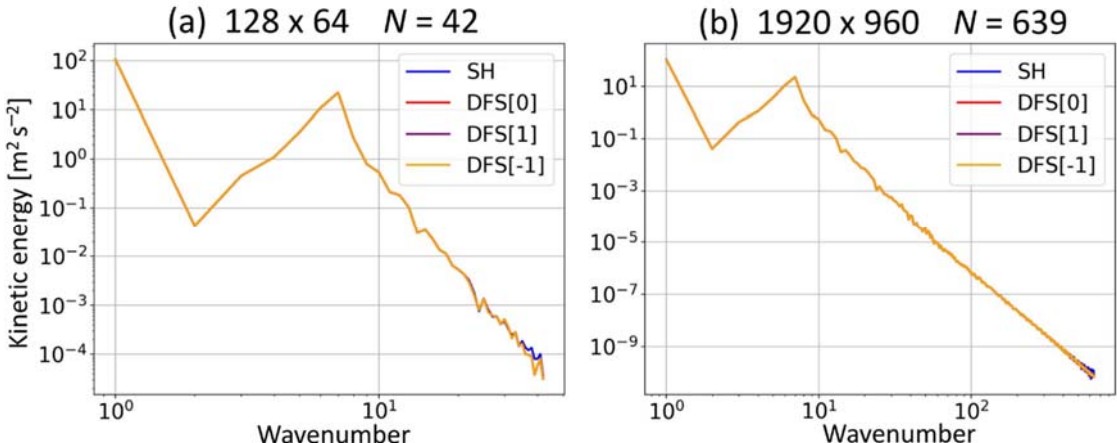

**Figure 10.** Same as Fig. 8, except with truncation wavenumber *N*.

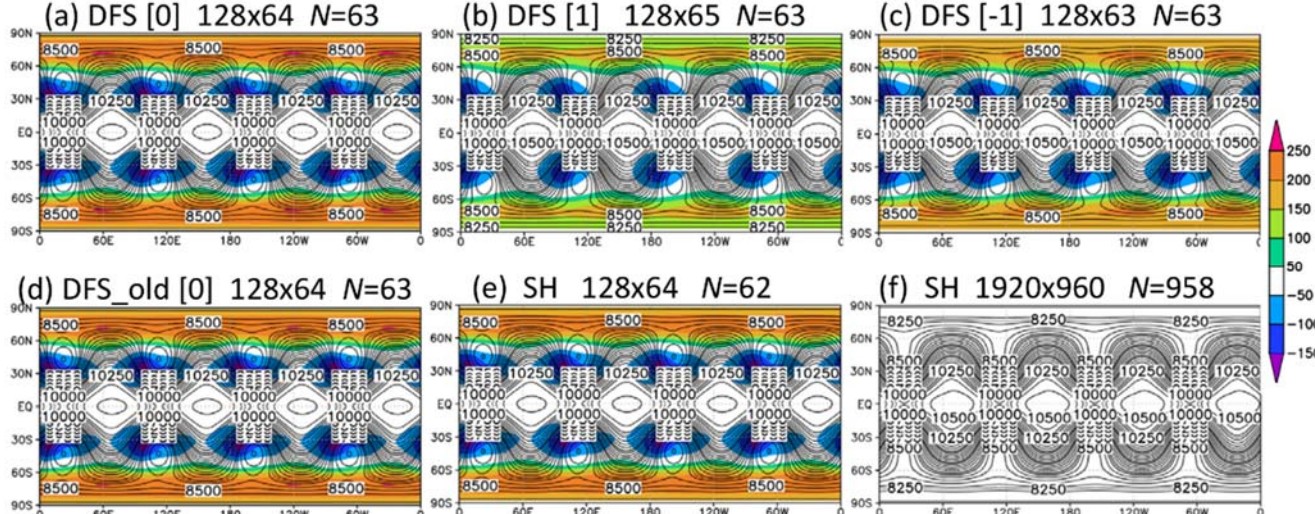

**Figure 11.** Same as Fig. 6 except for predicted height (m) after a 14-day integration in Williamson test case 6.

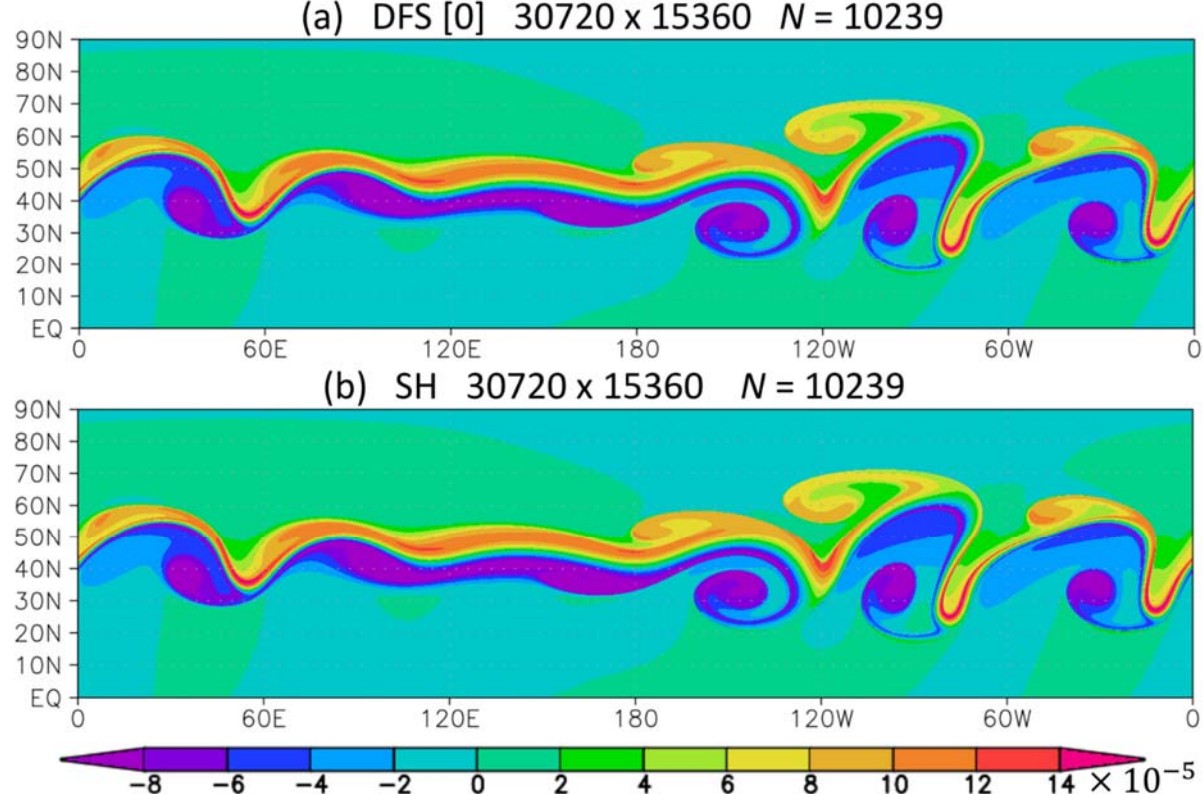

**Figure 12.** Predicted vorticity ($s^{-1}$) after a 6-day integration in the Galewsky test case. (a) The new DFS model with Grid [0], and (b) the SH model at 1.3 km resolution with $I = 30720$, $J^0 = 15360$ and $N = 10239$.

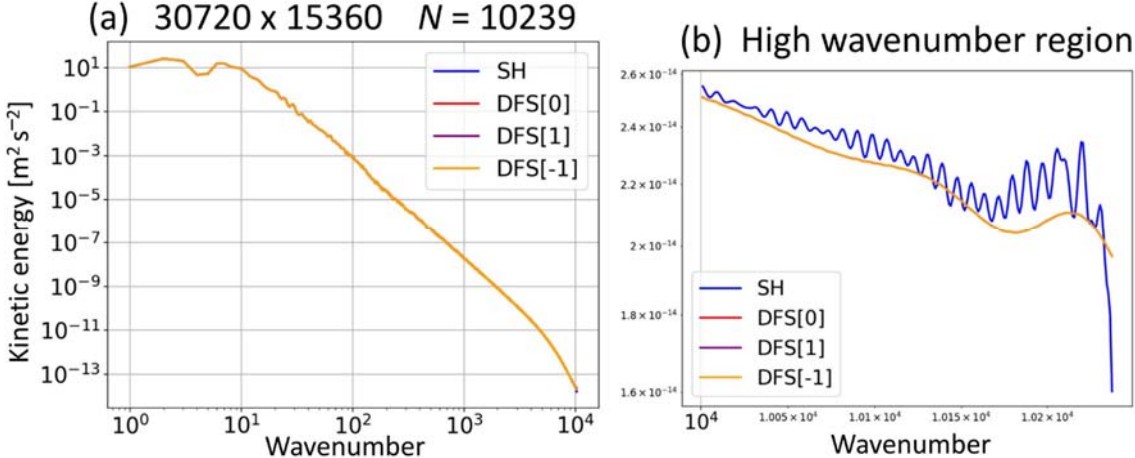

**Figure 13.** Kinetic energy spectrum of horizontal winds ($m^2s^{-2}$) after a 6-day integration in the Galewsky test case. (a) Results of the models with $I = 30720$, $J^0 = 15360$ and $N = 10239$. The colors blue, red, purple, and orange represent the models using SH, DFS with Grid [0], DFS with Grid [1], and DFS with Grid [−1], respectively. (b) As (a), but showing the high-wavenumber region.

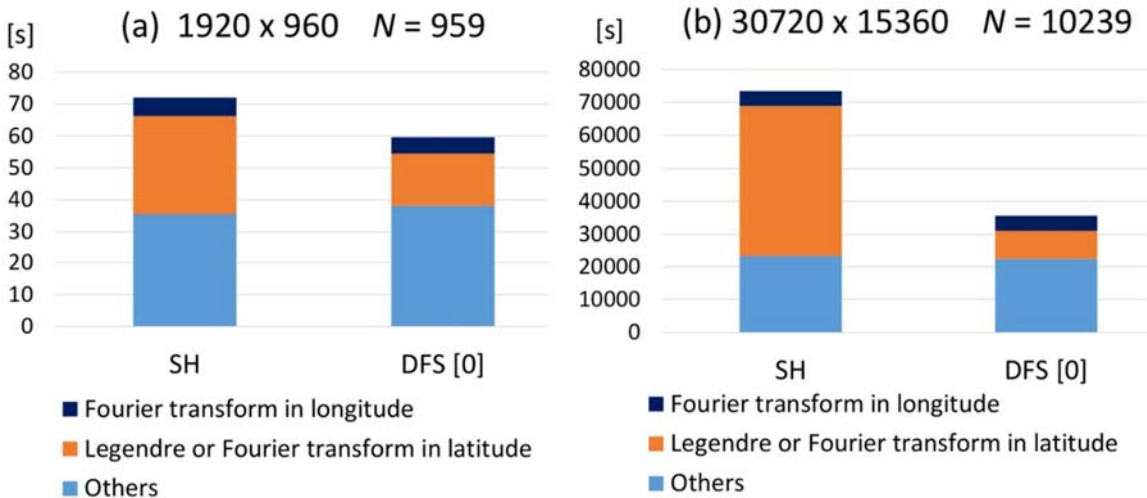

**Figure 14.** Elapsed time (s) for (a) 15-day integration in Williamson test case 5 in the SH model and the new DFS model at 20 km resolution with $I = 1920$, $J^0 = 960$ and $N = 959$, and (b) 6-day integration in the Galewsky test case at 1.3 km resolution with $I = 30720$, $J^0 = 15360$ and $N = 10239$. There is no monitoring output during elapsed time measurement