# Peer review of "Improved double Fourier series on a sphere and its application to a semi-implicit semi-Lagrangian shallow water model"

_Geoscientific Model Development, 2021_

## Referee Comment (RC1)

**Comments to the manuscript**

**General comments**

  This paper deals with the Double Fourier Series (DFS) on the sphere where new DFS functions are used to represent the variables on the global domain. Discretization procedures for the spatial differentiations, elliptic equations, and the shallow water models are shown in some detail using the trigonometric identities. Combining the DFS method with the semi-Lagrangian time-differentiation, the paper provided simulation results for a couple of shallow water model test cases, including the standard test cases of Williamson et al. (1992). The author emphasizes that the DFS functions in the present study improves the simulation results over the DFS models in the previous studies. However, it is hard to be convinced of the improvement of the new DFS method due to less rigorous assessment of the solution method for differential operators such as elliptic equations, least square method, and also limited test case results. Specific comments are shown below.

**Specific major comments**

[1] One of the most important aspect of the paper is that the DFS expansion coefficients ($T_{n,m}^c, T_{n,m}^s$) are calculated based on the least square method, as is shown section 2.3. I am afraid, however, that the derivation procedure does not seem to be the least square method which is required for determining the expansion coefficients. The residual function here is defined using the difference between the spectral representation of the function ($T_m^c, T_m^s$) two different sets of DFS. The fact that, for the spherical harmonics model (SHM), the spectral coefficients are determined in the least square sense on the spherical domain is explained as below:

$$E = \int_0^\pi \left[ T_m^c(\theta) - \sum_{n=m}^N c_n P_n^m(\theta) \right]^2 \sin\theta d\theta$$

$$\frac{\partial E}{\partial c_n} = 0 \ \cdots \ \text{least squared error}$$

$$2\int_0^\pi P_n^m(\theta) \left[ T_m^c(\theta) - \sum_{n=m}^N c_n P_n^m(\theta) \right] \sin\theta d\theta = 0$$

$$\therefore \ c_n = \int_0^\pi P_n^m(\theta) T_m^c(\theta) \sin\theta d\theta$$

$$\left[ \because \ \int_0^\pi P_n^m(\theta) P_{n'}^m(\theta) \sin\theta d\theta = \delta_{nn'} \right]$$

That is, $c_n$ is obtained with least squared error on the sphere.

The least square method for the SHM in the present study is different from above equations.

[2] It looks like that the equations (24a)-(24d) are just algebraic equations resulted from simply multiplying $\sin\theta$ or $\sin^2\theta$ or $\sin^4\theta$ to the same equation. For instance, in the case of odd $m$ ($>=3$), it follows:

$$T_m^c(\theta) = \sum_n T_{n,m}^c \sin^2\theta\sin n\theta$$

$$= \sum_n \tilde{T}_{n,m}^c \sin n\theta$$

$$\sin^2\theta T_m^c(\theta) = \sum_n T_{n,m}^c \sin^4\theta\sin n\theta$$

$$= \sum_n \tilde{T}_{n,m}^c \sin^2\theta\sin n\theta$$

$$\sum_n T_{n,m}^c \sin^4\theta\sin n\theta = \sum_n h_{n,m}^c \sin n\theta$$

$$\Rightarrow \underbrace{\begin{bmatrix} 5-diagonal \\ matrix \end{bmatrix}}_{\text{lhs of } (24d)} \begin{bmatrix} T_{1,m}^c \\ T_{3,m}^c \\ T_{5,m}^c \\ \vdots \end{bmatrix} = \begin{bmatrix} h_{1,m}^c \\ h_{3,m}^c \\ h_{5,m}^c \\ \vdots \end{bmatrix}$$

$$\sum_n \tilde{T}_{n,m}^c \sin^2\theta\sin n\theta = \sum_n h_{n,m}^c \sin n\theta$$

$$\Rightarrow \underbrace{\begin{bmatrix} 3-diagonal \\ matrix \end{bmatrix}}_{\text{rhs of } (24d)} \begin{bmatrix} \tilde{T}_{1,m}^c \\ \tilde{T}_{3,m}^c \\ \tilde{T}_{5,m}^c \\ \vdots \end{bmatrix} = \begin{bmatrix} h_{1,m}^c \\ h_{3,m}^c \\ h_{5,m}^c \\ \vdots \end{bmatrix}$$

It should be explained why above equations are the same as those the author derived.

[3] The largest wavenumber (truncation wavenumber) in (8b) should be determined considering the grid structure, grid[0] or grid[1] or grid[-1] to make completeness of spectral expansion issue clear (refer to Cheong et al. 2004).

[4] Section 2.12 presents the Laplacian operator and the Poisson's equation. The accuracy of the new DFS method for these basic operators and others such as biharmonic diffusion operator should be addressed with detailed error magnitude. Also important is the global mean associated with the Poisson's equation.

[5] One of the most basic test case is the cosine-bell advection, which is not included in this manuscript. The test case is simple but useful to demonstrate the advantage and disadvantages of a numerical method.

[6] It is very nice to see that the simulations are carried out without numerical instability even without horizontal diffusion. The author may address why it is possible. Is it due to the diffusive property of the semi-Lagrangian?

[7] **Figure 2**. The problem setting is quite strange. In principle, any scalar function with $m>0$ should vanish at poles. Nevertheless, the 'original' function is given to have value of unity at north pole. Therefore, the computation and comparison are not meaningful.

[8] **Figure 5**. Result of DFS0 appears to be too much smooth compared to DFS_old. Why is it?

**Specific minor comments**

[1] The right hand side of (25) should be represented with matrix-vector multiplication as in the left hand side.

[2] Terms associated with $\tilde{T}_m^{c,J}$ and $\tilde{T}_m^{s,J}$ in (36) do not appear in (37). The reason should be explained.

[3] Equation (B1) can be found in Cheong 2000a.

End of file.

---

## Referee Comment (RC2)

**Review: Improved double Fourier series on a sphere ...**

General Comment:

The Double Fourier Series on the sphere approach historically is compared to the spherical harmonic representation, e.g. Boer & Steinberg (1975) state: "the ease of calculation using the FFT must be weighed against the "pole problem" and the fact that the expansion functions are not orthogonal with respect to area weighting on the sphere." In this sense, the author makes a very good case resolving the "pole problem" with the improved DFS, and the result of the very high resolution Galewsky test is convincing. I like the approach and comparison to spherical harmonics and I certainly recommend publication. But I have a few comments below that should be addressed.

[1]As I understand it the author approximates orthogonality and it is strictly satisfied for $m <= 3$. We know that the spherical harmonics are eigensolutions of the barotropic vorticity equation on the sphere. Hough functions as eigensolutions of Laplace's tidal equation go further in providing eigensolutions for atmospheric Rossby and gravity wave dynamics of the linearised primitive equations (see also a recent article Vasylkevych & Zagar, Q J R Meteorol Soc. 2021;147:1989?2007). So the DFS approach is still a deviation from the "normal mode approach", although this does not mean of course that the fundamental modes of predictability are not well captured. It would nevertheless be interesting so see Rossby and gravity waves somehow in separation and the effect of the numerical method on these (and in combination with the time-stepping on propagation speed). For example, one could force a particular set of Hough modes for the shallow water equations and test this ? Also, if the Galewsky test was set simultaneously in the southern and the northern hemisphere (possibly with an onset delay between the two), would one expect more differences between DFS and SH ?

[2] The author focusses on computational performance and memory requirement as a primary reason for the advantage of the DFS to the SH. However, while memory may be an issue in the short term, there is a large trend towards very high memory nodes in the future. The fast Legendre transform (FLT) also effectively reduces the memory requirement. It is wrong to state that the FLT compromises accuracy (see also Wedi, 2014 Phil. Trans. R. Soc. A 372: 20130289). The FLT is not very sensitive to the threshold epsilon which is essentially a 'selection of zeros (that do not need to be computed)' threshold parameter.

[3] In above paper there is also the case made for cubic truncation with increasing resolution, which aligns much better the cost of grid point and spectral calculations in global numerical weather prediction (NWP) and climate models. In terms of grid point calculations, the DFS operates on a latitude-longitude grid (and associated area weighting of the basis functions is similar) and the author makes a case for applying the spherical harmonics filter in practical applications (even if in the idealised cases shown this may not be necessary). But in today's models 50 percent of the computations are done in grid point space (e.g. physics computations, SL advection). Can the DFS be applied on a reduced grid saving 50 percent of these grid point computations ? What would it do to the accuracy ? What is the average grid distance near the pole at 1km resolution with the

latitude longitude grid ?

[4] The differentiation and advantage of methods will not be decided, as done in this paper, on the computational order of complexity ($n^3$ or $n^2 \log n$) but rather on how well a method computes on accelerators such as GPUs and how well the method parallelises across MPI nodes. In my experience on GPUs the matrix-matrix multiplies, regardless of complexity, are so fast that these parts of the computation in practice reduce to $c*n^2$ where $c- > 0$. Can the author say more about the inherent parallelism that may be exploited in the DFS method in comparison to SH models ?

[5] Necessarily this paper needs the mathematical detail to be able to reproduce the results. This is good. However, it would be much more readable by potentially moving some of the repetition (scalar/vector) into the appendix and pointing just out where there are differences. The semi-implicit time-integration is fairly standard now and could also be in the appendix or supplementary material ? I think it otherwise distracts to much from what is new and what is already elsewhere in the literature. I think this will improve readership of this article. Also the discussion on the different grids is a little confusing and may be moved, eg to the beginning of the article ?

[6] On the least squares approach for spherical harmonics, the author may want to refer to appendix A6 in www.ppsloan.org/publications/StupidSH36.pdf, a nice article on SH.

[7] The author compares with a specific implementation of the SH, and refers to the oscillations near the pole and the cost benefit as advantages of the DFS method. This may be read as rather general statements, e.g. in the abstract: "The new DFS model is faster than the SH model, especially at high resolutions, and gives almost the same results." Is the SH model the one used operationally at JMA ? The author states that the oscillations near the pole can be overcome in the SH method, so how relevant is Figure 10 in comparing the two methods ? I would also suggest to slightly rephrase the abstract in light of this comment.

Specific comments:

page 2, line 7 "The FFT ... and is much faster than the fast Legendre transform", this is not necessarily true (e.g. with GPUs) and is purely judged on computational complexity, I would delete this phrase or qualify.

page 10, on the least-squares approach, how do you know the solution found is unique ?

page 25, line 4-5, what does the choice eq 84 imply more generally and thus not strictly satisfying the differential relationships stated ?

page 33, Test case 5 topography can give rise to spectral ringing in the SH model, what happens in the DFS model, did the author test this ?

Figure 11, Does the SH model employ the FLT, what would this look like if it had (e.g. based on complexity arguments?) ? This could be stated explicitly in the caption.

---

## Author Comment (AC1)

[Reply on RC1]

We are very grateful to the referee #1 for the useful comments. They help to improve the paper quality. Below, we will reply to the specific comments.

**Specific major comments**

**[1] One of the most important aspect of the paper is that the DFS expansion coefficients ($T_{n,m}^c$, $T_{n,m}^s$) are calculated based on the least square method, as is shown section 2.3. I am afraid, however, that the derivation procedure does not seem to be the least square method which is required for determining the expansion coefficients. The residual function here is defined using the difference between the spectral representation of the function ($T_m^c$, $T_m^s$) two different sets of DFS. The fact that, for the spherical harmonics model (SHM), the spectral coefficients are determined in the least square sense on the spherical domain is explained as below:**

$$E \equiv \int_0^\pi \left[ T_m^c(\theta) - \sum_{n=m}^N c_n P_n^m(\theta) \right]^2 \sin\theta \, d\theta \tag{Q1}$$

$$\frac{\partial E}{\partial c_n} = 0 \, \ldots \, \text{ least squares error} \tag{Q2}$$

$$\int_0^\pi P_n^m(\theta) \left[ T_m^c(\theta) - \sum_{n=m}^N c_n P_n^m(\theta) \right] \sin\theta \, d\theta = 0 \tag{Q3}$$

$$\therefore c_n = \int_0^\pi P_n^m(\theta) T_m^c(\theta) \sin\theta \, d\theta \tag{Q4}$$

$$\left[ \because \int_0^\pi P_n^m(\theta) P_{n'}^m(\theta) \sin\theta \, d\theta = \delta_{nn'} \right] \tag{Q5}$$

That is, $c_n$ is obtained with least squared error on the sphere.

**The least square method for the SHM in the present study is different from above equations.**

From Eq. (R18) below (fixed from Eq (37)), Eqs. (30) and (31), and from $\partial E^{\text{SH}} / \partial T_{n,m}^{c,\text{SH}} = 0$, we can derive Eq. (34). Equation (34) is similar to Eq. (Q4), but $\tilde{T}_m^{c,J}(\theta)$ is used instead of $T_m^c(\theta)$. The referee probably considers the difference between $\tilde{T}_m^{c,J}(\theta)$ and $T_m^c(\theta)$ as a problem. The use of Eq. (34) is one way to calculate the expansion coefficients of Legendre functions (e.g., Sneeuw and Bun, 1996), although the coefficients calculated from Eq. (34) are different from those calculated from Eq. (Q4) with Gaussian quadrature (or Clenshaw–Curtis quadrature), that is,

$$\int_0^\pi P_n^m(\theta) T_m^c(\theta) \sin\theta \, d\theta \neq \int_0^\pi P_n^m(\theta) \tilde{T}_m^{c,J}(\theta) \sin\theta \, d\theta. \tag{R1}$$

On the other hand, when we use sine series (for odd $m$) as basis functions, $\tilde{T}_{n,m}^c$ in Eq. (19) is calculated by the forward Fourier sin transform as

$$\tilde{T}_{n,m}^{c} = \frac{2}{\pi} \int_{0}^{\pi} \sin n\theta \, T_{m}^{c}(\theta) d\theta. \tag{R2}$$

From Eq. (19),

$$\frac{2}{\pi} \int_{0}^{\pi} \sin n\theta \, \tilde{T}_{m}^{c,J}(\theta) d\theta = \tilde{T}_{n,m}^{c} \tag{R3}$$

is also derived. Therefore,

$$\frac{2}{\pi} \int_{0}^{\pi} \sin n\theta \, T_{m}^{c}(\theta) d\theta = \frac{2}{\pi} \int_{0}^{\pi} \sin n\theta \, \tilde{T}_{m}^{c,J}(\theta) d\theta \left( = \tilde{T}_{n,m}^{c} \right) \tag{R4}$$

is satisfied. Eq. (R4) is in contrast to Eq. (R1).

From Eqs. (R4) and (A2c),

$$\frac{2}{\pi} \int_{0}^{\pi} \sin^2 \theta \sin n\theta \, T_{m}^{c}(\theta) d\theta = \frac{2}{\pi} \int_{0}^{\pi} \sin^2 \theta \sin n\theta \, \tilde{T}_{m}^{c,J}(\theta) d\theta \tag{R5}$$

is derived. From Eqs. (R4) and (R5), $\tilde{T}_{m}^{c,J}(\theta)$ in Eq. (R7) below for odd $m$ ($\geq 3$) can be changed to $T_{m}^{c}(\theta)$. The same thing can be said for other $m$. Therefore, in Eq. (21), $\tilde{T}_{m}^{c,J}(\theta)$ can be changed to $T_{m}^{c}(\theta)$. This is important and we will describe this in the paper (or in the supplement).

**[2] It looks like that the equations (24a)-(24d) are just algebraic equations resulted from simply multiplying $\sin \theta$ or $\sin^2 \theta$ or $\sin^4 \theta$ to the same equation. For instance, in the case of odd $m$ ($\geq 3$), it follows:**

$$T_{m}^{c}(\theta) = \sum_{n} T_{n,m}^{c} \sin^2 \theta \sin n\theta$$

$$= \sum_{n} \tilde{T}_{n,m}^{c} \sin n\theta \tag{Q6}$$

$$\sin^2 \theta \, T_{m}^{c}(\theta) = \sum_{n} T_{n,m}^{c} \sin^4 \theta \sin n\theta$$

$$= \sum_{n} \tilde{T}_{n,m}^{c} \sin^2 \theta \sin n\theta \tag{Q7}$$

$$\sum_{n} T_{n,m}^{c} \sin^4 \theta \sin n\theta = \sum_{n} h_{n,m}^{c} \sin n\theta \tag{Q8}$$

$$\Rightarrow \underbrace{\begin{bmatrix} 5 - diagonal \\ matrix \end{bmatrix}}_{\text{lhs of (24d)}} \begin{bmatrix} T_{1,m}^{c} \\ T_{3,m}^{c} \\ T_{5,m}^{c} \\ \vdots \end{bmatrix} = \begin{bmatrix} h_{1,m}^{c} \\ h_{3,m}^{c} \\ h_{5,m}^{c} \\ \vdots \end{bmatrix} \tag{Q9}$$

$$\sum_{n} \tilde{T}_{n,m}^{c} \sin^2 \theta \sin n\theta = \sum_{n} h_{n,m}^{c} \sin n\theta \tag{Q10}$$

$$\Rightarrow \underbrace{\begin{bmatrix} 3-diagonal \\ matrix \end{bmatrix}}_{\text{rhs of } (24d)} \begin{bmatrix} \tilde{T}^c_{1,m} \\ \tilde{T}^c_{3,m} \\ \tilde{T}^c_{5,m} \\ \vdots \end{bmatrix} = \begin{bmatrix} h^c_{1,m} \\ h^c_{3,m} \\ h^c_{5,m} \\ \vdots \end{bmatrix} \quad \textbf{(Q11)}$$

**It should be explained why above equations are the same as those the author derived.**

For instance, in the case of odd $m$ ($\geq 3$), the following equation are obtained from Eqs. (27a), (28), (29) and (21):

$$\frac{1}{2\pi^2} \int_0^{2\pi} \int_0^{\pi} (\sin^2 \theta \sin n\theta) \cos m\lambda \left[ \sum_{m=0}^{M} \left( T_m^{c,N}(\theta) - \tilde{T}_m^{c,J}(\theta) \right) \cos m\lambda \right] d\theta d\lambda = 0$$

$$(n = 1, \dots, N-2) \quad \text{(R6)}$$

From Eq. (R6),

$$\frac{1}{2\pi} \int_0^{\pi} \sin^2 \theta \sin n\theta \left( T_m^{c,N}(\theta) - \tilde{T}_m^{c,J}(\theta) \right) d\theta = 0 \quad \text{(R7)}$$

is derived. From Eqs. (R7), (8a) and (19), we derive

$$\frac{1}{2\pi} \int_0^{\pi} \sin n\theta \left[ \sum_{n=1}^{N-2} T_{n,m}^c \sin^4 \theta \sin n\theta - \sum_{n=1}^{J} \tilde{T}_{n,m}^c \sin^2 \theta \sin n\theta \right] d\theta = 0. \quad \text{(R8)}$$

Here, we define $h^c_{n,m}$ and $\tilde{h}^c_{n,m}$ as

$$\sum_{n=1}^{N+2} h^c_{n,m} \sin n\theta \equiv \sum_{n=1}^{N-2} T^c_{n,m} \sin^4 \theta \sin n\theta, \quad \text{(R9)}$$

$$\sum_{n=1}^{J+2} \tilde{h}^c_{n,m} \sin n\theta \equiv \sum_{n=1}^{J} \tilde{T}^c_{n,m} \sin^2 \theta \sin n\theta. \quad \text{(R10)}$$

From Eqs. (R8), (R9) and (R10), we obtain

$$h^c_{n,m} = \tilde{h}^c_{n,m} \qquad (n = 1, \dots, N-2). \quad \text{(R11)}$$

Thus, Eqs. (Q8), (Q9), (Q10) and (Q11) are derived. This is interesting. We will describe this in the paper or in the supplement.

**[3] The largest wavenumber (truncation wavenumber) in (8b) should be determined considering the grid structure, grid[0] or grid[1] or grid[-1] to make completeness of spectral expansion issue clear (refer to Cheong et al. 2004).**

Thank you for the advice. In the old DFS method using the Cheong's basis functions in Eq. (6) with Grid [0], $N$ can be up to $J^0 - 1$ for $m = 0$, and $J^0$ for $m \neq 0$ (Cheong et al. 2004). In the new DFS method using Eqs. (8), (19), (53) and (54) with Grid [0], $N$ can be up to $J^0 - 1$ for each $m$ because Eq. (66) is used for a scalar variable when $m$ is even and vector components when $m$ is

odd. In the new DFS method with Grid [1], $N$ can be up to $J^0 - 1$ for each $m$ because Eq. (69) is used for a scalar variable when $m$ is odd and vector components when $m$ is even. In the new DFS method with Grid [−1], $N$ can be up to $J^0 - 1$ for $m \geq 2$ because Eq. (69) is used for a scalar variable when $m\ (\geq 3)$ is odd and vector components when $m\ (\geq 2)$ is even. With Grid [−1], $N$ can be up to $J^0 - 2$ for $m = 0,1$ because Eqs. (73) and (75) are used for a scalar variable when $m = 1$, and vector components when $m = 0$. When we use the model at the resolution $N = 63$ and $J^0 = 64$ using the new DFS method in Sect. 5, we set $N = 63$ for each $m$ except that we set $N = 62$ for $m = 0,1$, with Grid [−1]. We will describe this in the paper.

**[4] Section 2.12 presents the Laplacian operator and the Poisson's equation. The accuracy of the new DFS method for these basic operators and others such as biharmonic diffusion operator should be addressed with detailed error magnitude. Also important is the global mean associated with the Poisson's equation.**

The Laplacian operator and the Poisson equation is represented as
$$g = \nabla^2 f. \tag{R12}$$
Here, the global mean of $g$ must be zero. Before calculating $f$ from a given $g$ in the Poisson equation, we should subtract the global mean from $g$ (Cheong 2000b). We will describe this in the paper.

We examined the accuracy of the old and new DFS methods for the Laplacian operator in Eq. (R12) and the Helmholtz equation
$$h = (1 - \varepsilon \nabla^2) f. \tag{R13}$$
Here, we give the function $f$ as
$$f = \begin{cases} \dfrac{H}{4}\left(1 + \cos\dfrac{\pi r}{R}\right)^2 & \text{if } r < R \\ 0 & \text{if } r \geq R \end{cases}, \tag{R14}$$
$$r = a \cos^{-1}[\sin\phi_c \sin\phi + \cos\phi_c \cos\phi \cos(\lambda - \lambda_c)], \tag{R15}$$
where $H = 1000$, $R = a/3$, $\phi$ is latitude, $\lambda$ is longitude, $a$ is the radius of the earth and $r$ is the distance between $(\lambda, \phi)$ and the center $(\lambda_c, \phi_c) = (3\pi/2, \pi/2 - 0.05)$. The function $f$ is similar to the cosine bell in the Williamson test case 1, but $(1 + \cos\pi r/R)$ is squared so that the second derivative of $f$ is continuous. To easily calculate the exact values of $\nabla^2 f$, the center is temporarily set to the North Pole, that is, $(\lambda_c, \phi_c) = (0, \pi/2)$ and $r = a\cos^{-1}[\sin\phi] = a\theta$, where $\theta$ is colatitude. At this time, $g$ is calculated as follows:
$$g = \nabla^2 f = \frac{1}{a^2}\left[\frac{1}{\sin^2\theta}\frac{\partial^2 f}{\partial\lambda^2} + \frac{1}{\sin\theta}\frac{\partial}{\partial\theta}\left(\sin\theta\frac{\partial f}{\partial\theta}\right)\right]$$
$$= -\frac{\cos\theta}{\sin\theta}\frac{H}{2a^2}\frac{\pi a}{R}\left[\left(1 + \cos\frac{\pi r}{R}\right)\sin\frac{\pi r}{R}\right] + \frac{H}{2a^2}\left(\frac{\pi a}{R}\right)^2\left[\sin^2\frac{\pi r}{R} - \left(1 + \cos\frac{\pi r}{R}\right)\cos\frac{\pi r}{R}\right]. \tag{R16}$$

Equation (R16) is satisfied at any position of the center. The function $h$ is calculated by

$$h = (1 - \varepsilon\nabla^2)f = f - \varepsilon g, \tag{R17}$$

where $\varepsilon = 0.01a^2$, and $f$ and $g$ are given by Eqs. (R14) and (R16).

To examine the accuracy for the Laplacian operator, $f$ is given by (R14), and $\nabla^2 f$ is calculated from $f$ with the old DFS method, the new DFS method (See Sect. 2.12) and the SH method. The calculated values are compared with the exact values of $\nabla^2 f$ in Eq. (16). Here, the exact values of $\nabla^2 f$ are truncated by the forward transform followed by the inverse transform in order to see the error that does not include the error due to the wavenumber truncation. Table R1 shows the root mean squared errors (RMSEs) between the calculated values and the exact values. The differences in error between the methods are small, and which is better depends on the resolution. Table R2 shows the global mean value of calculated $\nabla^2 f$. The exact value of the global mean of $\nabla^2 f$ is zero. In Table R2, the global mean values calculated with each method are very close to zero.

To examine the accuracy of the solution of the Helmholtz equation, $h$ is given in Eq. (R17) and the Helmholtz equation in Eq. (R13) is solved with the old DFS method, the new DFS method (See Sect. 2.13) and the SH method. The calculated values are compared with the exact solution $f$ in Eq. (R14). The exact values of $f$ are also truncated as described above. Table R3 shows the RMSEs between the calculated values and the exact values. The difference in error between the methods are small, and which is better depends on the resolution and the arrangement of the grid points. This kind of accuracy test is important, and we will describe this in the paper or in the supplement.

| | Old DFS [0] | New DFS [0],[1],[−1] | SH [Gaussian] |
|---|---|---|---|
| 128x64, $N$=42 | $7.479772 \times 10^{-13}$ | $4.179854 \times 10^{-13}$ $4.089208 \times 10^{-13}$ $4.772618 \times 10^{-13}$ | $5.278089 \times 10^{-13}$ |
| 320x160, $N$=106 | $4.033842 \times 10^{-14}$ | $4.302001 \times 10^{-14}$ $4.242989 \times 10^{-14}$ $4.244875 \times 10^{-14}$ | $3.933373 \times 10^{-14}$ |
| 1920x960, $N$=639 | $4.407625 \times 10^{-16}$ | $6.478465 \times 10^{-16}$ $6.514562 \times 10^{-16}$ $6.478465 \times 10^{-16}$ | $4.257350 \times 10^{-16}$ |

Table R1. The RMSEs of Laplacian operator calculation ($\nabla^2 f$) with the old and new DFS methods and the SH method. In the new DFS method, the results of Grid [0], Grid [1] and Grid [−1] are shown in this order. The number of longitudinal ($I$) and latitudinal ($J$) grid points is shown in the form $I \times J$. $N$ is the truncation wavenumber.

|  | Old DFS [0] | New DFS [0],[1],[-1] | SH [Gaussian] |
|---|---|---|---|
| 128x64, $N$=42 | $2.766693 \times 10^{-26}$ | $1.474223 \times 10^{-26}$ $-1.303193 \times 10^{-25}$ $-6.176935 \times 10^{-25}$ | $-4.644813 \times 10^{-27}$ |
| 320x160, $N$=106 | $-1.647899 \times 10^{-26}$ | $2.035640 \times 10^{-26}$ $3.226382 \times 10^{-25}$ $-4.130437 \times 10^{-25}$ | $3.004992 \times 10^{-26}$ |
| 1920x960, $N$=639 | $-4.215168 \times 10^{-26}$ | $-3.068185 \times 10^{-26}$ $2.356491 \times 10^{-24}$ $4.677050 \times 10^{-25}$ | $9.469697 \times 10^{-27}$ |

Table R2. Same as Table R1 except that the global mean values of calculated $\nabla^2 f$ are shown.

|  | Old DFS [0] | New DFS [0],[1],[-1] | SH [Gaussian] |
|---|---|---|---|
| 128x64, $N$=42 | $3.814822 \times 10^{-2}$ | $3.597815 \times 10^{-2}$ $3.731604 \times 10^{-2}$ $3.856617 \times 10^{-2}$ | $3.335080 \times 10^{-2}$ |
| 320x160, $N$=106 | $8.785639 \times 10^{-4}$ | $8.781645 \times 10^{-4}$ $8.080622 \times 10^{-4}$ $8.091810 \times 10^{-4}$ | $1.531216 \times 10^{-3}$ |
| 1920x960, $N$=639 | $2.193258 \times 10^{-6}$ | $2.193256 \times 10^{-6}$ $1.937225 \times 10^{-6}$ $1.945943 \times 10^{-6}$ | $1.918862 \times 10^{-6}$ |

Table R3. Same as Table R1 except that the RMSEs of the solution of the Helmholtz equation are shown.

**[5] One of the most basic test case is the cosine-bell advection, which is not included in this manuscript. The test case is simple but useful to demonstrate the advantage and disadvantages of a numerical method.**

The cosine-bell advection test case is certainly one of the most basic test cases. Since the advection equation is highly non-linear, it is challenging to solve the equation on the longitude-latitude grid using the Eulerian scheme with the DFS spectral method instead of the semi-Lagrangian scheme. We have run the Williamson test case 1 simulating the cosine-bell advection in the old DFS, new DFS, and SH Eulerian models. The advection equation is integrated by the leap-frog scheme with the Robert-Asselin time filter (Robert, 1969; Asselin, 1972) with a coefficient of 0.1. The horizontal diffusion is not used, but the zonal Fourier filter is used in the old and new DFS methods. In Eq. (76),

the value $M_0 = 20$ is used in the DFS shallow water models. However, the larger the value $M_0$ is, the higher the longitudinal resolution around the pole is. Because of this, when the Eulerian scheme is used and $M_0$ is large, a timestep must be very short due to the CFL condition. Therefore $M_0$ should be as small as possible. We have tested $M_0 = 0$, but this degrades the result of the Williamson test case 1. We have also tested $M_0 = 1$ and this result is good. Therefore, we use $M_0 = 1$ here.

Figure R1 shows the predicted height after a 12-day integration in the Williamson test case 1. The number of grid points is around $128 \times 64$. The truncation wavenumber $N$ is 42 because the 2/3 rule (Orszag, 1971) is used in order to avoid aliasing in the nonlinear advection term. The timestep is 30 minutes. The results for DFS [0], DFS [1], DFS [-1] and SH are very similar. Instability occurs in the old DFS model without horizontal diffusion. Table R4 shows the errors of the predicted height after a 12-day integration in Williamson test case 1 (See Fig. 1) in the models at the resolution $N$=42 with around $128 \times 64$ grid points, and Table R5 shows the same as Table R4 except that the resolution is $N = 639$ with around $1920 \times 960$ grid points and the timestep is 150 seconds. The errors are very close among the models. At the resolution $N = 639$, the new DFS model without horizontal diffusion is unstable when the timestep is 200 seconds. The SH model without horizontal diffusion is stable when the timestep was 240 seconds and unstable when the timestep is 300 seconds. One reason for this difference in timestep is probably that the longitudinal resolution near the poles is higher in the new DFS model than in the SH model when $M_0 = 1$. When the fourth order horizontal diffusion in Eq. (100) is used, the both new DFS and SH models are stable when the timestep is 240 seconds and are unstable when the timestep is 300 seconds. The old DFS model is unstable even when the same fourth order horizontal diffusion is used. Higher-order horizontal diffusion, which effectively smooths out the high wavenumber components, stabilizes the Eulerian old DFS model (Cheong, 2000b; Cheong et al., 2002).

These results are very important and we will describe this in the paper.

Asselin, R. A.: Frequency filter for time integrations. Mon. Wea. Rev., 100, 487–490, 1972.

Robert, A. J.: The integration of a low order spectal form of the primitive meteorological equations. J. Meteor. Soc. Japan, 44, 237–245, 1966.

[Figure]

Figure R1. Predicted height (m) after a 12-day integration in Williamson test case 1. In the upper figures, the black contour shows the predicted height, and the red contour shows the reference solution. In the lower figures, color shading shows the difference between the predicted height and the reference solution.

|  | DFS Grid[0] | DFS Grid[1] | DFS Grid[-1] | SH Gaussian grid |
|---|---|---|---|---|
| $L_1$ error | 0.16761 | 0.163979 | 0.164086 | 0.167435 |
| $L_2$ error | 0.115567 | 0.115594 | 0.115593 | 0.115539 |
| $L_{max}$ error | 0.09711 | 0.0934617 | 0.0934646 | 0.0954416 |

Table R4. The error of predicted height after a 12-day integration in Williamson test case 1 (See Fig. 1). The number of grid points is around $128 \times 64$, and the truncation wavenumber N is 42.

|  | DFS Grid[0] | DFS Grid[1] | DFS Grid[-1] | SH Gaussian grid |
|---|---|---|---|---|
| $L_1$ error | 0.00980851 | 0.00980841 | 0.00980842 | 0.00981044 |
| $L_2$ error | 0.00824238 | 0.00824238 | 0.00824238 | 0.00824238 |
| $L_{max}$ error | 0.0067255 | 0.0067281 | 0.0067281 | 0.00672429 |

Table R5. Same as Table R4 except that the number of grid points is around $1920 \times 960$, and the truncation wavenumber $N$ is 639.

**[6] It is very nice to see that the simulations are carried out without numerical instability even without horizontal diffusion. The author may address why it is possible. Is it due to the diffusive property of the semi-Lagrangian?**

One reason is due to the stability of the semi-Lagrangian scheme. Especially, the old DFS method

needs to use the semi-Lagrangian scheme for stability without horizontal diffusion. The new DFS method probably does not need to use the semi-Lagrangian scheme from the results obtained in the reply to the comment [5] above. We will add this explanation in the paper.

**[7] Figure 2. The problem setting is quite strange. In principle, any scalar function with $m > 0$ should vanish at poles. Nevertheless, the 'original' function is given to have value of unity at north pole. Therefore, the computation and comparison are not meaningful.**

In Figure 2, Grid [0] is used and there are no grid points at the poles. The original values are set to one at the grid points north of 30°N, and the value at the north pole is zero at the same time. This means that the original values abruptly change around the north pole. In the method of Cheong using Eq. (6), this abrupt change around the poles causes the large oscillations when the truncation wavenumber $N$ for even $m \, (\geq 2)$ is lower than $J$. This explanation is necessary, and we will modify the explanation in Sect. 3.

**[8] Figure 5. Result of DFS0 appears to be too much smooth compared to DFS_old. Why is it?**

This is because the least squares method is used to calculate the expansion coefficients in DFS [0]. In Fig. 2, the number of latitudinal grid points $J$ is 64, and the truncation wavenumber $N$ is 63. Figure R2(a) shows the same figure as Fig. 2(a) except that $J = 64$ and $N = 63$. When $N = 63$, in the old DFS method using the Cheong's basis functions in Eq. (6), we set $N = 63$ for $m = 0$, and $N = 64$ for $m \neq 0$. Because $N = J$ for even $m \, (\geq 2)$, the forward transform followed by the inverse transform does not change the original values at the grid points, and the oscillations do not appear in the old DFS method. Fig. R2(b) is the same as Fig. R2(a) except that it also shows the values between grid points calculated from the expansion coefficients by using Eq. (6) or Eq. (8). The large oscillations appear in the old DFS method, and it makes the latitudinal derivative at the grid points large. This is probably one reason that high zonal wavenumber noise appears near the noise in the old DFS model without horizontal diffusion. In the new DFS method, only small ocillations appear in Fig. R2(b) because the error is minimized by using the least squares method when calculating the expansion coefficients. We think that this is important and we will add this explanation and Fig. R2 in this paper.

[Figure]

Figure. R2. Change in values due to the meridional wavenumber truncation for even $|m| (\geq 2)$. We use Grid [0] with the number of latitudinal grid points $J = 64$. Original values (black) are meridionally transformed from grid space to spectral space, truncated with $N = 63$, and transformed back from spectral space to grid space. Green: Cheong's expansion method. Red: the new expansion method. (a) Values at the grid points. (b) Values at the grid points and between grid points calculated from the expansion coefficients.

**Specific minor comments**

**[1] The right hand side of (25) should be represented with matrix-vector multiplication as in the left hand side.**

Thank you for the advice. We will modify Eq. (25).

**[2] Terms associated with $\widetilde{T}_{1,m}^{c,J}$ and $\widetilde{T}_{1,m}^{c,J}$ in (36) do not appear in (37). The reason should be explained.**

I am sorry this is a typo. The right equation for Eq. (37) is

$$E^{\text{SH}} = \frac{1}{2} \int_0^\pi \left[ \left( T_{m=0}^{c,\text{SH},N}(\theta) - \tilde{T}_{m=0}^{c,J}(\theta) \right)^2 + \frac{1}{2} \sum_{m=1}^{M} \left( T_m^{c,\text{SH},N}(\theta) - \tilde{T}_m^{c,J}(\theta) \right)^2 \right. \tag{R18}$$
$$\left. + \frac{1}{2} \sum_{m=1}^{M} \left( T_m^{s,\text{SH},N}(\theta) - \tilde{T}_m^{s,J}(\theta) \right)^2 \right] \sin\theta \, d\theta.$$

**[3] Equation (B1) can be found in Cheong 2000a.**

We have forgotten to cite Cheong 2000a. We will fix it.

---

## Author Comment (AC2)

[Reply on RC2]

We are very grateful to the referee #2 for the useful comments. They help to improve the paper quality. Below, we will reply to the specific comments.

**[1]As I understand it the author approximates orthogonality and it is strictly satised for $m \leq 3$. We know that the spherical harmonics are eigensolutions of the barotropic vorticity equation on the sphere. Hough functions as eigensolutions of Laplace's tidal equation go further in providing eigensolutions for atmospheric Rossby and gravity wave dynamics of the linearised primitive equations (see also a recent article Vasylkevych & Zagar, Q J R Meteorol Soc. 2021;147:1989?2007). So the DFS approach is still a deviation from the "normal mode approach", although this does not mean of course that the fundamental modes of predictability are not well captured. It would nevertheless be interesting so see Rossby and gravity waves somehow in separation and the effect of the numerical method on these (and in combination with the time-stepping on propagation speed). For example, one could force a particular set of Hough modes for the shallow water equations and test this ? Also, if the Galewsky test was set simultaneously in the southern and the northern hemisphere (possibly with an onset delay between the two), would one expect more differences between DFS and SH ?**

Since we do not know how to perform simulations using a particular set of Hough modes, we have run Williamson's test case 6 to simulate Rossby-Haurwitz wave. Figure R3 shows the predicted height after a 14-day integration in Williamson test case 6. The error is similar among the old and new DFS models using Grid [0] and the SH model. The error in the new DFS model using Grid [1] is smallest. This may be because Grid [1] has grid points at the poles, where the minimum height exists, and on the equator, where the maximum height exists.

To set the Galewsky test simultaneously in the southern and the northern hemisphere, we have run the Galewsky-like test case using the north-south symmetric initial conditions created by adding the north-south opposite distribution of height and winds with perturbations in the southern and the northern hemisphere. Figure R4 shows the predicted vorticity after a 6-day integration in the Galewsky-like test case at 1.3 km resolution. The result in the new DFS model using Grid [0] is almost the same as in the SH model. Figure R5 shows Kinetic energy spectrum of horizontal winds after a 6-day integration in the Galewsky-like test case. The results are almost the same for the DFS model and the SH model, but small oscillations appear near the truncation wavenumber in the SH model. The differences between DFS and SH in the Galewsky-like test case are very similar to those in the Galewsky test case.

We will describe these results in the paper or in the supplement.

[Figure]

Fig. R3. Predicted height (m) after a 14-day integration in Williamson test case 6. (a) New DFS model with Grid [0]. (b) New DFS model with Grid [1]. (c) New DFS model with Grid [−1]. (d) Old DFS model with Grid [0]. (e) SH model. (f) SH model at high resolution, which is regarded as the reference solution. The number of longitudinal ($I$) and latitudinal ($J$) grid points is shown in the form $I \times J$. $N$ is the truncation wavenumber. Color shading shows the error with respect to the reference solution.

[Figure]

Fig. R4. Predicted vorticity $(s^{-1})$ after a 6-day integration in the Galewsky-like test case with north-south symmetry. (a) The new DFS model with Grid [0], and (b) the SH model at 1.3 km resolution with $30720 \times 15360$ grid points and $N = 10239$.

[Figure]

Fig. R5. Kinetic energy spectrum of horizontal winds $(m^2 s^{-2})$ after a 6-day integration in the Galewsky test case. (a) Results of the models with $30720 \times 15360$ grid points. The colors blue and red represent the models using SH and DFS with Grid [0], respectively. (b) As (a), but showing the high-wavenumber region.

**[2] The author focusses on computational performance and memory requirement as a primary reason for the advantage of the DFS to the SH. However, while memory may be an issue in the short term, there is a large trend towards very high memory nodes in the future. The fast Legendre transform (FLT) also effectively reduces the memory requirement. It is wrong to state that the FLT compromises accuracy (see also Wedi, 2014 Phil. Trans. R. Soc. A 372: 20130289). The FLT is not very sensitive to the threshold epsilon which is essentially a 'selection of zeros (that do not need to be computed)' threshold parameter.**

Thank you for the information about FLT. I will describe in the paper that the FLT effectively reduces the memory usage. I have understood that the FLT does not compromises accuracy. I will describe instead that "the threshold parameter affecting the accuracy-cost balance in the FLT is chosen so that a loss of accuracy is sufficiently small." I will also cite Wedi (2014) in the paper.

**[3] In above paper there is also the case made for cubic truncation with increasing resolution, which aligns much better the cost of grid point and spectral calculations in global numerical weather prediction (NWP) and climate models. In terms of grid point calculations, the DFS operates on a latitude-longitude grid (and associated area weighting of the basis functions is similar) and the author makes a case for applying the spherical harmonics filter in practical applications (even if in the idealised cases shown this may not be necessary). But in today's models 50 percent of the computations are done in grid point space (e.g. physics computations, SL advection). Can the DFS be applied on a reduced grid saving 50 percent of these grid point computations ? What would it do to the accuracy ? What is the average grid distance near the pole at 1km resolution with the latitude longitude grid ?**

Our DFS models do not need the spherical harmonics filter. In our atmospheric DFS model, we use the same fourth-order hyper-diffusion (see Sect. 2.14) as in our atmospheric SH model (Yoshimura and Matsumura, 2005; Yoshimura, 2012). In the paper, we choose a latitude-longitude grid with a zonal Fourier filter in the shallow water model for the simplicity of the source code. However, we can use a reduced grid instead of the latitude-longitude grid. In our atmospheric model using SH or DFS, we use the reduced grid. We will reflect this in the paper.

If we use an octahedral reduced grid, about 50 % of the grid point computations are saved. When the cubic truncation is used, the octahedral reduced grid probably does not reduce the accuracy in the DFS and SH models. The longitudinal grid distance near the pole at 1km resolution with the latitude-longitude grid is about 0.08 m. However, the resolution near the pole is not so high when the zonal Fourier filter is used.

**[4] The differentiation and advantage of methods will not be decided, as done in this paper, on the computational order of complexity (*n^3* or *n^2 log n*) but rather on how well a method computes on accelerators such as GPUs and how well the method parallelises across MPI nodes. In my experience on GPUs the matrix-matrix multiplies, regardless of complexity, are so fast that these parts of the computation in practice reduce to *c\*n^2* where *c-> 0*. Can the author say more about the inherent parallelism that may be exploited in the DFS method in comparison to SH models ?**

Thank you for providing information about execution speed on the GPU. Since we have not performed our model on the GPU yet, we have described the computational order of complexity and the elapsed time on the CPU in the paper. We should enable the execution of our model on the GPU and test it in the future. The execution on the GPU is a big issue and is beyond the scope of this paper.

Our shallow water model in the paper is parallelized only with OpenMP. We think that when we perform the DFS or SH model with the truncation wavenumber $N$ and $K$ vertical levels using $T$ threads with OpenMP and $P$ processes with MPI, $T \times P$ can be up to $(N + 1) \times K$. If the FFT, the Legendre transform, and the matrix calculations such as Eq. (26) are parallelized with OpenMP, $T \times P$ can be more than $(N + 1) \times K$. These are the same between the DFS and SH models. In the future, we will write a paper about the DFS atmospheric model with MPI and OpenMP parallelization, where we will discuss the parallelization. We will describe this simply.

**[5] Necessarily this paper needs the mathematical detail to be able to reproduce the results. This is good. However, it would be much more readable by potentially moving some of the repetition (scalar/vector) into the appendix and pointing just out where there are differences. The semi-implicit time-integration is fairly standard now and could also be in the appendix or supplementary material ? I think it otherwise distracts to much from what is new and what is already elsewhere in the literature. I think this will improve readership of this article. Also the discussion on the different grids is a little confusing and may be moved, eg to the beginning of the article ?**

Thank you for the advice. We will move some of the repetition and the semi-implicit time-integration into the appendix. We will also move the discussion on the different grids to the beginning of the article.

**[6] On the least squares approach for spherical harmonics, the author may want to refer to appendix A6 in www.ppsloan.org/publications/StupidSH36.pdf, a nice article on SH.**

Thank you for introducing us to a nice article. It is interesting to minimize some form of variational function instead of just the standard least-squares error in order to minimize ringing artifacts.

**[7] The author compares with a specific implementation of the SH, and refers to the oscillations near the pole and the cost benefit as advantages of the DFS method. This may be read as rather general statements, e.g. in the abstract: "The new DFS model is faster than the SH model, especially at high resolutions, and gives almost the same results." Is the SH model the one used operationally at JMA ? The author states that the oscillations near the pole can be overcome in the SH method, so how relevant is Figure 10 in comparing the two methods ? I would also suggest to slightly rephrase the abstract in light of this comment.**

Thank you for the suggestion. In the paper, the SH model is the shallow water model using SH. The oscillations near the pole appear in the old DFS model, and do not appear in the new DFS model and SH model. In the SH shallow water model, small oscillations appear near the truncation wavenumber in the kinetic energy spectrum, but not in the new DFS model. This problem in the SH model can probably be solved by using the vector harmonic transform as described in the paper. We will modify "The new DFS model is faster than the SH model, especially at high resolutions, and gives almost the same results" in the abstract as follows:

The shallow water model using the new DFS method is faster than the shallow water model using SH, especially at high resolutions, and gives almost the same results, except that small oscillations near the truncation wavenumber in the kinetic energy spectrum appear only in the SH model. This problem in the SH model can probably be solved by using the vector harmonic transform which is similar to the vector transform using the least-squares method in the DFS model.

**Specific comments:**
**page 2, line 7 "The FFT ... and is much faster than the fast Legendre transform", this is not necessarily true (e.g. with GPUs) and is purely judged on computational complexity, I would delete this phrase or qualify.**

I will delete "much" and describe "The FFT ... and is faster than the fast Legendre transform" in the paper.

**page 10, on the least-squares approach, how do you know the solution found is unique ?**

The new DFS meridional basis functions are not orthogonal but independent. Therefore, by using Gram-Schmidt orthogonalization, the basis functions can be converted to orthogonalized basis

functions, where the latitudinal weight is constant. By using the least squares method with the orthogonalized basis functions, expansion coefficients are calculated uniquely by inner product similar to Eq. (34) except that the latitudinal weight is constant. Thus, the solution function like $T_m^{c,SH,N}(\theta)$ in Eq. (37) is obtained uniquely. This unique solution function is the same as that calculated by the least-squares method with the original non-orthogonal basis functions.

**page 25, line 4-5, what does the choice eq 84 imply more generally and thus not strictly satisfying the differential relationships stated ?**

We use Eq. (84) obtained from the Galerkin method instead of the equation obtained from the least-squares method when calculating $f$ from a given $g$. We always use the Galerkin method regardless of whether the equation obtained from the Galerkin method is the same as that obtained from the least-squares method. It does not mean "not strictly satisfying the differential relationships stated". We will add the explanation in the paper.

**page 33, Test case 5 topography can give rise to spectral ringing in the SH model, what happens in the DFS model, did the author test this ?**

In test case 5, spectral ringing seen in the test caes 5 in Jakob-Chien et al. (1995) does not appear in our SH and DFS models. This is probably because the semi-Lagrangian scheme improves numerical stability.

Jakob-Chien, R., Hack, J. J., and Williamson, D. L.: Spectral transform solutions to the shallow water test set, J. Comput. Phys. 119, 164-187, doi:10.1006/jcph.1995.1125, 1995.

**Figure 11, Does the SH model employ the FLT, what would this look like if it had (e.g. based on complexity arguments?) ? This could be stated explicitly in the caption.**

The operation count of FLT is proportional to $N^2(\log N)^3$, and that of the usual Legendre transform is proportional to $N^3$. However, since we do not know their proportional coefficients, it is difficult to estimate the elapsed time when FLT is used. We will describe in the paper that the FLT will reduce the execution time of the Legendre transform.

---

## Author Response (AR1)

[Reply on Referee Comment 1]

**Comments to the manuscript**

**General comments**

**This paper deals with the Double Fourier Series (DFS) on the sphere where new DFS functions are used to represent the variables on the global domain. Discretization procedures for the spatial differentiations, elliptic equations, and the shallow water models are shown in some detail using the trigonometric identities. Combining the DFS method with the semi-Lagrangian time differentiation, the paper provided simulation results for a couple of shallow water model test cases, including the standard test cases of Williamson et al. (1992). The author emphasizes that the DFS functions in the present study improves the simulation results over the DFS models in the previous studies. However, it is hard to be convinced of the improvement of the new DFS method due to less rigorous assessment of the solution method for differential operators such as elliptic equations, least square method, and also limited test case results. Specific comments are shown below.**

We are very grateful to the referee #1 for the useful comments, which help to improve the paper quality. We reply to the specific comments below.

**Specific major comments**

**[1] One of the most important aspect of the paper is that the DFS expansion coefficients ($T_{n,m}^c$, $T_{n,m}^s$) are calculated based on the least square method, as is shown section 2.3. I am afraid, however, that the derivation procedure does not seem to be the least square method which is required for determining the expansion coefficients. The residual function here is defined using the difference between the spectral representation of the function ($T_m^c$, $T_m^s$) two different sets of DFS. The fact that, for the spherical harmonics model (SHM), the spectral coefficients are determined in the least square sense on the spherical domain is explained as below:**

$$E \equiv \int_0^\pi \left[ T_m^c(\theta) - \sum_{n=m}^N c_n P_n^m(\theta) \right]^2 \sin\theta \, d\theta \tag{Q1}$$

$$\frac{\partial E}{\partial c_n} = 0 \ldots \text{ least squares error} \tag{Q2}$$

$$\int_0^\pi P_n^m(\theta) \left[ T_m^c(\theta) - \sum_{n=m}^N c_n P_n^m(\theta) \right] \sin\theta \, d\theta = 0 \tag{Q3}$$

$$\therefore c_n = \int_0^\pi P_n^m(\theta) T_m^c(\theta) \sin\theta \, d\theta \tag{Q4}$$

$$\left[ \because \int_0^\pi P_n^m(\theta) P_{n'}^m(\theta) \sin\theta \, d\theta = \delta_{nn'} \right] \tag{Q5}$$

That is, $c_n$ is obtained with least squared error on the sphere.

**The least square method for the SHM in the present study is different from above equations.**

We have modified the description about the derivation of Eq. (Q4) (Eq. (37) in the revised paper) using the least-squares method. The referee probably considers the use of $\tilde{T}_m^{c,N}(\theta)$ instead of $T_m^c(\theta)$ as a problem. In the SH method, $\tilde{T}_m^{c,N}(\theta)$ can be used instead of $T_m^c(\theta)$ (e.g., Sneeuw and Bun, 1996), although the values of $T_{n,m}^{c,SH}$ calculated from $\tilde{T}_m^{c,N}(\theta)$ are different from those calculated from $T_m^c(\theta)$. In the new DFS method, using $\tilde{T}_m^{c,N}(\theta)$ instead of $T_m^c(\theta)$ is not a problem, because the values of $T_{n,m}^c$ calculated using $\tilde{T}_m^{c,N}(\theta)$ are the same as those calculated using $T_m^c(\theta)$. This is important, and we have described this in Sect. 3.4 and Appendix D in the paper. Moreover, we have changed $\left( \sum_{m=0}^M \tilde{T}_m^{c,N}(\theta) \cos m\lambda + \sum_{m=1}^M \tilde{T}_m^{s,N}(\theta) \sin m\lambda \right)$ to $T(\lambda, \theta)$ in the residual in Eqs. (25) and (39).

**[2] It looks like that the equations (24a)-(24d) are just algebraic equations resulted from simply multiplying $\sin\theta$ or $\sin^2\theta$ or $\sin^4\theta$ to the same equation. For instance, in the case of odd $m$ ($\geq 3$), it follows:**

$$T_m^c(\theta) = \sum_n T_{n,m}^c \sin^2\theta \sin n\theta$$

$$= \sum_n \tilde{T}_{n,m}^c \sin n\theta \tag{Q6}$$

$$\sin^2\theta \, T_m^c(\theta) = \sum_n T_{n,m}^c \sin^4\theta \sin n\theta$$

$$= \sum_n \tilde{T}_{n,m}^c \sin^2\theta \sin n\theta \tag{Q7}$$

$$\sum_n T_{n,m}^c \sin^4\theta \sin n\theta = \sum_n h_{n,m}^c \sin n\theta \tag{Q8}$$

$$\Rightarrow \underbrace{\begin{bmatrix} 5 - diagonal \\ matrix \end{bmatrix}}_{\text{lhs of } (24d)} \begin{bmatrix} T_{1,m}^c \\ T_{3,m}^c \\ T_{5,m}^c \\ \vdots \end{bmatrix} = \begin{bmatrix} h_{1,m}^c \\ h_{3,m}^c \\ h_{5,m}^c \\ \vdots \end{bmatrix} \tag{Q9}$$

$$\sum_n \tilde{T}_{n,m}^c \sin^2\theta \sin n\theta = \sum_n h_{n,m}^c \sin n\theta \tag{Q10}$$

$$\Rightarrow \underbrace{\begin{bmatrix} 3-diagonal \\ matrix \end{bmatrix}}_{\text{rhs of } (24d)} \begin{bmatrix} \widetilde{T}^{\mathrm{c}}_{1,m} \\ \widetilde{T}^{\mathrm{c}}_{3,m} \\ \widetilde{T}^{\mathrm{c}}_{5,m} \\ \vdots \end{bmatrix} = \begin{bmatrix} h^{\mathrm{c}}_{1,m} \\ h^{\mathrm{c}}_{3,m} \\ h^{\mathrm{c}}_{5,m} \\ \vdots \end{bmatrix} \qquad (Q11)$$

**It should be explained why above equations are the same as those the author derived.**

This is interesting. We have changed the derivation of Eq. (30), and we have explained in Appendix E in the paper why above equations are the same as those we derived.

**[3] The largest wavenumber (truncation wavenumber) in (8b) should be determined considering the grid structure, grid[0] or grid[1] or grid[-1] to make completeness of spectral expansion issue clear (refer to Cheong et al. 2004).**

Thank you for the advice. In the old DFS method using the Cheong's basis functions in Eq. (6) with Grid [0], the upper limit of $N$ is $J-1$ for $m=0$, and $J$ for $m \neq 0$ (Cheong et al. 2004). We have described in Sect. 1 in the paper as follows:

2. The meridional truncation with $N = J-1$ and $N' = J$ is used, which enables to reconstruct accurately the given grid-data with the expansion coefficients (Cheong et al., 2004) and avoid the error due to the meridional truncation.

We have also described the upper limit of $N$ in the new DFS method in Appendix C. We have also cited Cheong et al. (2004) in Sect. 2.

**[4] Section 2.12 presents the Laplacian operator and the Poisson's equation. The accuracy of the new DFS method for these basic operators and others such as biharmonic diffusion operator should be addressed with detailed error magnitude. Also important is the global mean associated with the Poisson's equation.**

This kind of accuracy test is important, and we have described the results of the test in Sect. 5. The global mean associated with the Poisson's equation is also important, and we have described in Sect. 3.7 as follows:

In Eq. (65) ($g = \nabla^2 f$), the global mean of $g$ must be zero because the global mean of the right-hand side of Eq. (65) is zero. Before calculating $f$ from a given $g$ in the Poisson equation, we should subtract the global mean from $g$ (Cheong 2000b). See Eq. (G1) about the calculation of the global mean.

**[5] One of the most basic test case is the cosine-bell advection, which is not included in this manuscript. The test case is simple but useful to demonstrate the advantage and disadvantages of a numerical method.**

Thank you for the advice. Since the advection equation is highly non-linear, it is challenging to solve the equation in the longitude-latitude grid using the Eulerian scheme with the DFS spectral method instead of the semi-Lagrangian scheme. We have run the Williamson test case 1 simulating the cosine-bell advection in the Eulerian and semi-Lagrangian models to examine the advantage and the disadvantage of the DFS methods. The Eulerian advection model using the new DFS method is stable without horizontal diffusion, but that using the old DFS method is not stable. The results of this test are important, and we have described the results in Sect. 6.3. We also change the title of this paper to "Improved double Fourier series on a sphere and its application to a semi-implicit semi-Lagrangian shallow water model and an Eulerian advection model".

**[6] It is very nice to see that the simulations are carried out without numerical instability even without horizontal diffusion. The author may address why it is possible. Is it due to the diffusive property of the semi-Lagrangian?**

One reason is due to the stability of the semi-Lagrangian scheme, which avoids numerical instability due to the nonlinear advection term. Especially, the old DFS method needs to use the semi-Lagrangian scheme for stability. We have described this in Sect.1.

We have shown in Sect. 4 that the error due to the wavenumber truncation is small in the new DFS method because the expansion coefficients are calculated using the least-squares method (or the Galerkin method) to minimize the error. This improves the stability of the model using the new DFS method. In the new DFS method, probably it is not always necessary to use the semi-Lagrangian scheme from the results of the Williamson test case 1. We have added the following description in Sect. 7:

In the old DFS model, the use of the semi-Lagrangian scheme is important for numerical stability. On the other hand, the advection model using the new DFS method is stable, even when the Eulerian scheme is used instead of the semi-Lagrangian scheme. The Eulerian shallow water model using the new DFS method is also likely to be stable, although we have not tested it yet.

**[7] Figure 2. The problem setting is quite strange. In principle, any scalar function with $m > 0$ should vanish at poles. Nevertheless, the 'original' function is given to have value of unity at north pole. Therefore, the computation and comparison are not meaningful.**

In Figure 2, Grid [0] is used and there are no grid points at the poles. The original (initial) values are set to one at the grid points north of 30°N (except for the North pole). Since the values at the North and South poles are zero due to the pole conditions in Eq. (3), the initial values abruptly change around the North pole. This explanation is necessary, and we have modified the explanation in Sect. 4.

**[8] Figure 5. Result of DFS0 appears to be too much smooth compared to DFS_old. Why is it?**

This is because the least squares method (or the Galerkin method) is used to calculate the expansion coefficients in new DFS [0]. In DFS_old, the latitudinal derivative at the grid points can be unrealistically large even when $N' = J$ for even $m \ (\geq 2)$ as described in Sect. 4 (Fig. 3b). The new DFS expansion method with the least-squares method does not have this problem. We have described this in Sect. 6.5.

**Specific minor comments**
**[1] The right hand side of (25) should be represented with matrix-vector multiplication as in the left hand side.**

Thank you for the advice. We have modified it.

**[2] Terms associated with $\widetilde{T}_{1,m}^{c,J}$ and $\widetilde{T}_{1,m}^{c,J}$ in (36) do not appear in (37). The reason should be explained.**

I am sorry this is a typo. $\widetilde{T}_{m=0}^{c,SH,J}(\theta)$ should be $\widetilde{T}_{m=0}^{c,J}(\theta)$, and $\widetilde{T}_{m=0}^{s,SH,J}(\theta)$ should be $\widetilde{T}_{m=0}^{s,J}(\theta)$. We no more use this equation in the revised paper.

**[3] Equation (B1) can be found in Cheong 2000a.**

Thank you for the information. We forgot to cite Cheong 2000a. We have cited Cheong 2000a in Appendix G.

[Reply on Referee Comment 2]

**Review: Improved double Fourier series on a sphere ...**
**General Comment:**
**The Double Fourier Series on the sphere approach historically is compared to the spherical harmonic representation, e.g. Boer & Steinberg (1975) state: "the ease of calculation using the FFT must be weighed against the "pole problem" and the fact that the expansion functions are not orthogonal with respect to area weighting on the sphere." In this sense, the author makes a very good case resolving the "pole problem" with the improved DFS, and the result of the very high resolution Galewsky test is convincing. I like the approach and comparison to spherical harmonics and I certainly recommend publication. But I have a few comments below that should be addressed.**

We are very grateful to the referee #2 for the useful comments, which help to improve the paper quality. We reply to the specific comments below.

**[1]As I understand it the author approximates orthogonality and it is strictly satised for $m \leq 3$. We know that the spherical harmonics are eigensolutions of the barotropic vorticity equation on the sphere. Hough functions as eigensolutions of Laplace's tidal equation go further in providing eigensolutions for atmospheric Rossby and gravity wave dynamics of the linearised primitive equations (see also a recent article Vasylkevych & Zagar, Q J R Meteorol Soc. 2021;147:1989?2007). So the DFS approach is still a deviation from the "normal mode approach", although this does not mean of course that the fundamental modes of predictability are not well captured. It would nevertheless be interesting so see Rossby and gravity waves somehow in separation and the effect of the numerical method on these (and in combination with the time-stepping on propagation speed). For example, one could force a particular set of Hough modes for the shallow water equations and test this ? Also, if the Galewsky test was set simultaneously in the southern and the northern hemisphere (possibly with an onset delay between the two), would one expect more differences between DFS and SH ?**

Since we do not know how to perform simulations using a particular set of Hough modes, we have run Williamson test case 6 to simulate Rossby-Haurwitz wave. We have described the results of the test case 6 in Sect. 6.6.

To set the Galewsky test simultaneously in the southern and the northern hemisphere, we have run the Galewsky-like test case using the north-south symmetric initial conditions created by adding the north-south opposite distribution of height and winds with perturbations in the southern and the northern hemisphere. We have described the results of the Galewsky-like test case in the supplement.

The results of the Galewsky-like test case are similar to those of the Galewsky test case.

**[2] The author focusses on computational performance and memory requirement as a primary reason for the advantage of the DFS to the SH. However, while memory may be an issue in the short term, there is a large trend towards very high memory nodes in the future. The fast Legendre transform (FLT) also effectively reduces the memory requirement. It is wrong to state that the FLT compromises accuracy (see also Wedi, 2014 Phil. Trans. R. Soc. A 372: 20130289). The FLT is not very sensitive to the threshold epsilon which is essentially a 'selection of zeros (that do not need to be computed)' threshold parameter.**

Thank you for the information about FLT. I have described in the paper that the FLT effectively reduces the memory usage. I have understood that the FLT does not compromises accuracy. I have described instead that "In the fast Legendre transform, the threshold parameter affecting the accuracy-cost balance is chosen so that a loss of accuracy is sufficiently small." I have also cited Wedi (2014) in the paper.

**[3] In above paper there is also the case made for cubic truncation with increasing resolution, which aligns much better the cost of grid point and spectral calculations in global numerical weather prediction (NWP) and climate models. In terms of grid point calculations, the DFS operates on a latitude-longitude grid (and associated area weighting of the basis functions is similar) and the author makes a case for applying the spherical harmonics filter in practical applications (even if in the idealised cases shown this may not be necessary). But in today's models 50 percent of the computations are done in grid point space (e.g. physics computations, SL advection). Can the DFS be applied on a reduced grid saving 50 percent of these grid point computations ? What would it do to the accuracy ? What is the average grid distance near the pole at 1km resolution with the latitude longitude grid ?**

Our DFS models do not need the spherical harmonics filter. In our atmospheric DFS model, we use the same fourth-order horizontal diffusion as in our atmospheric SH model (Yoshimura and Matsumura, 2005; Yoshimura, 2012). We have described this in Sect. 1.

We can use a reduced grid instead of the latitude-longitude grid. In our atmospheric DFS model, we use the reduced grid. In this study, we use the longitude–latitude grid with the zonal Fourier filter, not the reduced grid, for the simplicity of the source code. We have described this in APPENDIX F.

If we use an octahedral reduced grid (Malardel, 2016), about 50 % of the grid point computations are saved. When the cubic truncation is used, the octahedral reduced grid probably does not reduce the accuracy in the DFS and SH models. We have cited Malardel (2016) in APPENDIX F.

The longitudinal grid distance near the pole at 1km resolution with the latitude-longitude grid is about 0.08 m. The resolution near the pole is not so high when the zonal Fourier filter is used. Numerical instability occurs in some test cases in the old DFS model without the zonal Fourier filter, but stable integration is possible in all test cases shown here in the new DFS model even without the zonal Fourier filter as described in Sect. 6.2.

**[4] The differentiation and advantage of methods will not be decided, as done in this paper, on the computational order of complexity ($n$^3 or $n$^2 log $n$) but rather on how well a method computes on accelerators such as GPUs and how well the method parallelises across MPI nodes. In my experience on GPUs the matrix-matrix multiplies, regardless of complexity, are so fast that these parts of the computation in practice reduce to $c*n$^2 where $c$-> 0. Can the author say more about the inherent parallelism that may be exploited in the DFS method in comparison to SH models ?**

Thank you for providing information about execution speed on the GPU. Since we have not performed our model on the GPU yet, we have described the computational order of complexity and the elapsed time on the CPU in the paper. We should enable the execution of our model on the GPU and test it in the future. The execution on the GPU is a big issue and is beyond the scope of this paper. We have described in Sect. 7 as follows:

> We have executed our shallow water models on Intel CPUs. The execution on GPUs is one important topic, but we have not tested our models on GPUs because the execution on GPUs is not an easy task.

Our shallow water model in the paper is parallelized only with OpenMP. We think that when we perform the DFS or SH model with the truncation wavenumber $N$ and $K$ vertical levels using $T$ threads with OpenMP and $P$ processes with MPI, $T \times P$ can be up to $(N + 1) \times K$. If the FFT, the Legendre transform, and the matrix calculations such as Eq. (31) are parallelized with OpenMP, $T \times P$ can be more than $(N + 1) \times K$. These are the same between the DFS and SH models. In the future, we will write a paper about the DFS atmospheric model with MPI and OpenMP parallelization, where we will discuss the parallelization. We have described in Sect. 7 as follows:

> MPI parallelization is another important topic. However, in our shallow water models, we use only OpenMP parallelization, not MPI parallelization for the simplicity of the source code.
> We will describe the nonhydrostatic DFS model and the MPI parallelization in another paper after improving the nonhydrostatic dynamical core as needed.

**[5] Necessarily this paper needs the mathematical detail to be able to reproduce the results. This is good. However, it would be much more readable by potentially moving some of the repetition**

**(scalar/vector) into the appendix and pointing just out where there are differences. The semi-implicit time-integration is fairly standard now and could also be in the appendix or supplementary material ? I think it otherwise distracts to much from what is new and what is already elsewhere in the literature. I think this will improve readership of this article. Also the discussion on the different grids is a little confusing and may be moved, eg to the beginning of the article ?**

Thank you for the advice. We have moved the description about the semi-implicit time-integration scheme to the appendix I. We have moved the description about the different grids to Sect. 2. Some topics, such as the zonal Fourier filter and the Fourier discrete cosine and sine transforms, are also moved into the appendices. To avoid repetition, we use $S_{n,m}(\theta)$ to show the basis functions of the new DFS method, and use $T_m^{c(s)}(\theta)$ instead of $T_m^c(\theta)$ and $T_m^s(\theta)$. We also use the following expression to reduce the number of lines.

$$\begin{bmatrix} \chi(\lambda,\theta) \\ \psi(\lambda,\theta) \end{bmatrix} \cong \sum_{m=0}^{M} \begin{bmatrix} \chi_m^c(\theta) \\ \psi_m^c(\theta) \end{bmatrix} \cos m\lambda + \sum_{m=1}^{M} \begin{bmatrix} \chi_m^s(\theta) \\ \psi_m^s(\theta) \end{bmatrix} \sin m\lambda, \tag{45}$$

**[6] On the least squares approach for spherical harmonics, the author may want to refer to appendix A6 in www.ppsloan.org/publications/StupidSH36.pdf, a nice article on SH.**

Thank you for introducing us to a nice article. It is interesting to minimize some form of variational function instead of just the standard least-squares error in order to minimize ringing artifacts.

**[7] The author compares with a specific implementation of the SH, and refers to the oscillations near the pole and the cost benefit as advantages of the DFS method. This may be read as rather general statements, e.g. in the abstract: "The new DFS model is faster than the SH model, especially at high resolutions, and gives almost the same results." Is the SH model the one used operationally at JMA ? The author states that the oscillations near the pole can be overcome in the SH method, so how relevant is Figure 10 in comparing the two methods ? I would also suggest to slightly rephrase the abstract in light of this comment.**

Thank you for the suggestion. In the paper, the SH model is the shallow water model (or the advection model) using SH. The oscillations near the pole appear in the old DFS model, and do not appear in the new DFS model and SH model. In the SH shallow water model, small oscillations appear near the truncation wavenumber in the kinetic energy spectrum, but not in the new DFS model. The small oscillations in the SH model can probably be eliminated by using the vector harmonic transform as described in the paper. We will modify "The new DFS model is faster than the SH model, especially at high resolutions, and gives almost the same results" in the abstract as follows:

The shallow water model using the new DFS method is faster than that using SH, especially at high resolutions, and gives almost the same results, except that small oscillations near the truncation wavenumber in the kinetic energy spectrum appear only in the shallow water model using SH. This small oscillations in the SH model can probably be eliminated by using the vector harmonic transform which is similar to the vector transform using the least-squares method (or the Galerkin method) in the model using the new DFS method.

**Specific comments:**

**page 2, line 7 "The FFT ... and is much faster than the fast Legendre transform", this is not necessarily true (e.g. with GPUs) and is purely judged on computational complexity, I would delete this phrase or qualify.**

I will delete "much" and describe "The FFT ... and is faster than the fast Legendre transform" in the paper.

**page 10, on the least-squares approach, how do you know the solution found is unique ?**

We have described the reason that the solution is unique in the following part in Sect. 3.4:

The new DFS meridional basis functions $S_{n,m}(\theta)$ for each $m$ are not orthogonal but independent. Therefore, by using Gram-Schmidt orthogonalization, the basis functions can be converted to orthogonalized basis functions $S^{O}_{n,m}(\theta)$, …

… , we derive

$$T^{c(s),O}_{n,m} = \frac{1}{\pi} \int_0^\pi S^{O}_{n,m}(\theta) \tilde{T}^{c(s),N}_m(\theta) d\theta. \tag{43}$$

Thus, $T^{c(s),O}_{n,m}$ and $T^{c(s),O,N}_m(\theta)$ in Eqs. (43) and (42) are calculated uniquely. This unique solution $T^{c(s),O,N}_m(\theta)$ is the same as $T^{c(s),N}_m(\theta)$ in Eq. (29) obtained by the least-squares method with the non-orthogonal basis functions $S_{n,m}(\theta)$, because $S^{O}_{n,m}(\theta) \left(n = N_{\min,m}, …, N_{\max,m}\right)$ are the linear combination of $S_{n,m}(\theta) \left(n = N_{\min,m}, …, N_{\max,m}\right)$ for each $m$, and vice versa.

**page 25, line 4-5, what does the choice eq 84 imply more generally and thus not strictly satisfying the differential relationships stated ?**

We use Eq. (84) (Eq. (70) in the revised paper) obtained from the Galerkin method instead of the equation obtained from the least-squares method when calculating $f$ from a given $g$. Both the equations obtained from the Galerkin method and from the least-square method strictly satisfy $g^{N,M}(\lambda,\theta) = \nabla^2 f^{N,M}(\lambda,\theta)$ for $0 \le m \le 3$, but approximately satisfy the equation for $m \ge 4$.

Generally, it cannot be said that the least-squares method is superior to the Galerkin method or vice versa. We have described in Sect. 3.7. as follows.

> Generally it cannot be said that the least-squares method is superior to the Galerkin method or vice versa, and here we choose to use the Galerkin method because of the reason described above.

**page 33, Test case 5 topography can give rise to spectral ringing in the SH model, what happens in the DFS model, did the author test this ?**

In test case 5, spectral ringing seen in the test caes 5 in Jakob-Chien et al. (1995) does not appear in our SH and DFS models. This is probably because the semi-Lagrangian scheme improves numerical stability.

Jakob-Chien, R., Hack, J. J., and Williamson, D. L.: Spectral transform solutions to the shallow water test set, J. Comput. Phys. 119, 164-187, doi:10.1006/jcph.1995.1125, 1995.

**Figure 11, Does the SH model employ the FLT, what would this look like if it had (e.g. based on complexity arguments?) ? This could be stated explicitly in the caption.**

The operation count of FLT is proportional to $N^2(\log N)^3$, and that of the usual Legendre transform is proportional to $N^3$. However, since we do not know their proportional coefficients, it is difficult to estimate the elapsed time when FLT is used. We have added the description below in Sect. 6.8.

> If the fast Legendre transform, which requires only $N^2(\log N)^3$ operation, is used instead of the usual Legendre transform in the SH model, the difference of the elapsed time between the models will be reduced at high resolutions. We have not tested the fast Legendre transform yet because we do not have subroutines for the fast Legendre transform.

[Reply on Referee Comment 3]

**Comments on "Improved double Fourier series on a sphere and its application to a semi-implicit semi-Lagrangian shallow water model" by Hiromasa Yoshimura.**

**General comments**

**The author proposes an alternative formulation to alleviate the pole problem used in a shallow water model using double Fourier series. The paper presents specific forms for the gradient of a scalar, vectors, and Laplacian and Helmholtz operators. The comparison against spherical harmonics and the new double Fourier series provides insights into the properties of the former. I recommend the publication of the paper provided that the following concern is appropriately addressed.**

We are very grateful to the referee #3 for the useful comments, which help to improve the paper quality. We reply to the specific comments below.

**Major comments**

**As the other reviewers noted, the tests seem to be randomly chosen. I recommend the author to investigate at least the convergence (error vs horizontal resolution) and conservation (energy, vorticity, etc.). Avoid visual comparison where possible and conduct quantitative evaluation. The errors should be given in standard error norms by numerical values.**

Thank you for the advice. We have given the errors by numerical values at several horizontal resolutions in the tables of the paper. We have also given the conservation (height, energy, vorticity) in the tables of the supplement.

**The presentation of the paper can be improved. For example, Subsection 5.2 and 5.4 are short and 5.3 is of two long paragraphs. Author's intention for the tests should be provided. The author discusses the pole problem, but Figures 4, 7 and 9 are shown in longitude–latitude; Figure 9 omits the polar region. It would be nice to add a diagram to show the differences of expansion visually.**

Thank you for the advice. We have described our intention for the tests in the first part of Sect. 6 as follows:

We ran the Williamson test cases 1, 2, 5 and 6 (Williamson et al., 1992), and the Galewsky test case (Galewsky et al., 2004) in the model using the new DFS method described in Sect. 3, the model using the old DFS method of Yoshimura and Matsumura (2005), and the model using the SH method. By comparing the results of these model, we evaluated the old and new DFS methods.

We have shown the northern hemisphere region including the polar region in Fig. 12, and the global

region including the polar region in Fig. S1 in the supplement. Figure 2 explains the differences between the DFS expansion methods. We have also given the same figures for the SH expansion method for comparison. Sections 6.3 and 6.5 have become long because we have a lot of things to describe.

**Minor comments**

**Page 1, Line 21: more accurate rather than good**

We have changed "the accuracy in horizontal derivatives is good" to "the horizontal derivatives are accurate".

**Page 1, Line 25: O(N3) memory usage, unless calculated on-the-fly**

We have added "(unless the fast Legendre transform or on-the-fly computation of the associated Legendre functions shown below is not used)".

**Page 2, Line 4: Alternatively (avoid repeating the same word, another).**

We have changed the sentence to "Alternatively, we can use double Fourier series (DFS) as basis functions to reduce the operation count and the memory usage in the global spectral model."

---

## Author Response (AR2)

[Reply to Topical Editor report]

We are very grateful to the topical editor and the reviewers for their useful comments and the decision "Publish subject to minor revisions". Below we provide point-by-point responses to the comments.

**Dear Author,**

**Please consider the following points raised by the reviewer.**

**"The author made a good effort to make the paper more accessible and I appreciate the responses to my comments. Except one perhaps, if they already have a parallelised version, why not comment on the connectivity needed between processors (need all zonal values for zonal expansion but parallel over m, need xx for the meridional, parallel over xx ?).**

**But I appreciate this may be the subject of future work to refine, so looking forward to more information on this in the future.**

Thank you very much for the comments. In this paper, we use the DFS shallow water model. This model is parallelized with OpenMP, not with MPI, and can utilize only one node of a supercomputer. We have already developed the DFS global atmospheric model, which is parallelized with both OpenMP and MPI and can utilize many nodes. So, we will describe the parallelization for the DFS global atmospheric model in the future paper.

**I find some of the notation too much (e.g. upper case N,M and equation 15 (needed?), many of these could be dropped in my view, just stating that the prognostic variables are expressed as truncated series, this would make many equations a lot less busy.**

We also think that there are many equations and it is better if some of these can be dropped. On the other hand, we would like to distinguish between the original variable $T(\lambda, \theta)$ and the truncated variable $T^{N,M}(\lambda, \theta)$, because it is important that the values of $T^{N,M}(\lambda, \theta)$ are generally different from those of $T(\lambda, \theta)$. In our new DFS method, expansion coefficients are determined so that the difference between $T^{N,M}(\lambda, \theta)$ and $T(\lambda, \theta)$ is minimized.

**I think what is missing is a quick summary "cook book", which could take the form of (if I understood correctly):**

**(1) Define expansion in double Fourier series, with zonal expansion [equation (8)] and meridional expansion [equation (10)]**

**(2) Apply recursive formulas for the inverse transform from spectral to gridpoint [equations 13-14)**

**(3) For the direct transform (grid to spectral) using the new DFS method:**

**a) define new expansion (tilde values, equation 23), and evaluate coefficients by forward transform directly from the values at the gridpoints (see appendix B)**

**b) minimise (with variation of the unknown spectral expansion coefficients of 10) over the sphere the squared residual R defined in equation 25: R is the distance between the series approximations defined in 10 and 23 leading to conditions 26 and with 27 to equation 28, noting the orthogonality condition with respect to R.**

**c) Evaluate the unknown spectral coefficients using the resulting matching conditions with the known expansion coefficients (tilde), derived by inserting 10 (in lhs of 28) and 23 (in rhs of 28) leading to equations 30 and the evaluation procedure 31.**

> Thank you for the advice. The essential summary (cook book) of the new DFS method is useful, and we have added the essential summary (cook book) of the new DFS method in Sect. 3.10. In the last paragraph of the introduction (Sect. 2), we have added "Section 3 …, and includes the essential summary of the new DFS method". In the first part of Sect. 3, we have added "The essential summary (cook book) of the new DFS method is in Sect. 3.10". In Appendix I, we have added "(See Sect. 3.10 for the spectral transform with the new DFS method.)"

**Notably, above is slightly different to what is in the current manuscript, in particular the derivation of 29 with appendix D and the current definition of R. I find confusing even if it leads to the same result. Is my interpretation above incorrect ?**

> The reviewer's interpretation is basically right. Moreover, although $T_m^{c(s)}(\theta)$ and $\tilde{T}_m^{c(s),N}(\theta)$ are generally different, Eq. (D4) is satisfied. By substituting Eqs. (10) and (23) into Eq. (29), we lead Eq. (30). In Appendix E, we forgot to describe that Eqs. (10) and (23) are used, and we have changed "By using Eqs. (11)" to "By using Eqs. (10), (11), (23)". We have also corrected mistakes in Eq. (E2).

**Other comments:**

**The new title is very long, I am wondering if "Improved double Fourier series for the shallow water equations on the sphere" would be more fitting.**

> Thank you for the advice. Surely, the new title is long, and the shorter one is preferable. We have changed the title back to "Improved double Fourier series on a sphere and its application to a semi-implicit semi-Lagrangian shallow water model". The words "semi-implicit semi-Lagrangian" can be removed. However, since the explanation of time integration using a semi-implicit semi-Lagrangian scheme is included in the paper, we keep "semi-implicit semi-Lagrangian" in the title.

**The rest are details that belong in the abstract. In the abstract: "These small oscillations..."**

Thank you for the advice. We have removed "These small oscillations …" in the abstract. We have also changed "except that small oscillations" to "except that very small oscillations", because these oscillations are very small and seem to have almost no problem in practical use. We have also modified the parts related to the small oscillations a little in Sects. 6 and 7.

**I would also remove the sentence in the conclusions : "The Eulerian shallow water model using the new DFS method without horizontal diffusion is also likely to be stable, although we have not tested it yet."**

Thank you for the advice. We have removed this sentence.

**Overall the text has changed/moved a lot, and it is difficult to verify all the revised cross references, but I assume editorial work will check this once more.**

We have checked all the cross references.

**Otherwise I found the appendices helpful, albeit many. In the end it is justified as this serves as a documentation of a new dynamical core and a reference in comparison to the spectral transform models based on spherical harmonics.**

Thank you for the comment. We also agree with this comment.

**I would like to thank the author for his hard work on addressing the different test cases and questions related to the accuracy of the wave operator."**

Thank you for the comment. We are grateful to all the comments from the reviewers and the editor, which have been very useful to improve the paper.

**Non-public comments to the Author:**
**I very much subscribe with the reviewer about the need for an essential summary (a "cook book" as the reviewer says). I think it would greatly improve the manuscript from a pedagogical point of view, ie explaining the method in a nutshell for newcomers. This could go in the introduction, the conclusions, or both with the right choice of words and references.**

Thank you for the advice. We have put the essential summary (cook book) of the new DFS method in Sect. 3.10. In the introduction (Sect. 2), Sect. 3, and Appendix I, we have added references to the essential summary as described above.

**Please verify all the revised cross references, for citations and equations etc..**

We have verified all, and have corrected many minor mistakes. There was a mistake in the calculation of kinetic energy spectrum, so we have also modified Figs. 8, 10, 13 and S2. (We hope that there are no more mistakes.)

---

## Author Response (AR3)

[Reply to Topical Editor's comments]

We are very grateful to the topical editor for the useful comments and the decision "publish subject to technical corrections". Below we provide point-by-point responses to the comments.

**Comments to the author:**
**I run through the supplement and compiled and run the examples.**
**Notice I only succeeded in compiling them all with ifort and icc.**
**With gcc 8.3 I encountered the follwing error:**
**gcc -O3 -march=native -c mxallc.c**
**gcc: warning: couldn't understand kern.osversion '18.7.0**
**mxallc.c:20:20: fatal error: stdlib.h: No such file or directory**
**#include**
**^**
**compilation terminated.**
**It seems to be indicated as a possible gcc bug.**

Thank you very much for testing our source code.
The error may be because of a gcc bug, a wrong setting with gcc, or a gcc installation problem.

In my environment with gcc version 9.3.0 (Ubuntu 9.3.0-17ubuntu1~20.04), ispack-3.0.1 can be compiled successfully, even though the warning below appears:
gcc -O3 -march=native -c mxallc.c
mxallc.c: In function 'mxallc_':
mxallc.c:23:3: warning: ignoring return value of 'posix_memalign', declared with attribute warn_unused_result [-Wunused-result]
   23 |    posix_memalign(p, 64, (size_t)((size_t)*n *8));

**There are some minor typos:**
**in /Supplement_Yoshimura/Source_code/README.txt**
**it is written:**
**> cd netlib**
**isn't it actually:**
**"> cd bihar"?**
**In many places it is written "warer" instead of "water". Please run a search and correct all the instances.**

**Edit sw_dfs.F90 (shallow warer model) and adv_dfs.F90 (advection model).**

**Edit sw_dfs.F90 (shallow water model) and adv_dfs.F90 (advection model).**

**> ./sw_dfs (shallow warer model)**

**Edit sw_dfs.F90 (shallow warer model) and adv_dfs.F90 (advection model).**

**> ./sw_dfs (shallow warer model)**

Thank you for checking README.txt. We have corrected these typos.